# Microglia regulate GABAergic neurogenesis in prenatal human brain through IGF1

Diankun Yu[1,12✉], Samhita Jain[1,2,12], Andi Wangzhou[1], Beika Zhu[1], Wenyuan Shao[1], Elena J. Coley-O'Rourke[1], Stacy De Florencio[1], JaeYeon Kim[1,3,4], Jennifer Ja-Yoon Choi[5], Mercedes F. Paredes[1,3,4], Tomasz J. Nowakowski[1,3,6,7,8], Eric J. Huang[1,3,5,9,11] & Xianhua Piao[1,2,3,10✉]

GABAergic neurons are essential cellular components of neural circuits. Their abundance and diversity have increased significantly in the human brain, contributing to the expanded cognitive capacity of humans[1]. However, the developmental mechanism underlying the extended production of GABAergic neurons in the human brain remains elusive. Here we uncovered the microglial regulation of the sustained proliferation of GABAergic progenitors and neuroblasts in the human medial ganglionic eminence (hMGE). We showed that microglia are preferentially distributed in the proliferating zone and identified insulin-like growth factor 1 (IGF1) and its receptor IGR1R as the predicted top ligand–receptor pair underlying microglia–progenitor communication in the prenatal hMGE. Using our newly developed neuroimmune hMGE organoids, which mimic the hMGE cytoarchitecture and developmental trajectory, we demonstrated that microglia-derived IGF1 promotes progenitor proliferation and production of GABAergic neurons. Conversely, IGF1-neutralizing antibodies and *IGF1* knockout human embryonic stem-cell-induced microglia abolish the induced microglia-mediated progenitor proliferation. Together, these findings revealed a previously unappreciated role of microglia-derived IGF1 in promoting the proliferation of neural progenitors and the development of GABAergic neurons in the human brain.

In the adult human neocortex, about 25–50% of cortical neurons are γ-aminobutyric acid-containing (GABAergic) inhibitory interneurons[1–4]. They serve as the principal sources of cortical inhibitory input and have a crucial role in maintaining excitation–inhibition balance and functional rhythms in the brain[5,6]. Disruptions in interneuron number and function have been implicated in various neurological and psychiatric disorders, including autism spectrum disorder[7,8], epilepsy[9] and schizophrenia[10]. In the prenatal human brain, GABAergic interneurons are generated in ganglionic eminences of the ventral telencephalon[11–13]. The medial ganglionic eminence (MGE) gives rise to most parvalbumin-positive (PV[+]) and somatostatin-positive (SST[+]) cortical interneurons[7]. Human MGE (hMGE) possesses several unique features that are distinct from those of other species to meet the needs of a drastically expanded cerebral cortex. First, neurogenesis in hMGE is most active during the second and third trimesters[14]. This active neurogenesis is followed by extensive migration of GABAergic neurons around the periventricular zone in the neonatal stage for at least 6 months postnatally[15]. Second, young GABAergic neuroblasts are organized as DCX[+] cell-enriched nests (DENs) that have sustained proliferation up to the early postnatal stage. GABAergic neuroblasts within DENs are surrounded by NESTIN[+]

and SOX2[+] radial glia that extend from the ventricular zone and inner subventricular zone (iSVZ) to the outer subventricular zone (oSVZ)[11,14]. Third, the DCX[+]SOX2[+] neuroblasts inside DENs exhibit regional differences in their proliferation potentials, with those at the edge of DENs showing a higher Ki-67 labelling index[14]. Although these observations support the notion that DCX[+] neuroblasts in DENs have the capacity to undergo sustained neurogenesis in hMGE[11,14], the external environmental cues that regulate hMGE neurogenesis remain largely unclear.

Microglia, which are specialized tissue-resident macrophages in the central nervous system, are the primary immune cells in the brain parenchyma. Originating from the yolk sac and entering the brain during early gestation[16], microglia have been shown to be involved in brain development, including neurogenesis[17], angiogenesis[18,19], neuronal survival[20], myelination[21–24], synaptogenesis[25] and synaptic pruning[26–30]. It has been shown that microglia regulate PV[+] interneuron development and positioning[31] and lead to PV[+] interneuron deficits in setting of maternal immune activation in the mouse brain[32]. However, it remains unclear whether and how microglia affect the development of GABAergic neurons in the human brain. Here we show that during the late second trimester, the iSVZ and oSVZ of hMGE are populated

[1]Weill Institute for Neurosciences, University of California San Francisco, San Francisco, CA, USA. [2]Division of Neonatology, Department of Pediatrics, University of California San Francisco, San Francisco, CA, USA. [3]Eli and Edythe Broad Center of Regeneration Medicine and Stem Cell Research, University of California San Francisco, San Francisco, CA, USA. [4]Department of Neurology, University of California San Francisco, San Francisco, CA, USA. [5]Department of Pathology, University of California San Francisco, San Francisco, CA, USA. [6]Department of Neurological Surgery, University of California San Francisco, San Francisco, CA, USA. [7]Department of Anatomy, University of California San Francisco, San Francisco, CA, USA. [8]Department of Psychiatry and Behavioral Sciences, University of California San Francisco, San Francisco, CA, USA. [9]Pathology Service, San Francisco VA Health Care System, San Francisco, CA, USA. [10]Newborn Brain Research Institute, University of California San Francisco, San Francisco, CA, USA. [11]Present address: Department of Pathology & Immunology, Washington University School of Medicine, St. Louis, MO, USA. [12]These authors contributed equally: Diankun Yu, Samhita Jain. ✉e-mail: diankun.yu@ucsf.edu; xianhua.piao@ucsf.edu

with microglia that are near the radial glia and Ki-67[+] proliferating neuroblasts at the periphery of DENs. Single-nucleus transcriptomic studies of human brains from the late second trimester to the early postnatal stage uncovered insulin-like growth factor 1 (IGF1) and its receptor IGF1R as the top candidate pathways in mediating the communication between microglia and interneuron progenitors. Further studies using human MGE neuroimmune organoids (MGEOs), in which human embryonic stem-cell (hESC)-induced microglia (iMG) were transplanted into human pluripotent stem-cell (hPSC)-derived MGE organoids, support that the preferential accumulation of microglia in subventricular zone (SVZ) is related to progenitor proliferation, and that microglia-derived IGF1 promotes progenitor proliferation and interneuron production.

## Microglia distribution in hMGE

To study microglial function in hMGE development, we first surveyed the temporal and spatial distributions in the hMGE of neurotypical human postmortem brains from gestational week (GW) 15 to postnatal week 3 using IBA1 immunohistochemistry (IHC) (Fig. 1a–d). At GW15–17, microglia were sparsely sprinkled throughout hMGE (Fig. 1a,d). However, by GW22–25, there was a clear locational preference, with most microglia seen in iSVZ and oSVZ (Fig. 1b,d). Microglia in oSVZ accumulated in the area outside of DENs (oDENs) and encased the outer rim of DENs, with rare microglial processes extending into the inside of DENs (Fig. 1b,d). By GW39 to postnatal week 3, the density of microglia in the iSVZ and oSVZ of hMGE remained high, and more microglial processes could be identified in the inside of DENs than those in the late second trimester (Fig. 1c,d).

To further examine the spatial relationship between microglia and proliferating cells in hMGE during GW22–25, we performed a combined IHC with IBA1 and Ki-67, a marker for cell proliferation. We observed that microglia mostly resided in regions with more abundant proliferative progenitors (Fig. 1e–i). Specifically, many microglia were embedded within the Ki-67[+] proliferating progenitors in the iSVZ and oDENs of oSVZ (Fig. 1e,f). Through three-dimensional (3D) reconstruction of DENs using IMARIS software, we found that Ki-67[+] cells were largely clustered around the microglia on the outside and border of DENs, with a gradual decrease in the ratio of Ki-67[+] cells to Ki-67[−] cells as the distance to the microglia increased (Fig. 1g,j). Furthermore, Ki-67[+] progenitors in proximity to microglia were largely SOX2[+] progenitors, including SOX2[+]DCX[−] radial glia and SOX2[+]DCX[+] neuroblasts (Fig. 1e–g,k). To further document the spatial relationship between microglia and MGE progenitors, we performed IHC of IBA1, DCX, NESTIN and Ki-67. We found that microglia were in close contact with and often followed along NESTIN[+] radial glial fibres (Fig. 1l). In iSVZ, the microglia were mostly ramified and intermingled with several NESTIN[+] fibres (Fig. 1l(1–4)). Most microglia in oSVZ exhibited polarized morphology along the NESTIN[+] fibres, with their processes largely aligning and in close contact with those fibres (Fig. 1l(5–8)).

Distinct from what was observed in hMGE, the density of microglia was significantly higher in the ventricular zone of mouse MGE on embryonic days 14.5 (Extended Data Fig. 1a–c) and 16.5 (Extended Data Fig. 1d–f), coinciding with the higher density of active proliferating progenitors in the ventricular zone of mouse MGE (Extended Data Fig. 1g,h). This contrast between mouse MGE and hMGE highlights a potential species-specific mechanism for interneuron development.

## Cell communication in the developing human brain

To identify the molecular underpinnings of microglial regulation of extended neurogenesis during human interneuron development, we generated a single-nucleus RNA sequencing (snRNA-seq) dataset of the late embryonic stage (GW22–30) to the perinatal stage (postnatal weeks 2–3) using a droplet-based 10X Genomics platform (*n* = 6 donors;

Fig. 2a and Supplementary Tables 1 and 2). We focused on human tissues that contained ganglionic eminences, adjacent periventricular regions (containing the Arc, an area of SVZ composed of migratory interneurons[15]) and cortical regions where mature interneurons reside (Fig. 2a and Supplementary Table 1). Neurons were over-represented in snRNA-seq[33]. To better capture both neurons and glia, we adapted flow cytometry-based strategies to enrich microglia through PU.1[+] and oligodendrocyte lineage cells and MGE progenitors through OLIG2[+] in addition to DAPI[+] nuclei[34] (Fig. 2b and Supplementary Table 1). After quality control and doublet removal (Methods), we recovered 124,411 nuclei with a median of 7,341 unique molecular identifiers (UMIs) and 2,925 genes per cell. We then performed principal component analysis (PCA) of normalized read counts, followed by uniform manifold approximation and projection (UMAP) using Seurat v.5 (ref. 35). We identified 11 main cell clusters from our transcriptomic data profiles on the basis of the expression of canonical marker genes (Fig. 2c, Extended Data Fig. 2 and Supplementary Fig. 1). Notably, our results contained 17,342 cells in the microglia cluster and 17,667 cells in the cortical GABAergic interneuron cluster, which includes ganglionic eminence progenitors, young GABAergic interneurons and mature GABAergic interneurons (Extended Data Fig. 2).

To predict cell-type-specific interactions between microglia and other cell types in the developing hMGE, we leveraged a highly curated database of receptor–ligand interactions to predict possible interactions among different types of cells[36] (Supplementary Fig. 2). This approach also revealed development-related pathways by comparative analysis of cell–cell communications at the late embryonic and perinatal stages (Supplementary Fig. 3). To gain insight into the interaction involved in the microglial regulation of cortical interneurons, we extracted these ligand–receptor pairs with statistical significance (Fig. 2d). Notably, IGF1 and IGF1R constitute the ligand–receptor pair with the highest communication probability, which was predominantly observed during the embryonic stages (Fig. 2d). Chord, violin and feature plots showed that IGF1 was predominantly derived from microglia during the embryonic stage, although it was also generated by interneurons during the perinatal stage (Fig. 2e,f and Extended Data Fig. 3). To clarify which interneuron population expresses IGF1, we conducted interneuron subcluster analysis and identified unique populations, namely radial glia, MGE neuroblasts, MGE-derived young interneurons, MGE-derived mature interneurons, caudal ganglionic eminence (CGE)-derived young interneurons and CGE-derived mature interneurons, on the basis of unbiased clustering and canonical markers (Fig. 2g and Extended Data Fig. 4). We observed that IGF1 was mainly expressed by mature interneurons but was barely detected in interneuron progenitors, including radial glia and MGE neuroblasts (Fig. 2h). These results indicate that microglia are a key source of IGF1 during interneuron proliferation in developing hMGE. Accordingly, IGF1R was highly expressed in interneurons, particularly at the embryonic stage (Fig. 2e,f and Extended Data Fig. 3). Subcluster analysis further confirmed the high expression levels of IGF1R in interneuron progenitors and young interneurons (Fig. 2g,h). IHC showed that in GW22–25 hMGE, IGF1 was specifically expressed in microglia (Fig. 2i), whereas IGF1R was widely expressed in progenitors in the hMGE (Fig. 2j). Taken together, our data indicate that IGF1 is primarily derived from microglia during the embryonic stage and has the potential to support interneuron development in hMGE.

## HPSC-derived MGEOs recapitulate hMGE development

To investigate the role of microglia and microglial IGF1 in hMGE development, we generated hPSCs, including both human induced pluripotent stem cells (hiPSCs) and hESC-derived ventral organoids, referred to as MGEOs, adapted from established protocols[37,38] (Fig. 3a). As expected, our MGEOs had robust NKX2.1[+] and Ki-67[+] ventricular zone-like rosettes at 6 weeks of age (Fig. 3b) and demonstrated sequential expression

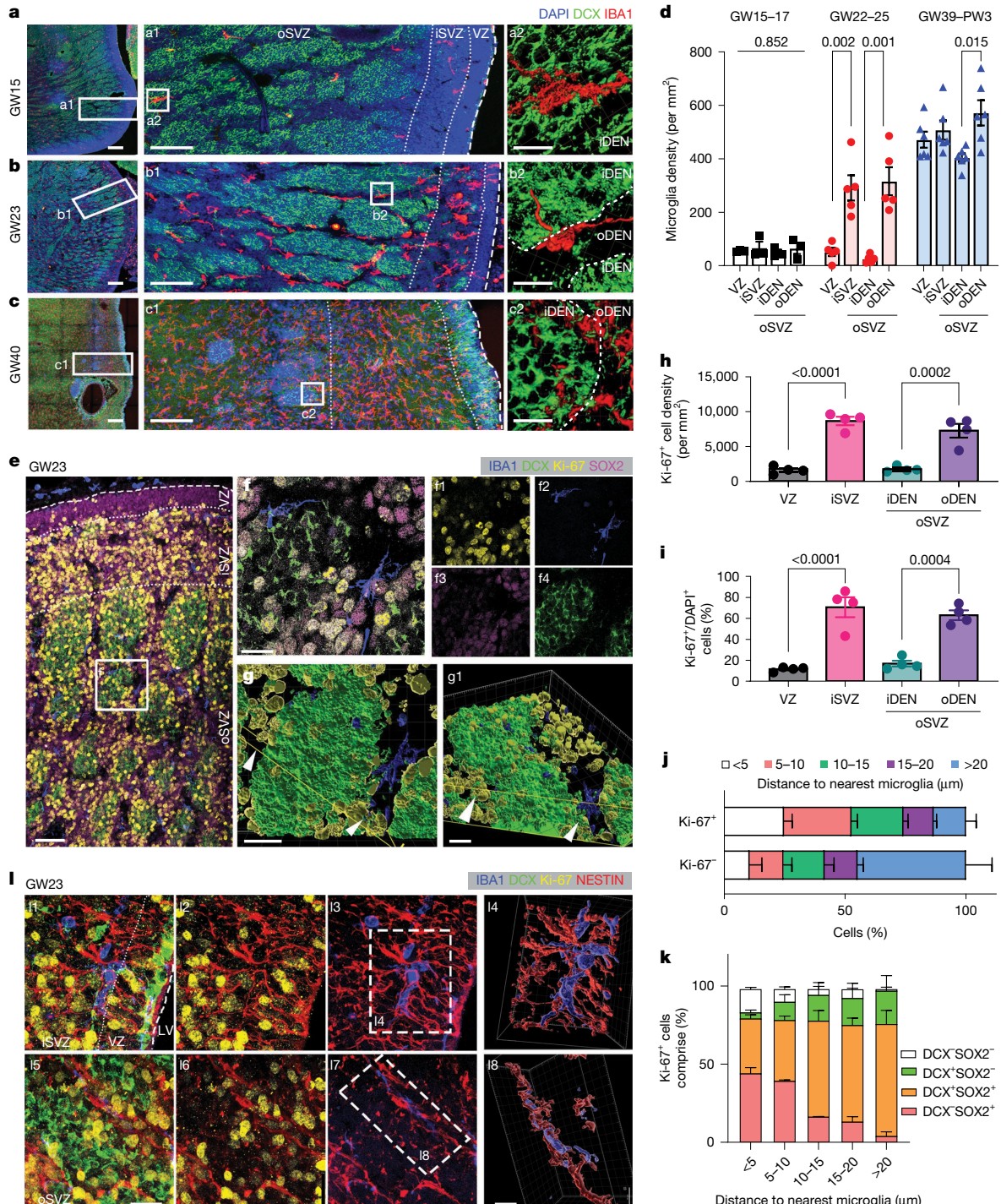

**Fig. 1 | Microglia are in proximity to MGE progenitors in the developing human brain. a–d**, IHC images (**a–c**) and bar graphs (**d**) showing microglia in the developing hMGE at GW15 (**a,d**), GW23 (**b,d**) and GW40 (**c,d**). Microglia were highly concentrated in the iSVZ and oDENs of oSVZ at GW22–25. **e,f**, IHC images showing the spatial relationship between Ki-67[+] proliferating progenitors and microglia in the GW23 hMGE sections; **f** provides a magnified view of the region indicated in **e**. **g**, Three-dimensional reconstruction revealing the close proximity between microglia and Ki-67[+] proliferating progenitors in the oSVZ of GW23 hMGE. Arrowheads point to the Ki-67[+] clusters near the microglia in oSVZ. **h,i**, Bar graphs showing the density (**h**) and percentage (**i**) of Ki-67[+] progenitors in GW22–25 hMGE. **j**, Percentage of Ki-67[+] and Ki-67[−] cells as their distance to microglia increased in the oSVZ of hMGE. **k**, Cell composition analysis of Ki-67[+] progenitors showing Ki-67[+] cells in proximity to microglia comprising SOX2[+]DCX[−] radial glia and SOX2[+]DCX[+] neuroblasts. **l**, IHC images and 3D reconstruction indicating that microglia closely contacted NESTIN[+] projections in GW23 hMGE. For statistics, n (biological repeats) = 3, 5 and 6 (**d**); n = 4 (**h,i**); n = 4 (**j,k**). One-way analysis of variance (ANOVA) and post hoc Bonferroni's test for **d,h** and **i**; two-way repeated measures ANOVA for **j** and **k** showed significant interaction effects (P = 0.006 (**j**) and P < 0.0001 (**k**)). Data in **d** and **h–k** are shown as mean ± s.e.m. Scale bars, 200 μm (**a–c** (left)), 100 μm (**a–c** (middle)), 10 μm (**a–c** (right), **l** (right)), 50 μm (**e**), 20 μm (**f,g,l** (left)).

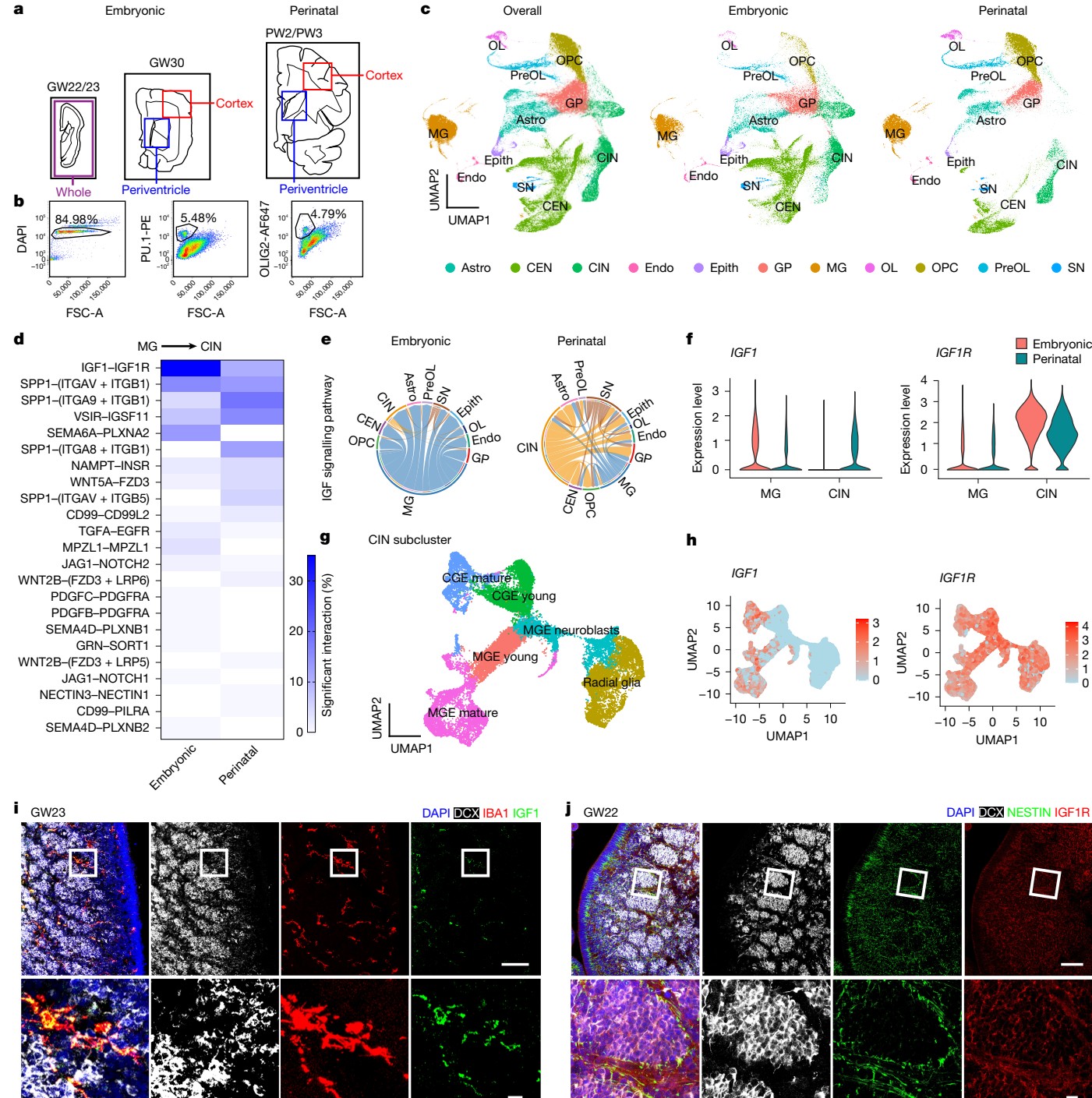

**Fig. 2 | Transcriptomic profiling and cell–cell interaction analysis revealed IGF1–IGF1R as the top potential signalling pathway mediating the communication between interneuron progenitors and microglia in developing hMGE. a**, Diagrams showing the brain regions used for the snRNA-seq experiment. **b**, Flow cytometry strategies for unenriched DAPI⁺ nuclei and enrichment of PU.1⁺ and OLIG2⁺ nuclei. **c**, UMAP plots showing the 11 identified cell types. **d**, Heat-map plot showing the significant ligand–receptor pairs that potentially mediate microglial regulation of interneuron development at different developmental stages. **e**, Chord plots of IGF signalling pathways at the embryonic and perinatal stages. **f**, Violin plots of *IGF1* and *IGF1R* expression by microglia and interneurons at the embryonic and perinatal stages. **g**, UMAP plots of the subtypes of cortical GABAergic interneurons, including radial glia,

MGE neuroblasts, MGE-derived young interneurons (MGE young), MGE-derived mature interneurons (MGE mature), CGE-derived young interneurons (CGE young) and CGE-derived mature interneurons (CGE mature). **h**, Feature plots showing the expression patterns of *IGF1* and *IGF1R* in the interneuron subtypes. **i**, Representative IHC from three biological repeats showing the specific expression of IGF1 in microglia in developing hMGE. **j**, Representative IHC from three biological repeats showing the expression pattern of IGF1R in the developing hMGE. Astro, astrocytes; CEN, cortical excitatory neurons; CIN, cortical GABAergic interneurons; Endo, endothelial cells; Epith, epithelial cells; GP, glia progenitors; MG, microglia; OL, oligodendrocytes; OPC, oligodendrocyte precursor cells; PreOL, pre-myelinating oligodendrocytes; SN, subpallial neurons. Scale bars, 100 μm (**i,j** (top)), 10 μm (**i,j** (bottom)).

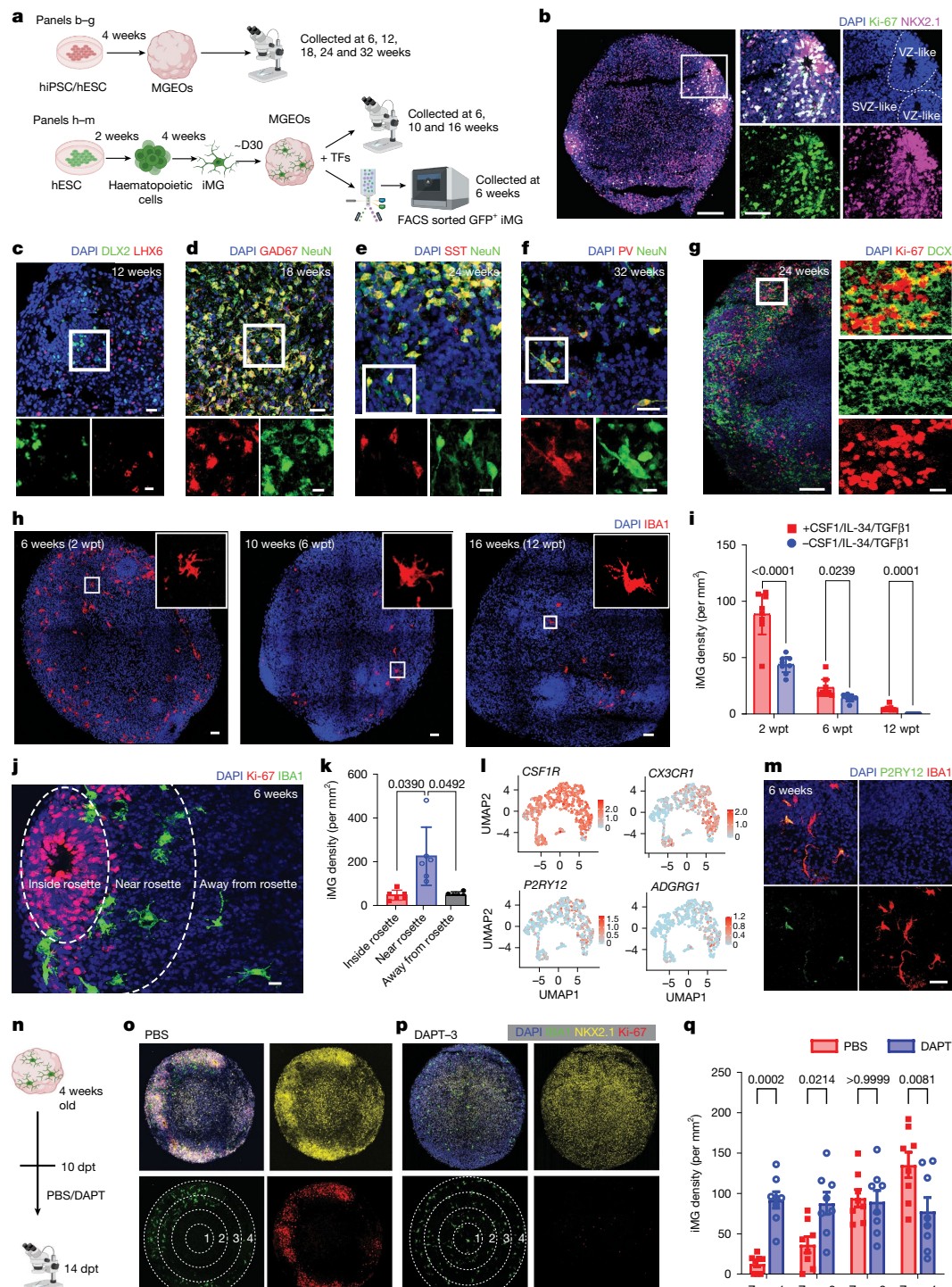

**Fig. 3 | MGEO models hMGE development and interaction between microglia and hMGE progenitors. a**, Schematic diagram of experimental flow. Transcriptional factors (TFs) include CSF1, IL-34 and TGFβ1. **b**, MGEOs containing 'rosette'-like proliferating centres, recapitulating the ventricular zone (VZ)-like and SVZ-like areas of hMGE. **c**–**f**, Sequential expression of markers for MGE progenitors and MGE-derived interneurons that mimic the temporal progression of hMGE development in MGEOs including DLX2 and LHX6 (**c**), GAD67 and NeuN (**d**), SST (**e**) and PV (**f**). **g**, Clusters of DCX⁺ neuroblasts sprinkled with Ki-67⁺ cells noted in 24-week-old MGEOs, resembling DEN-like structures in hMGE. **h**,**i**, IHC images (**h**) and bar graphs (**i**) showing iMG survived up to 12 weeks post-iMG transplantation (wpt) in MGEOs. **j**,**k**, IHC images (**j**) and quantification (**k**) showing the distribution of iMG around rosettes. **l**, Feature plots of microglial homeostatic markers expressed in FACS-isolated iMG from 6-week-old MGEOs. **m**, IHC images showing the expression of P2RY12 in iMG. **n**, Schematic outlining

the administration of PBS or DAPT from 10 to 14 dpt before the IHC analysis of iMG distribution. **o**–**q**, IHC images of control (**o**) and DAPT-treated MGEOs (**p**) and bar graphs (**q**) showing that DAPT treatment sharply reduced the number of Ki-67⁺ proliferating cells and rosette formation in MGEOs. As proliferation decreased, iMG became more evenly distributed throughout the organoids. The white dashed lines mark the concentric ellipses (zones 1–4) used for binned quantification of iMG density in **q**. For statistics, $n = 8, 8, 8, 7, 8$ and 7 (each group in **i**); $n = 6$ (**k**); and $n = 8$ (**q**). Unpaired two-tailed $t$-test (**i**); one-way repeated measures ANOVA and post hoc Bonferroni's test (**k**); two-way repeated measures ANOVA and post hoc Bonferroni's test (**q**); data in **i**,**k** and **q** are shown as mean ± s.e.m. Scale bars, 100 μm (**b** (left), **g** (left)), 50 μm (**b** (middle), **h**,**m**), 25 μm (**c**–**f** (top), **g** (right)), 10 μm (**c**–**f** (bottom)), 20 μm (**j**). Illustrations in **a** and **n** were created using BioRender (https://biorender.com).

of markers for MGE progenitors and interneurons, including DLX2, SOX2, LHX6, DCX, GAD67, NeuN, SST and PV (Fig. 3c–f and Extended Data Fig. 5a–c), confirming MGE identity and generation of GABAergic interneurons. Notably, Ki-67-expressing cells were distributed in the centre of ventricular zone-like rosettes as well as in the edge and neighbouring SVZ-like regions of rosettes at 6 weeks of age (Fig. 3b and Extended Data Fig. 5d). IHC analysis showed that Ki-67[+] progenitors comprised both SOX2[+]DCX[−] radial glia-like cells and SOX2[+]DCX[+] neuroblasts in 6-week-old MGEOs (Extended Data Fig. 5d), recapitulating the spatial organization of radial glia and proliferating neuroblasts in developing hMGE. We observed clusters of DCX[+] neuroblasts sprinkled with Ki-67[+] cells in 24-week-old MGEOs, which may represent DEN-like clusters (Fig. 3g). Taken together, our data indicate that MGEOs can serve as a model to study hMGE development.

Microglia do not appear in situ in organoids. To study the role of microglia in hMGE development, we established microglia-containing MGEOs by transplanting hESC-induced iMG into MGEOs at 4 weeks of age, mimicking the stage when microglia are detected in developing human brains at GW4.5 (refs. 16,39) (Fig. 3a). The addition of recombinant human colony-stimulating factor 1 (CSF1), interleukin-34 (IL-34) and transforming growth factor-β1 (TGFβ1) to the culture medium significantly improved iMG survival up to 12 weeks post-transplantation (wpt) (Fig. 3h,i), which is consistent with previous reports[40]. We observed that iMG invaded MGEOs after 24 h, at 1 day post-transplantation (dpt), and reached the organoid centre at 5 dpt. The preferential accumulation around rosette-like proliferative centres became apparent at 8–14 dpt (Extended Data Fig. 5e). We observed that iMG mostly avoided the ventricular zone-like rosettes but accumulated in the SVZ-like neighbouring regions in the MGEOs (Fig. 3j,k). As seen in hMGE in vivo, iMG were in close proximity to Ki-67[+] and SOX2[+] progenitors (Extended Data Fig. 5f–i) and intimately interacted with NESTIN[+] projections (Extended Data Fig. 5j).

To further document the properties of the transplanted iMG, we performed single-cell RNA sequencing (scRNA-seq) analysis of iMG and found that they demonstrated similar developmental trajectory and heterogeneity as primary embryonic human microglia[39,41–50] (Extended Data Fig. 6a–e). They expressed homeostatic microglial markers, such as *CSF1R*, *CX3CR1*, *P2RY12*, as well as *ADGRG1* (Fig. 3l,m), one of the few genes that define yolk sac-derived 'true' microglia[51]. Notably, we detected little, if any, *SALL1* and *TEME119* transcripts in iMG (Extended Data Fig. 6f), which is consistent with previous studies[40,52]. Morphologically, the transplanted iMG demonstrated a larger soma volume but a similar ramification as human primary microglia in hMGE around GW23 (Extended Data Fig. 6g–j).

It is intriguing why microglia exhibited a preferential distribution in the SVZ of hMGE and the proliferating zone of MGEOs. Human microglia have low proliferation capacity at this developmental stage[53]. Our snRNA-seq results showed that 1.64% (75 of 4,561) and 0.15% (14 of 9,631) of microglia were Ki-67[+] at GW23–30 and postnatal weeks 2–3, respectively. We did not observe any Ki-67[+] microglia in the GW23 MGE in our immunostaining (*n* = 3). These results challenged proliferation as the primary mechanism for microglial accumulation in SVZ. To probe whether proliferating progenitors promote microglial chemotaxis towards the proliferating zone, we blocked cell proliferation by administering the Notch pathway inhibitor DAPT[54] from 10 to 14 dpt (Fig. 3n). The DAPT treatment effectively eliminated cell proliferation and rosette formation (Fig. 3o,p). We observed that iMG was evenly distributed in organoids treated with DAPT (Fig. 3p,q). Taken together, our results indicate that microglia are likely to migrate to the SVZ of hMGE and the proliferating zone of MGEOs in response to proliferating progenitors.

## Microglia promote MGE neurogenesis

To investigate microglial regulation of interneuron development, we conducted scRNA-seq of 6-week-old MGEOs with and without transplanted iMG at 4 weeks of age (Fig. 4a). We recovered 21,136 cells, with a median of 5,045 UMIs and 2,629 genes per cell. On the basis of unbiased clustering and canonical marker genes, we identified clusters of radial glia, neuroblasts, young MGE-derived GABAergic interneurons and a small group of CGE-like cells (Fig. 4b and Extended Data Fig. 7). Notably, without a fluorescence-activated cell sorting (FACS)-based enrichment strategy, iMG were barely detected in this scRNA-seq result, probably because of their low abundance. We observed a significant increase in the proportion of radial glia and a significantly higher percentage of Ki-67[+] cells in MGEOs transplanted with iMG (Fig. 4c,d), indicating the role of iMG in promoting progenitor proliferation. Differentially expressed gene (DEG) analysis showed that upregulated genes in MGEOs transplanted with iMG were enriched in pathways and gene ontologies related to cell mitosis (Fig. 4e,f and Extended Data Fig. 8). The downregulated genes in MGEOs with iMG were largely enriched in pathways and gene ontologies related to oxidative stress (Fig. 4e,f and Extended Data Fig. 8), as shown in previous studies[45,55]. Additionally, upregulated genes in radial glia were significantly enriched in Gene Expression Omnibus (GEO) terms related to IGF1R downstream signalling (Fig. 4e,f), such as *CCND1*, *TMPO*, *RRM2* and *MCM3*, in MGEOs with iMG, suggesting a role for IGF1–IG1R in microglial regulation of interneuron progenitor proliferation.

We next conducted IHC to characterize the effect of iMG on interneuron development in MGEOs. Using Ki-67 and NKX2.1 as markers, we observed that the density of proliferating MGE progenitors was significantly increased in 6-week-old MGEOs transplanted with iMG (Fig. 4g,h and Supplementary Fig. 4). Further analysis revealed that the presence of iMG significantly increased the density of proliferating radial glia (Ki-67[+]SOX2[+]DCX[−]) and neuroblasts (Ki-67[+]SOX2[+]DCX[+]) (Fig. 4i,j). To document the dynamic interaction between iMG and proliferating progenitors, we generated MGEOs using NKX2.1-GFP hESCs[56] and transplanted them with tdTomato-labelled iMG. We observed active interactions between microglia and NKX2.1[+] cells. We captured one dividing NKX2.1-GFP cell after direct contact from microglia (Extended Data Fig. 9 and Supplementary Video 1). We found that iMG transplantation significantly increased the density of NeuN[+]GAD67[+] mature interneurons in 18-week- to 24-week-old organoids (Fig. 4k,l), as well as increased NeuN[+]SST[+] and a trend of increased NeuN[+]PV[+] MGE-derived subtypes of interneurons in 24-week and 32-week-old organoids (Extended Data Fig. 10). Taken together, our data indicate that microglia promote the proliferation and generation of MGE-derived interneurons.

## Microglial IGF1 promotes MGE proliferation

We next investigated whether microglia exert their function in hMGE development through IGF1 using our neuroimmune MGEOs (Fig. 5a). We showed that IGF1 was specifically detected in iMG, whereas IGF1R was highly expressed in NESTIN[+] radial glia in 6-week-old MGEOs (Fig. 5b,c), as observed in hMGE (Fig. 2i,j).

To examine the role of IGF1 in hMGE development, we treated 6-week-old MGEOs, transplanted with or without iMG at 4 weeks of age, with either a carrier control (phosphate-buffered saline (PBS)), recombinant human IGF1 proteins (IGF1) or IGF1-neutralizing antibodies (anti-IGF1) for 48 h. BrdU was added during the last 4 h of the 48-h treatment for precise evaluation of cell proliferation cells (Fig. 5a). We observed that the presence of iMG increased the density of BrdU[+]NKX2.1[+] cells in MGEOs (Fig. 5d,e). IGF1 treatment drastically increased the density of BrdU[+]NKX2.1[+] cells in MGEOs without iMG transplantation, whereas IGF1-neutralizing antibodies abolished the elevated density of BrdU[+]NKX2.1[+] cells in organoids with iMG (Fig. 5d,e). To definitively demonstrate that IGF1 is the key regulator mediating microglial regulation of interneuron development, we generated *IGF1* loss-of-function mutation (*IGF1* knockout) hESC lines using CRISPR–Cas9-based non-homology end joining (Extended Data Fig. 11). We then generated MGEOs transplanted with no iMG, control iMG or *IGF1*

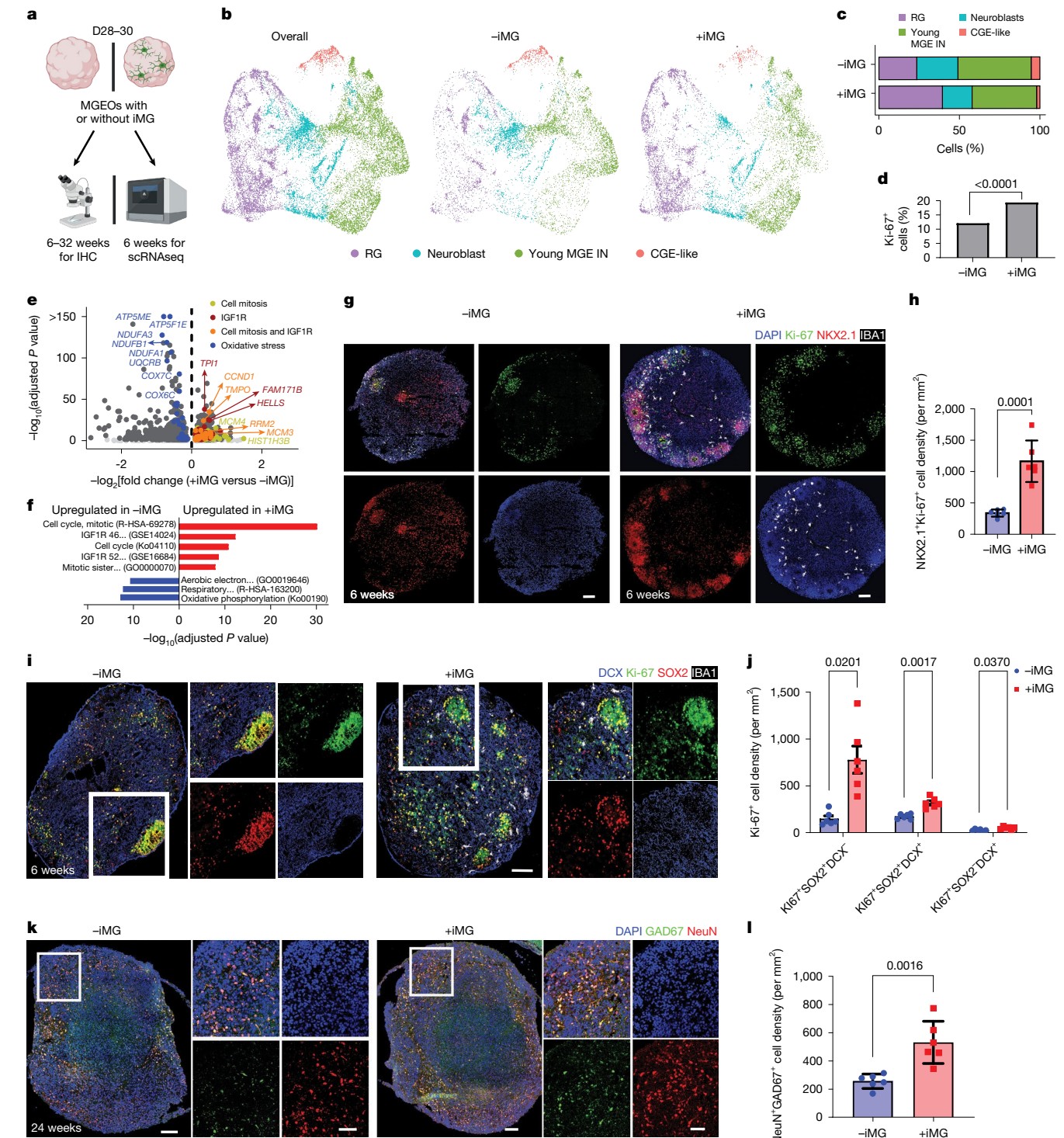

**Fig. 4 | Microglia promote MGE proliferation and interneuron production. a**, Experimental diagrams to examine iMG effects in MGEOs. **b**, UMAP plots showing cell type clusters from 6-week-old MGEOs with and without iMG. **c**, The proportion of RG significantly increased in MGEOs transplanted with iMG (*P* < 0.0001). **d**, The percentage of Ki-67+ nuclei significantly increased in MGEOs with iMG. **e**, Volcano plot showing DEGs in the RG of MGEOs with and without iMG. **f**, Upregulated DEGs in MGEOs with iMG are enriched in pathways and gene ontologies related to cell mitosis and the IGF1R signalling pathway; upregulated DEGs in MGEOs without iMG were enriched in pathways and gene ontologies related to oxidative stress. See Methods for the full names of pathways. **g,h**, IHC images (**g**) and bar graph (**h**) showing that iMG increased the density of NKX2.1+ Ki-67+ progenitors in 6-week-old MGEOs. **i,j**, IHC images (**i**) and bar graph (**j**) showing iMG increased the density of SOX2+DCX− radial glia in 6-week-old MGEOs. **k,l**, IHC images (**k**) and bar graph (**l**) showing that iMG increased the density of

NeuN+GAD67+ interneurons in 18-week to 24-week-old MGEOs. For statistics, *n* = 6 (**h**); *n* = 6 (**j**); *n* = 6 (**l**); $\chi^2$ test (two-sided), 2,221 (RG cell number) of 9,456 (overall cell number) versus 4,605 of 11,680, $\chi^2$ = 607.1, *P* < 0.0001 (**c**); $\chi^2$ test (two-sided), 1,146 (Ki-67+ cell number) of 9,456 (overall cell number) versus 2,269 of 11,680, $\chi^2$ = 206.0 (**d**); $\chi^2$ tests in **c** and **d** were on the basis of the fractions of targeted cells among the total cells recovered in scRNA-seq data from 6-week-old organoids. The adjusted *P* value in **e** was calculated on the basis of the Seurat-default non-parametric Wilcoxon rank-sum test. The adjusted *P* value in **f** was calculated in Enrichr, using Benjamini–Hochberg correction. Unpaired two-tailed *t*-test in **h** and **l**. Two-way repeated measures ANOVA and post hoc Bonferroni's test in **j**. Data in **h,j** and **l** are shown as mean ± s.e.m. RG, radial glia; IN, interneurons. Scale bars, 100 μm (**g,i,k** (main image)), 50 μm (**k** (zoomed image)). Illustration in **a** was created using BioRender (https://biorender.com).

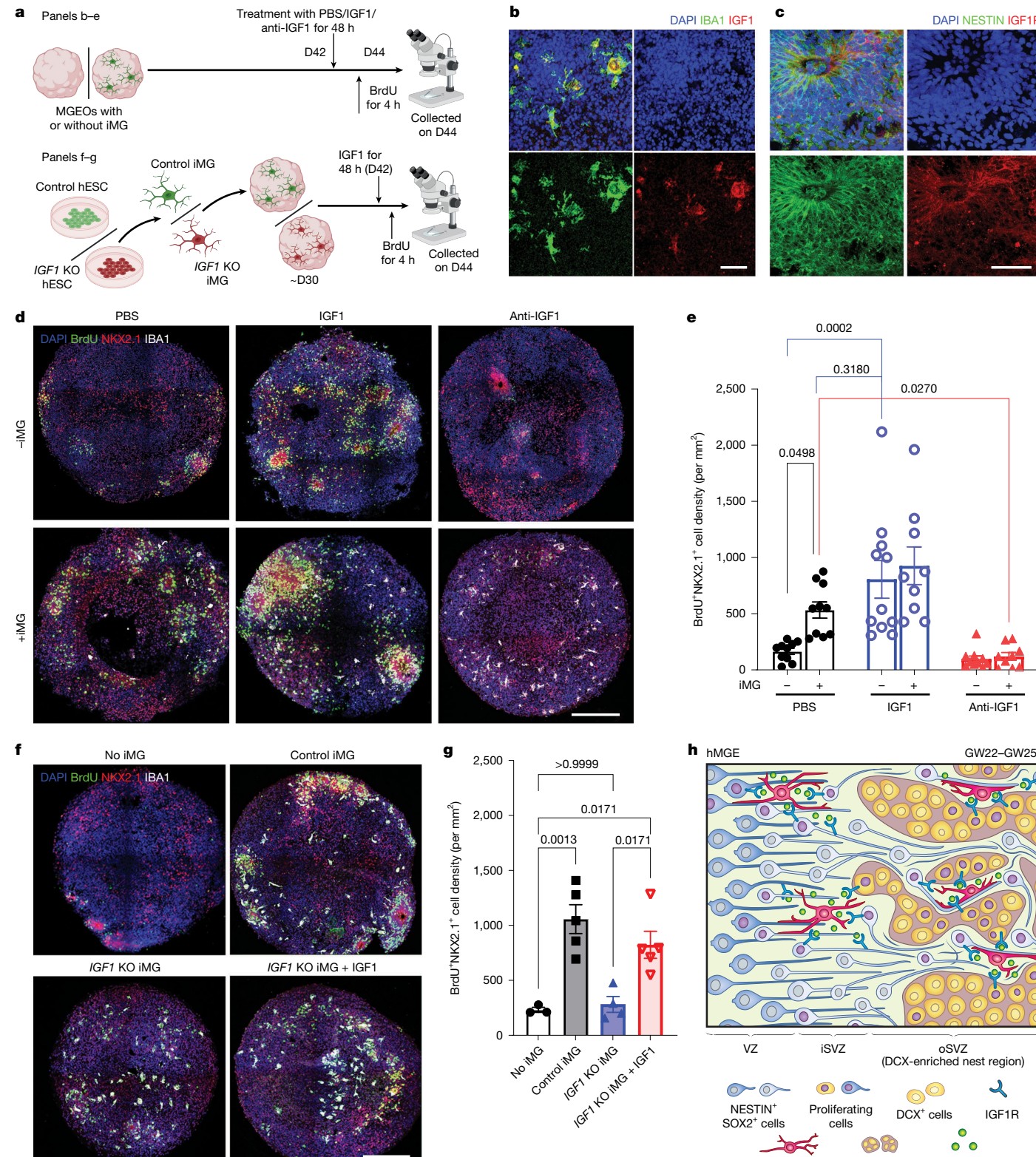

**Fig. 5 | Microglia-derived IGF1 promotes MGE proliferation in MGEOs.**
**a**, Experimental schematics. **b**, IHC images indicating the specific expression of IGF1 in iMG of 6-week-old MGEOs. **c**, IHC images showing the expression of IGF1R in progenitors in the rosettes of 6-week-old MGEOs. **d,e**, IHC images (**d**) and bar graphs (**e**) showing that IGF1 promotes MGE progenitor proliferation in MGEOs without iMG to the level seen in MGEOs with iMG, and that IGF1-neutralizing antibody treatment abolished iMG-mediated progenitor proliferation. **f,g**, IHC images (**f**) and bar graphs (**g**) showing the inability of *IGF1*

knockout (KO) iMG to promote MGEO progenitor proliferation, which can be rescued by the addition of recombinant IGF1 treatment. **h**, Graphical abstract showing microglial regulation of interneuron progenitor proliferation through IGF1 in hMGE. For statistics, $n = 10, 11$ and 9 for MGEO −iMG; $n = 10, 9$ and 9 for MGEO +iMG (**e**). $n = 3, 5, 5$ and 5 (**g**). Two-way ANOVA (**e**), one-way ANOVA (**g**) and post hoc Bonferroni's test for selected comparisons. Data in **e** and **g** are shown as mean ± s.e.m. Scale bars, 50 μm (**b,c**), 200 μm (**d,f**). Illustration in **a** was created using BioRender (https://biorender.com).

knockout iMG at 4 weeks of age and examined MGE progenitor proliferation by quantifying the density of BrdU and NKX2.1 double-positive cells at 6 weeks of age (Fig. 5a). We observed that MGE progenitor proliferation increased in MGEOs transplanted with control iMG but not in those transplanted with *IGF1* knockout iMG (Fig. 5f,g). The addition of recombinant human IGF1 rescued the phenotype associated with *IGF1* knockout iMG (Fig. 5f,g). On the other hand, we did not observe any significant changes in the morphology, distribution or density of iMG in MGEOs following *IGF1* knockout (Fig. 5f and Extended Data Fig. 12). To investigate IGF1 signalling through IGF1R, we administered IGF1R inhibitors (GSK4529 (ref. 57) and picropodophyllin[58,59]) to MGEO cultures. We observed that both inhibitors abolished iMG-induced MGE proliferation (Extended Data Fig. 13). In summary, our results indicate that iMG exert their function in promoting MGE progenitor proliferation through the IGF1–IGF1R pathway.

To examine whether IGF1 can promote cortical radial glial proliferation and projection neurogenesis, we generated cortical neuroimmune organoids and treated 6-week-old organoids with PBS, IGF1 or anti-IGF1. The iMG transplantation and IGF1 treatment significantly increased the density of PAX6[+]BrdU[+] cells, whereas anti-IGF1 significantly reduced it, indicating that iMG and iMG-derived IGF1 enhanced PAX6[+] neural progenitor proliferation in cortical organoids (Extended Data Fig. 14).

Finally, we probed whether the microglial IGF1 mechanism of the supporting interneuron development is species-specific. We first examined the expression of IGF1 protein in mouse microglia. Consistent with the literature, we found that IGF1 is expressed by axon tract-associated microglia (ATM)[42] in the developing white matter at P5 (Extended Data Fig. 15a) and by ATM-like microglia[60] at the cortico-striato-amygdalar boundary on embryonic day 14.5 (Extended Data Fig. 15b). However, we did not detect IGF1 protein in microglia located in the mouse MGE on embryonic day 14.5 (Extended Data Fig. 15c). Previous cross-species single-cell studies revealed that IGF1 is expressed at significantly higher levels in human microglia than in mice and other rodents[61]. Next, we investigated MGE proliferation in microglial *Igf1* conditional knockout (cKO) mice (*Igf1[f/f]*, *Cx3cr1[CreERt/+]*). We injected tamoxifen on embryonic days 11.5 and 12.5 to induce microglial *Igf1* knockout, followed by EdU pulse labelling of proliferating progenitors 4 h before brain collection on embryonic day 14.5, the peak time for late MGE neurogenesis[62]. Microglial *Igf1* cKO did not affect MGE proliferation on embryonic day 14.5 (Extended Data Fig. 15d–f). Combined with the differential distribution of microglia and proliferating progenitors in the MGE between humans and mice (Fig. 1 and Extended Data Fig. 1), our results indicate a species-specific mechanism for microglial regulation of MGE proliferation.

## Discussion

In summary, our findings provide critical insights into the role of microglia in the development of hMGE and their influence on GABAergic interneurons (Fig. 5h). We demonstrated that microglia exhibit a species-specific distribution pattern, closely interacting with proliferating progenitors in hMGE (Fig. 1). Further studies on the basis of transcriptomic analysis (Fig. 2) and 3D MGEO models demonstrated a clear function of microglia in hMGE progenitor proliferation through the IGF1 pathway (Fig. 5). By contrast, in mice, microglia are more abundant in the ventricular zone of MGE (Extended Data Fig. 1a–f), have minimal IGF1 expression (Extended Data Fig. 15c) and display independence of IGF1 in MGE progenitor proliferation (Extended Data Fig. 15d–f). These findings indicate an evolutionary adaptation of microglial function to support the increased demand for interneurons in the human cortex.

This study uncovered a developmentally specific role of microglia in IGF1-mediated neurogenesis in hMGE. Although the neurotrophic role of IGF1 has been reported[63–72], we found that microglia serve as a key source of IGF1 during early hMGE development. IGF1 expression in microglia decreased from the embryonic phase to the postnatal period (Fig. 2f), which is consistent with the literature[73]. As microglial IGF1 expression declines with age, IGF1 is expressed by mature interneurons (Fig. 2h), indicating a developmental switch in its cellular source. The role of microglia in cortical neurogenesis, particularly in humans, remains unclear. Although microglia have been shown to promote embryonic cortical neurogenesis in mice[74–76] and hPSC-derived cortical organoids (Extended Data Fig. 14), their contribution to human cortical development in vivo is unclear. Compared with rodents, microglial distribution varies significantly across cortical regions in macaques[77] and has not been consistently reported in the human cortex[53,77]. These differences highlight the need for further studies to clarify the extent and function of microglia in human cortical neurogenesis.

As a close family member of IGF1, IGF2 has been implicated in neurogenesis in mice[78,79]. Its source is largely endothelial cells[78,79]. Our snRNA-seq analysis showed that in humans, IGF2 is expressed at low levels and is specifically localized to endothelial cells (Extended Data Fig. 3), mirroring the pattern observed in mice[78,79]. IGF2 receptors (IGF2R) are expressed in interneuron progenitors (Extended Data Fig. 3), suggesting that IGF2 signalling may also contribute to developmental neurogenesis, although not through the microglia–progenitor axis.

Studies in rodents have shown a region-specific interaction between the radial glia and microglia. For example, microglia wrap their processes around radial glial projections in the embryonic mouse hypothalamus and respond to immune challenges[80]. Radial glial endfeet contact meningeal microglial precursors and regulate microglial development through integrin αVβ8–TGFβ1 signalling in the embryonic mouse cortex[81]. Our findings indicate that microglia closely interact with radial glia in the SVZ of hMGE (Fig. 1l). Furthermore, our data from neuroimmune organoid models indicate that microglia enhance the proliferation of radial glia by secreting IGF1 (Figs. 4 and 5), highlighting a vital neuroimmune mechanism in brain development. Further studies are needed to understand the mechanism underlying the intimate and preferential association between radial glia and microglia in hMGE.

Brain organoids have been used to model various aspects of brain development and disease in vitro[52,82–88]. This in vitro model enables mechanistic studies of human-specific aspects of brain development. We adapted a ventral telencephalon-oriented organoid model and demonstrated the sequential presence of MGE progenitors, hMGE unique DEN-like organization and MGE-derived PV[+] and SST[+] interneurons, resembling the developmental trajectory of the developing hMGE (Fig. 3b–g). The limitations of brain organoid models include the lack of endogenous microglia and short survival duration of transplanted microglia. Here we showed that the presence of trophic factors, recombinant human CSF1, IL-34 and TGFβ, supports iMG survival up to 12 wpt. These iMG resemble human primary microglial morphology, transcriptomics and physical proximity to proliferating radial glia and progenitors (Fig. 3j–m and Extended Data Figs. 5f–j and 6). Thus, our newly established neuroimmune MGEOs serve as a reliable model for studying the microglial regulation of hMGE development.

The results of this study have significant implications for understanding the development of GABAergic interneurons in the human brain and the potential aetiology of neurological and psychiatric disorders associated with microglial and interneuron dysfunction. Epidemiological studies have shown lower levels of IGF1 in the cerebrospinal fluid of children with autism[89,90]. Further human and mouse studies have demonstrated the potential efficacy of IGF1 administration in ameliorating autistic features[91–95]. This study indicates that microglia are the main sources of IGF1 in the embryonic human brain (Fig. 2e–i and Extended Data Fig. 3), and that IGF1 promotes interneuron production (Figs. 4 and 5), suggesting the possibility that low levels of IGF1 could result in interneuron deficits.

## Limitations of the study

Owing to the limitations of current technologies, we were unable to clarify the real-time dynamics of microglial migration into hMGE proliferative zones, their live interactions with progenitor cells and their direct effects on progenitor division.

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

# Methods

## Human tissue samples

De-identified human specimens were collected from the Autopsy Service in the Department of Pathology at the University of California San Francisco (UCSF) (Supplementary Table 2) with previous patient consent in strict observance of the legal and institutional ethical regulations. Autopsy consents and all protocols for human prenatal brain tissue procurement were approved by the Human Gamete, Embryo and Stem Cell Research Committee (Institutional Review Board GESCR no. 10-02693) at UCSF. All specimens received diagnostic evaluations by a board-certified neuropathologist as control samples and were free of brain-related diseases. The diagnostic panel included assessments of neural progenitor and immune cells using IHC to ensure that all control cases were not affected by any inflammatory diseases. Tissues used for snRNA-seq were snap-frozen, either on a cold plate placed on a slab of dry ice or in isopentane on dry ice. Tissues later used for IHC were cut coronally into 1-mm tissue blocks, fixed with 4% paraformaldehyde (PFA) for 2 days, cryoprotected in a 15–30% sucrose gradient, embedded in optimal cutting temperature (OCT; SciGen; 4586) compound, sectioned at 30 μm using a Leica cryostat and mounted onto glass slides.

## Authentication of cell lines used

All hiPSC and hESC lines used in this study were karyotyped and regularly tested for *Mycoplasma*. The eWT-1323-4 hiPSC line[84] (female; Research Resource Identifier (RRID): CVCL_0G84) was obtained from the Conklin Laboratory (UCSF). WA09/H9 (female; RRID: CVCL_9773; National Institutes of Health (NIH) registration number: NIHhESC-10_0062) and WA01/H1 (male; RRID: CVCL_9771; NIH registration number: NIHhESC-10-0043) were obtained from the WiCell Research Institute. NKX2.1-GFP hESC line[56] (female) was obtained from Murdoch Children's Research Institute and Monash University.

## Mice

All mice were handled in accordance with the guidelines of the Institutional Animal Care and Use Committee of UCSF. Minimal sample sizes were chosen on the basis of standards commonly used in the field and previous experience with similar experiments. All animals of the same genotype and sex were randomly selected for breeding and/or experimentation in this study. Wild-type C57/B6 mice were purchased from Taconic Biosciences and bred in the laboratory. *Igf1*$^{f/f}$ mice (strain number 012663) and *Cx3cr1*$^{CreERt/+}$ mice (strain number 020940) were purchased from The Jackson Laboratory. For timed pregnancy, males and females were paired, and females were observed daily for the presence of a copulation plug. The noon of the day when a plug was observed was noted as embryonic day 0.5. For *Igf1* cKO experiments, 100 mg kg$^{-1}$ of tamoxifen in corn oil was injected intraperitoneally into pregnant dams on embryonic days 11.5 and 12.5. *Igf1*$^{f/f}$; *Cx3cr1*$^{CreERt/+}$ fetuses were used as *Igf1* cKO mice, and their littermates *Igf1*$^{+/+}$; *Cx3cr1*$^{CreERt/+}$ and *Igf1*$^{f/f}$; *Cx3cr1*$^{+/+}$ fetuses were used as controls. During all subsequent experimental procedures, including sample collection, processing, imaging and quantification, the experimenter was blinded to the genotype, sex and age of the mice. Both males and females were included in the mouse experiments. In the EdU labelling experiment, a single dose of EdU (10 mg kg$^{-1}$; provided in the Click-iT EdU Alexa Fluor 647 Imaging Kit from Invitrogen; C10340) was injected intraperitoneally into pregnant mice at embryonic day 14.5. At embryonic days 14.5 and 16.5, the pregnant dams were killed, and the fetal brains were collected, fixed in 4% PFA at 4 °C overnight, cryopreserved in 30% sucrose at 4 °C overnight, embedded in OCT and cryosectioned at 20 μm (EdU labelling and IGF1 staining experiments) or 40 μm (microglia staining experiments) using a Leica cryostat. Additionally, wild-type P5 pups were transcardially perfused with 4% PFA, and their brains were extracted and post-fixed in 4% PFA overnight, cryoprotected in 30% sucrose overnight, embedded in OCT and cryosectioned at 20 μm.

## Human pluripotent stem-cell-derived organoids

The hPSC-derived organoids were generated largely following a previously established protocol[37,38]. In brief, 1323-4 hiPSCs or WA01/H1 and WA09/H9 hESCs were expanded in StemFlex Basal Medium (Gibco; A3349401). After reaching 80% coverage, hPSCs cultured on Matrigel were dissociated into clumps using ReLeSR (STEMCELL Technologies; 100-0483) and equally distributed into a V-bottom 96-well ultra-low-attachment PrimeSurface plate (S-BIO; MS-9096VZ). The rho kinase inhibitor Y-27632 (10 μM) was added during the first 24 h of neural induction to promote survival. Between days 0 and 5, organoids were cultured in neural induction medium (Dulbecco's modified Eagle medium/F-12, 20% knockout serum, 1% non-essential amino acids, 0.5% GlutaMAX, 0.1 mM β-mercaptoethanol and 1% penicillin–streptomycin) supplemented with the SMAD inhibitors SB431542 (10 μM) and dorsomorphin (5 μM). Between days 6 and 24, organoids were cultured in neural differentiation medium (Neurobasal-A medium, 2% B27 supplement, 1% GlutaMAX and 1% penicillin–streptomycin) supplemented with human recombinant EGF (20 ng ml$^{-1}$) and human recombinant FGF2 (20 ng ml$^{-1}$). Between days 25 and 43, organoids were maintained in neural differentiation medium supplemented with human recombinant brain-derived neurotrophic factor (20 ng ml$^{-1}$) and human recombinant neurotrophin 3 (20 ng ml$^{-1}$). For MGEOs, the media were also supplemented with 5 μM wnt inhibitor IWP-2 on days 4–23, 100 nM smoothened agonist on days 12–23, 100 nM retinoic acid on days 12–15 and 100 nM allopregnanolone on days 16–23 for ventral forebrain patterning. Cortical organoids were not supplemented with IWP-2, smoothened agonist, retinoic acid and allopregnanolone. Each organoid was then moved to six-well plates for long-term culture after week 5. All media and supplements used for organoid cultures were the same as those in a previously published protocol[37,38].

## Induced microglia

Induced microglial cells were generated from WA01/H1 or WA09/H9 hESC cells using STEMdiff kits, according to the manufacturer's protocols. In brief, hESCs were differentiated into CD43-expressing haematopoietic progenitor cells for 12 days using a STEMdiff Hematopoietic Kit (STEMCELL; 05310). Haematopoietic progenitor cells were differentiated for 24 days using the STEMdiff Microglia Differentiation Kit (STEMCELL; 100-0019) and matured for an extra 4 days using the STEMdiff Microglia Maturation Kit (STEMCELL; 100-0020) before being added to the organoid cultures for co-culture.

## iMG–organoid engraftment and co-culture

Mature iMG were immediately added to 4-week-old MGE organoids in 96-well ultra-low attachment PrimeSurface plates at 80–100 × 10$^3$ microglia per organoid. Trophic factors (100 ng ml$^{-1}$ of IL-34 (PeproTech; 200-34), 25 ng ml$^{-1}$ of CSF1 (PeproTech; 300-25) and 50 ng ml$^{-1}$ TGFβ1 (PeproTech; 100-21)) were added to the culture medium to support microglial survival. One wpt, co-cultured organoid–microglia (neuroimmune organoids) were transferred to a six-well plate and placed on an orbital shaker. The co-cultures were then maintained following the usual organoid maintenance protocol, with the addition of trophic factors.

## Pharmacological manipulation of organoids

Six-week-old organoids were treated with PBS, 100 ng ml$^{-1}$ of recombinant human IGF1 (Abcam; ab269169), 1 μg ml$^{-1}$ of IGF1-neutralizing antibodies (Abcam; ab9572), 1 μM GSK4529 (GSK1904529A; Selleckchem; S1093) or 1 μM picropodophyllin (Selleckchem; S7668) for 48 h. Then 10 μM BrdU (Abcam; ab142567) was added during the last 4 h to label proliferating cells. Organoids were collected immediately after the treatment for IHC analysis. For the DAPT treatment experiment, PBS or 10 μM DAPT (Abcam; ab120633) was applied to MGEOs transplanted with iMG from 10 to 14 dpt. Organoids were then collected at 14 dpt for IHC analysis.

## Immunohistochemistry

We followed the IHC protocol, as previously reported[14,32]. Human tissue samples were fixed and cryosectioned, as described above. Mouse samples were prepared, as described above in 'Mice'. Organoids were fixed in 4% PFA for 30–45 min at room temperature and cryopreserved in 30% sucrose in PBS overnight. The organoids were then embedded in OCT and cryosectioned at 14 μm using a Leica cryostat.

The mounted human slides were defrosted overnight at 4 °C and then dried at 37 °C for 3 h. The mounted organoids and mouse slides were dried directly at 37 °C for 30 min. Antigen retrieval was performed for 5–12 min at 95–99 °C using antigen retrieval buffer (BD Pharmingen; 550524). After antigen retrieval, tissue slices were washed with 1× PBS plus 0.1% or 0.3% Triton X-100 and then blocked in blocking buffer (5–10% serum, 1% bovine serum albumin (BSA) and 0.1% Triton X-100 in PBS, or 1% BSA in 0.3% Triton X-100 in PBS) for 1–1.5 h at room temperature before proceeding to incubation with primary antibodies (Supplementary Table 3) overnight at 4 °C. After washing, sections were incubated with species-specific secondary antibodies conjugated to Alexa Fluor dyes (1:500; Invitrogen) for 1.5–2 h at room temperature. For human and embryonic mouse slides, TrueBlack Lipofuscin Autofluorescence Quencher (1:20 in 70% alcohol; Biotium; 23007) was applied for 3–5 min to block autofluorescence. For EdU staining, the EdU working solution was applied to embryonic mouse brain slices after secondary antibody application following the manufacturer's instructions. Nuclei were counterstained with DAPI (1:1,000 from 1 mg ml$^{-1}$ of stock; Invitrogen; 2031179) for 5 min. Images were captured using a Leica STELLARIS 8 confocal microscope. For organoid experiments, three slices of each organoid were imaged, quantified using ImageJ (1.54) and averaged for the final statistical analysis.

## Three-dimensional reconstruction and image analysis

Three-dimensional reconstructions were generated using the Imaris software (Oxford Instruments). For distance analysis, microglia were reconstructed using surface modules, whereas Ki-67$^+$ or DAPI$^+$ cells were reconstructed with spot modules. The distance from the centre of each cell (spot) to the nearest microglia (surface) was determined using Imaris. The distance distributions of the Ki-67$^+$ and Ki-67$^-$ cells to the nearest microglia were calculated accordingly.

## Single-nucleus preparation

Single-nucleus suspensions were prepared from postmortem human samples. About 50 mg of sectioned freshly frozen human brain tissue was homogenized in lysis buffer (0.32 M sucrose, 5 mM CaCl$_2$, 3 mM MgAc$_2$, 0.1 mM EDTA, 10 mM Tris-HCl, 1 mM dithiothreitol and 0.1% Triton X-100 in diethyl pyrocarbonate-treated water) plus 0.4 U μl$^{-1}$ of RNase inhibitor (Takara; catalogue no. 2313A) on ice. Then, the homogenate was loaded into a 30-ml-thick polycarbonate ultracentrifuge tube (Beckman Coulter; catalogue number 355631), and 9 ml of sucrose cushion solution (1.8 M sucrose, 3 mM MgAc$_2$, 1 mM dithiothreitol and 10 mM Tris-HCl in diethyl pyrocarbonate-treated water) was added to the bottom of the tube. The tubes with tissue homogenate and sucrose cushions were then ultracentrifuged at 107,000$g$ for 2.5 h at 4 °C. The pellet was recovered in 250-μl ice-cold PBS for 20 min, resuspended in nuclei sorting buffer (PBS, 1% BSA, 0.5 mM EDTA and 0.1 U μl$^{-1}$ of RNase inhibitor) and filtered through a 40-μm cell strainer to obtain single-nucleus suspensions for FACS/fluorescence-activated nucleus sorting.

## Single-cell preparation

Single-cell suspensions of 1323-4 hiPSC-derived organoids were prepared using neural tissue dissociation kits (P) (Miltenyi Biotec; 130-092-628) following the manufacturer's instructions. In brief, 12–16 organoids per experimental condition were processed through a gentle two-step enzymatic dissociation procedure, as instructed. Five mg ml$^{-1}$ of Actinomycin D (Sigma-Aldrich; A1410), 10 mg ml$^{-1}$ of anisomycin (Sigma-Aldrich; A9789) and 10 mM triptolide (Sigma-Aldrich; T3652) were added before tissue digestion to inhibit the cellular transcriptome. Following digestion, organoids were mechanically triturated using fire-polished glass pipettes, filtered through a 40-μm cell strainer test tube (Corning; 352235), pelleted at 300$g$ for 5 min and washed twice with Dulbecco's phosphate-buffered saline (DPBS) before proceeding to 10× genomics scRNA library preparation. For samples that needed FACS, the single-cell pellet was resuspended in cell sorting buffer (DPBS, 1% BSA and 0.1 U μl$^{-1}$ of RNase inhibitor).

## FACS and fluorescence-activated nucleus sorting

Single-nucleus suspensions from fresh-frozen human samples were stained with antibodies of PU.1 (Cell Signaling Technology; 81886S; 1:100) and OLIG2 (Abcam; ab225100; 1:2,500) overnight at 4 °C. PU.1 and OLIG2 antibodies were conjugated with fluorescence upon purchase. DAPI (1:1,000) was added for 5 min on the second day. Single-cell suspensions from organoids were stained with DAPI (1:1,000) for 5 min in cell sorting buffer (DPBS; 1% BSA and 0.1 U μl$^{-1}$ of RNase inhibitor). The single-nucleus/cell suspension was then centrifuged at 300$g$ for 5 min, resuspended in nucleus/cell sorting buffer and filtered through a 40-μm cell strainer for final analysis and sorting using a FACSAria II Cell Sorter (BD Biosciences). Target cells were collected in nucleus/cell sorting buffer for future sequencing library preparation.

## Single-cell and single-nucleus RNA library preparation

Nuclei and cells were counted using a haemocytometer and resuspended to a final concentration of 300–1,000 cells/nuclei per microlitre in PBS. Single-nucleus/cell RNA-seq libraries were prepared using 10× Genomics Chromium Next GEM Single Cell v.3.1 kit according to the manufacturer's instructions, targeting for 5,000 nuclei/cells per sample. Single-cell/nucleus libraries were then sequenced on the NovaSeq 6000 machine, with a sequencing depth of 50,000 reads per cell.

## Single-cell and single-nucleus RNA-seq data analysis

Sequencing results were then aligned to the GRCh38 genome (gex-GRCh38-2020-A) using Cell Ranger v.6.1.2 (10× Genomics). Then '--include-introns' was used to include premature messenger RNA in single-nucleus samples. Gene counts then underwent a doublet removal step using DoubletFinder v.2.0.3 (https://www.cell.com/cell-systems/fulltext/S2405-4712(19)30073-0).

The output (count matrix) was used as the main input file for all downstream analyses using Seurat v.5.1.0. For human snRNA-seq, nuclei with UMIs of less than 1,000, gene features of less than 1,000 or more than 100,000 or percentage of mitochondrial genes of more than 3% were filtered out. For organoid scRNA-seq, cells with UMIs of less than 800 or more than 50,000, gene features of less than 500 or more than 10,000 or percentage of mitochondrial genes less than 2% or more than 25% were filtered out. For FACS-isolated iMG scRNA-seq, cells with UMIs of less than 1,000 or more than 80,000, gene features of less than 1,000 or more than 20,000 or mitochondrial genes less than 20% were filtered out. MALAT1, mitochondrial genes (MT-), ribosomal protein-encoding genes (RPS- and RPL-) and haemoglobin genes (HB-) were excluded from further analysis. Standard data normalization, variable feature identification, linear transformations, dimensional reduction, UMAP embedding and unsupervised clustering were conducted using the standard Seurat pipeline[35]. Cell-type cluster identification was performed on the basis of the expression of known marker genes, as shown in Extended Data Figs. 2, 7 and 10. For scRNA-seq of GFP-labelled iMG, iMG were purified in silico using canonical microglia/macrophage markers, including *AIF1*, *CX3CR1*, *C3*, *PTPRC*, *ITGAM* and *CD68*.

We analysed cell–cell interaction using CellChat v.2 (ref. 36). For development-based analysis, independent CellChat files were

generated from 'embryonic' and 'perinatal' Seurat objects, and a comparison analysis was conducted between them. A heat map was created using GraphPad Prism 9 according to the interacting probability of significant ligand–receptor interactions involved in microglial regulation of interneurons (CIN).

DEG analysis was conducted on the basis of the Seurat-default non-parametric Wilcoxon rank-sum test. Pathways with enriched DEGs were generated using Enrichr (https://maayanlab.cloud/Enrichr/#) on the basis of the Reactome Pathway Database, Kyoto Encyclopedia of Genes and Genomes, GEO and Gene Ontology database. The full names of the pathways shown in Fig. 4 are as follows: IGF1R 46: IGF1R drug inhibition 46 (kinase perturbations from GEO down; GSE14024); IGF1R 52: IGF1R knockdown 52 (kinase perturbations from GEO down; GSE16684); mitotic sister: mitotic sister chromatid segregation (GO:0000070); aerobic electron: aerobic electron transport chain; respiratory: respiratory electron transport, ATP synthesis by chemiosmotic coupling, heat production by uncoupling proteins (R-HSA-163200).

### Principal component analysis

The published sequencing datasets for comparison were collected from eleven previous papers[39,41–50]. The specific papers and corresponding NIH GEO datasets used were as follows: GSE89189 (ref. 39); (GSE123021, GSE123022, GSE123024 and GSE123025) (ref. 41); GSE121654 (ref. 42); GSE141862 (ref. 43); (GSE133345 and GSE137010) (ref. 44); GSE180945 (ref. 45); GSE178317 (ref. 46); (GSE139549 and GSE139550) (ref. 47); GSE85839 (ref. 48); GSE97744 (ref. 49); GSE99074 (ref. 50). Each dataset was collected, filtered and grouped by appropriate characteristics, including species, real/derived, bulk/single cell, age and protocol details. To facilitate comparison, the groups within each set were pooled into single representations.

Once the data were collected and preprocessed, the pooled samples were processed using Scanpy v.1.10.3 (https://github.com/scverse/scanpy). Specifically, the cells were normalized by total counts over all genes using scanpy.pp.normalize_total. They were then logarithmized using scanpy.pp.log1p. For use in downstream PCA, highly variable genes were calculated using scanpy.pp.highly_variable_genes. The number of top genes was configured to 2,000. In the final steps, PCA was performed using scanpy.tl.pca (default of 50 components), and a scatter plot using the coordinates of PCA 1 and 2 was plotted for each cell representation using scanpy.pl.pca.

### CRISPR–Cas9 gene editing

A WA09/H9 stem cell line with an IGF1 loss-of-function mutation (IGF1 knockout) was generated using CRISPR–Cas9-based non-homology end joining, largely following the protocol of the Alt-R CRISPR–Cas9 System from Integrated DNA Technologies (IDT). The guide RNA (5′TCGTGGATGAGTGCTGCTTC3′) was selected from the predesigned Alt-R CRISPR–Cas9 guide RNA (IDT). Equal amounts of CRISPR RNA and ATTO 550 labelled tracrRNA (IDT; 1075927) were mixed to a final concentration of 100 μM, heated to 95 °C for 5 min and then cooled to room temperature for annealing followed by the formation of the ribonucleoprotein complex with Alt-R S.p. HiFi Cas9 Nuclease V3 (IDT; 1081061) at room temperature for 20 min. The ribonucleoprotein complex was delivered to single-stem-cell suspensions using the Neon Electroporation System (1,400 V; 20 ms; one pulse) according to the manufacturer's instructions. After electroporation, ATTO 550+ cells were selected by FACS after 3 days of culture and sparsely seeded to form a single-cell colony. A loss-of-function mutation cell line was selected by Sanger sequencing with out-of-frame mutations at the target site, followed by exclusion of any mutations at the top 5 potential off-target sites. Further Sanger sequencing confirmation, reverse transcription–quantitative polymerase chain reaction and IHC were performed to confirm IGF1 knockout.

### Time-lapse imaging of microglia–MGE progenitor interactions

To visualize the interactions between engrafted microglia and MGE progenitors in MGEOs, we used MGE organoids generated from NKX2.1-GFP cells and iMG derived from tdTomato-labelled WA09/H9 cells using lentiviral transduction (SignaGen Laboratories; SL100289). Live imaging was performed 2–3 weeks after iMG transplantation. For imaging, engrafted organoids were transferred to a flat glass-bottom six-well plate, with one organoid per well with 500 μl of culture medium. Time-lapse imaging was conducted using a Leica SP8 confocal microscope at 37 °C and 5% $CO_2$. Z-stacks were captured every 5 min over a 12-h period and processed using maximum intensity projections to visualize dynamic cellular interactions.

### Data analysis, statistics and presentation

For all quantifications, images were acquired and quantified blindly to genotype or treatment. Statistical analyses were performed using GraphPad Prism (v.10.1.0), as shown in each figure legend.

### Reporting summary

Further information on research design is available in the Nature Portfolio Reporting Summary linked to this article.

## Data availability

All data are available in the main text or Supplementary Information. Raw sequencing datasets and processed Seurat objects were deposited in the Gene Expression Omnibus (accession numbers GSE296073 and GSE274829) and Zenodo (https://doi.org/10.5281/zenodo.15299853)[96], respectively. All the other data are available upon request. Source data are provided with this paper.

## Code availability

The custom code used in the analyses is available at GitHub (https://github.com/DIANKUNYU/R-script-used-for-Yu-2025 and https://github.com/codycollier/mglia-nat25).

96. Yu, D., Piao, X. & Wangzhou, A. Microglia regulate GABAergic neurogenesis in prenatal human brain through IGF1. *Zenodo* https://doi.org/10.5281/zenodo.15299853 (2025).

**Acknowledgements** We thank the families who generously donated the tissue samples used in this study. We thank A. Elefanty from the Murdoch Children's Research Institute and Monash University for generously providing the NKX2.1-GFP cell line; A. Alvarez-Buylla, A. Kriegstein, D. H. Rowitch and S. Fancy for discussion and thoughtful comments on the article; C. Mrejen of the UCSF Innovation Core at the Weill Institute of Neurosciences for technical support in microscopy; and C. M. Collier from the University of Texas at Dallas for his help in bioinformatics analysis. This study was supported by funding from the NIH/National Institute of Neurological Disorders and Stroke (NIH/NINDS) grant no. P01 NS083513 (M.F.P., E.J.H. and X.P.), NIH/NINDS grant nos. R01 NS094164 and R01 NS108446 (X.P.), California Institute for Regenerative Medicine grant no. DISCO0-15949 (X.P. and T.J.N.), NIH/NINDS grant no. R01 NS132595 (E.J.H.) and VA BLR&D Merit Review Award I01 BX001108 (E.J.H.).

**Author contributions** D.Y. and X.P. conceived and designed the study. M.F.P., T.J.N., E.J.H. and X.P. supervised the study. D.Y., A.W., B.Z., J.K. and J.J.-Y.C. performed the histological and sequencing experiments with postmortem human samples. D.Y., S.J., W.S., A.W. and S.D.F. performed experiments with organoids. D.Y., A.W. and S.J. conducted the bioinformatics analysis. E.J.C.-O., D.Y. and W.S. conducted mouse experiments. D.Y. and S.J. analysed most of the results and created the figures. S.J., D.Y., A.W. and E.J.C.-O. wrote the Methods. D.Y. and X.P. wrote the paper with comments from all authors.

**Competing interests** The authors declare no competing interests.

**Additional information**
**Correspondence and requests for materials** should be addressed to Diankun Yu or Xianhua Piao.

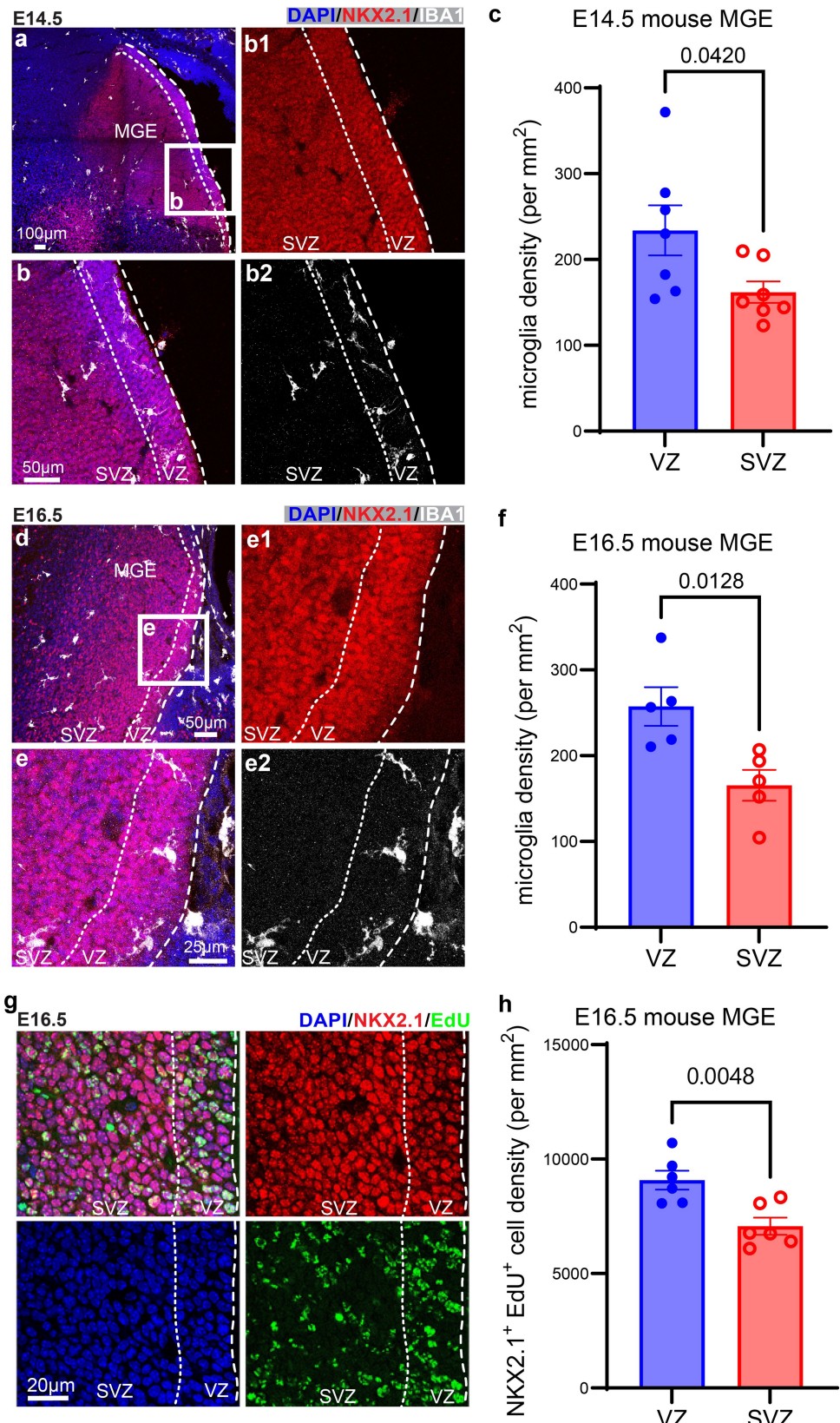

**Extended Data Fig. 1 | The distribution of microglia and proliferation progenitors in mouse MGE. a-b**, IHC images showing the distribution of IBA1[+] microglia in embryonic day (E)14.5 mouse MGE. MGE were labelled by NKX2.1. **c**, Bar graph showing the density of IBA1[+] microglia is significantly higher in the VZ of E14.5 mouse MGE. **d-e**, IHC images showing the distribution of IBA1[+] microglia in E16.5 mouse MGE. **f**, Bar graph showing the density of IBA1[+] microglia is significantly higher in the VZ of E16.5 mouse MGE. **g**, IHC images showing the distribution of EdU[+]NKX2.1[+] active progenitors in the E16.5 mouse MGE with EdU injected at E14.5. **h**, Bar graph showing the density of NKX2.1[+]EdU[+] active progenitors is significantly higher in the VZ of E16.5 mouse MGE with EdU injected at E14.5. N = 7, 5, 6 for (c), (f), (h), respectively; unpaired two-tailed t-test. Data are shown as means ± SEM.

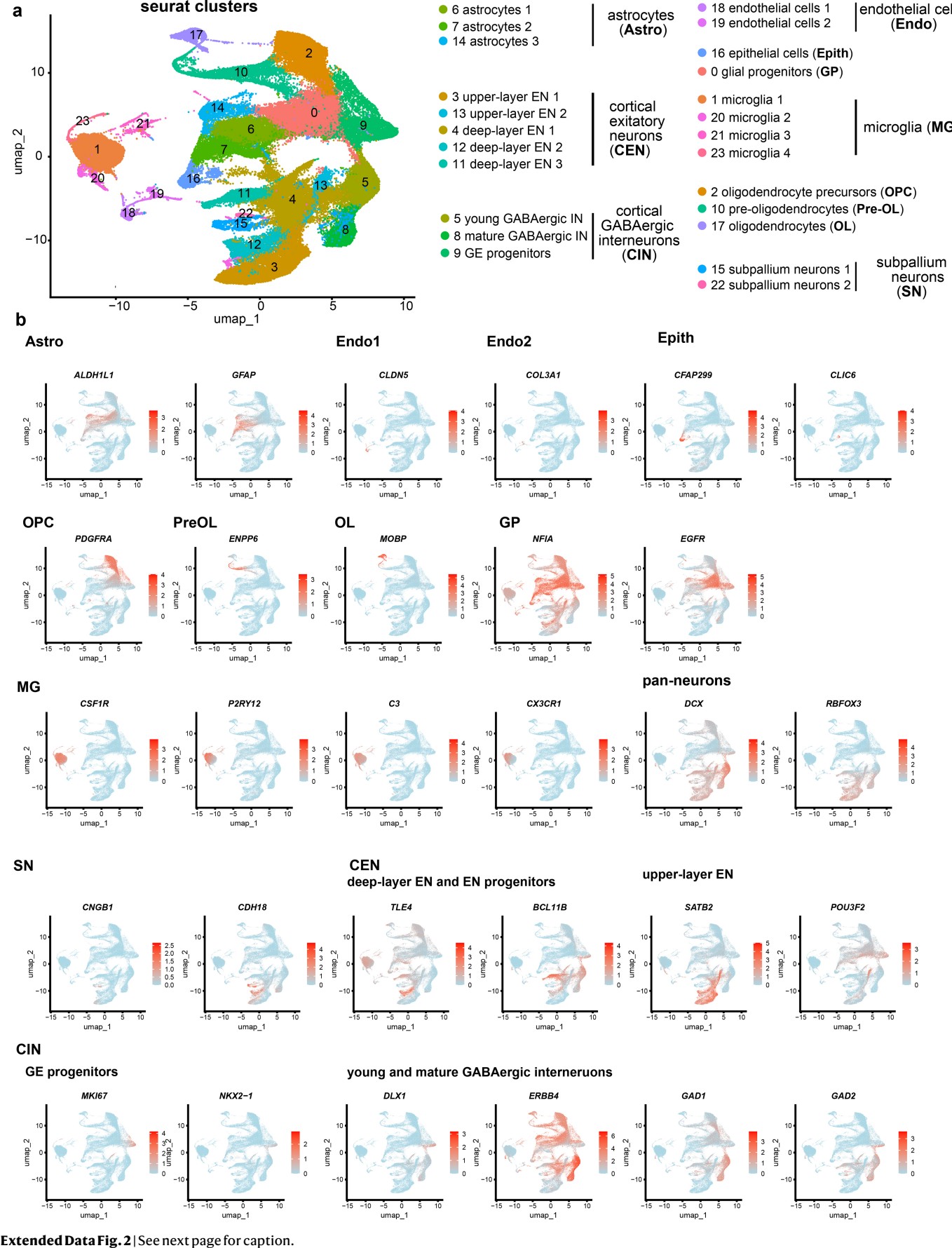

**Extended Data Fig. 2 |** See next page for caption.

**Extended Data Fig. 2 | Identification of cell types in human snRNAseq data.**
**a**, UMAP plots of Seurat clusters and cell type annotations. **b**, Feature plots showing selected canonical markers for different types of cells, including *ALDH1L1* and *GFAP* for astrocytes (Astro), *CLDN5* and *COL3A1* for endothelial cells (Endo), *CFAP299* and *CLIC6* for epithelial cells (Epith), *PDGFRA* for oligodendrocyte precursor cells (OPC), *ENPP6* for pre-myelinating oligodendrocytes (Pre-OL), *MOBP* for oligodendrocytes (OL), *NFIA* and *EGFR* for glia progenitors (GP), *CSF1R*, *P2RY12*, *C3*, and *CX3CR1* for microglia (MG), *DCX* and *RBFOX3* for pan-neurons, *CNGB1* and *CDH18* for subpallium neurons (SN), *TLE4*, *BCL11B*, *SATB2* and *POU3F2* for cortical excitatory neurons (CEN), *MKI67* and *NKX2-1* for GE progenitors, *DLX1*, *ERBB4*, *GAD1*, and *GAD2* for GABAergic interneurons. GE progenitors (cluster 9, *NKX2.1*[+]*MKI67*[+]), young GABAergic interneurons (cluster 5, *DCX*[+] *RBFOX3*[low]*GAD1*[+] *GAD2*[+]), and mature GABAergic interneurons (*RBFOX3*[+]*DCX*[low] *GAD1*[+] *GAD2*[+]) were combined as cortical interneurons (CIN) for the following analysis.

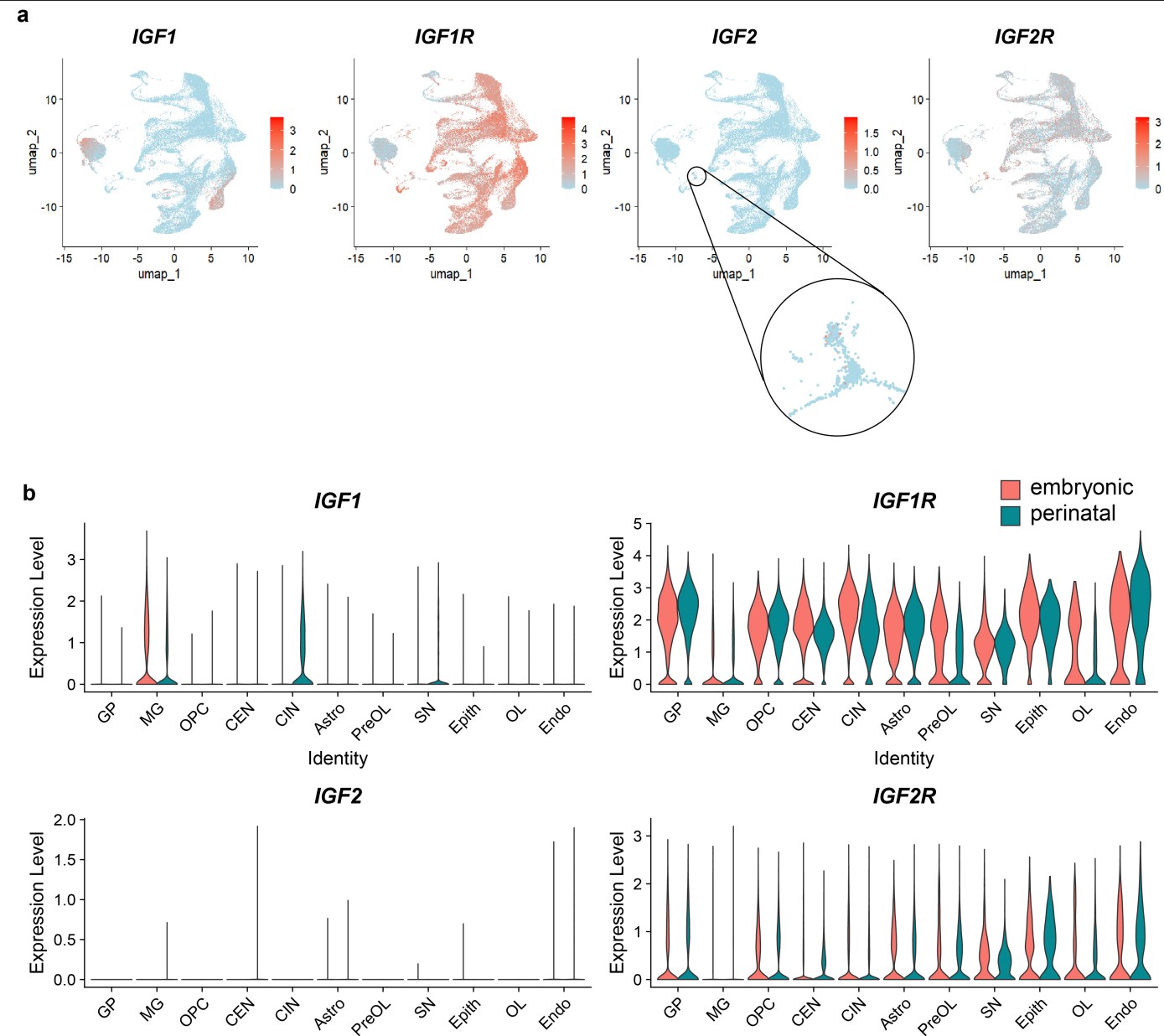

**Extended Data Fig. 3 | The expression patterns of *IGF1*, *IGF1R*, *IGF2*, and *IGF2R* in developing human samples. a**, Feature plots showing the expression of *IGF1*, *IGF1R*, *IGF2*, and *IGF2R* in developing human samples. **b**, Violin plots showing the expression of *IGF1*, *IGF1R*, *IGF2*, and *IGF2R* in embryonic and perinatal human samples.

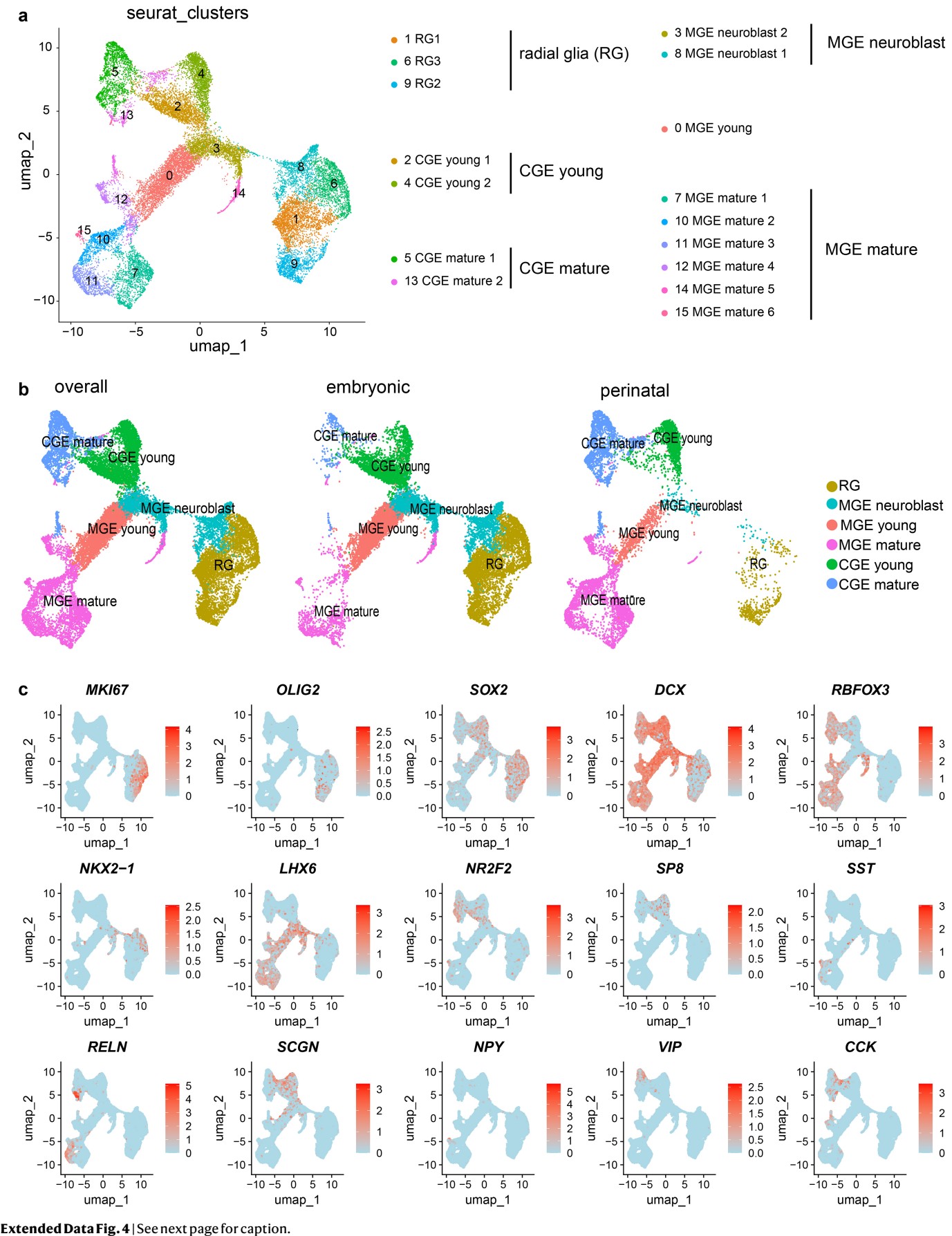

**Extended Data Fig. 4** | See next page for caption.

**Extended Data Fig. 4 | Identification of interneuron subtypes in human snRNAseq data. a**, Interneuron subtype recognition according to Seurat clustering and canonical markers. **b**, UMAP of interneuron subclusters in the embryonic and perinatal stages. **c**, Feature plots of canonical markers. Radial glia (RG) are cell clusters that express high *MKI67, OLIG2*, and *SOX2*; MGE neuroblasts are cell clusters with residual *MKI67, SOX2* and *NKX2-1* expression and starting to express *DCX* and *LHX6*, corresponding to the neuroblast cells within DENs in the hMGE. MGE-derived young interneurons (MGE young) are cell clusters that do not express *MKI67* and *NKX2-1*, but express high levels of *DCX* and *LHX6* and low levels of *RBFOX3*. MGE-derived mature interneurons (MGE mature) are cell clusters that express high levels of *RBFOX3* and *LHX6* and low levels of *DCX*. CGE-derived young interneurons (CGE young) are cell clusters that do not express *MKI67*, but express high levels of *DCX, NR2F2*, and *SP8*, and low levels of *RBFOX3*. CGE-derived mature interneurons (CGE mature) are cell clusters that express high levels of *RBFOX3, NR2F2*, and *LHX6*, and low levels of *DCX*.

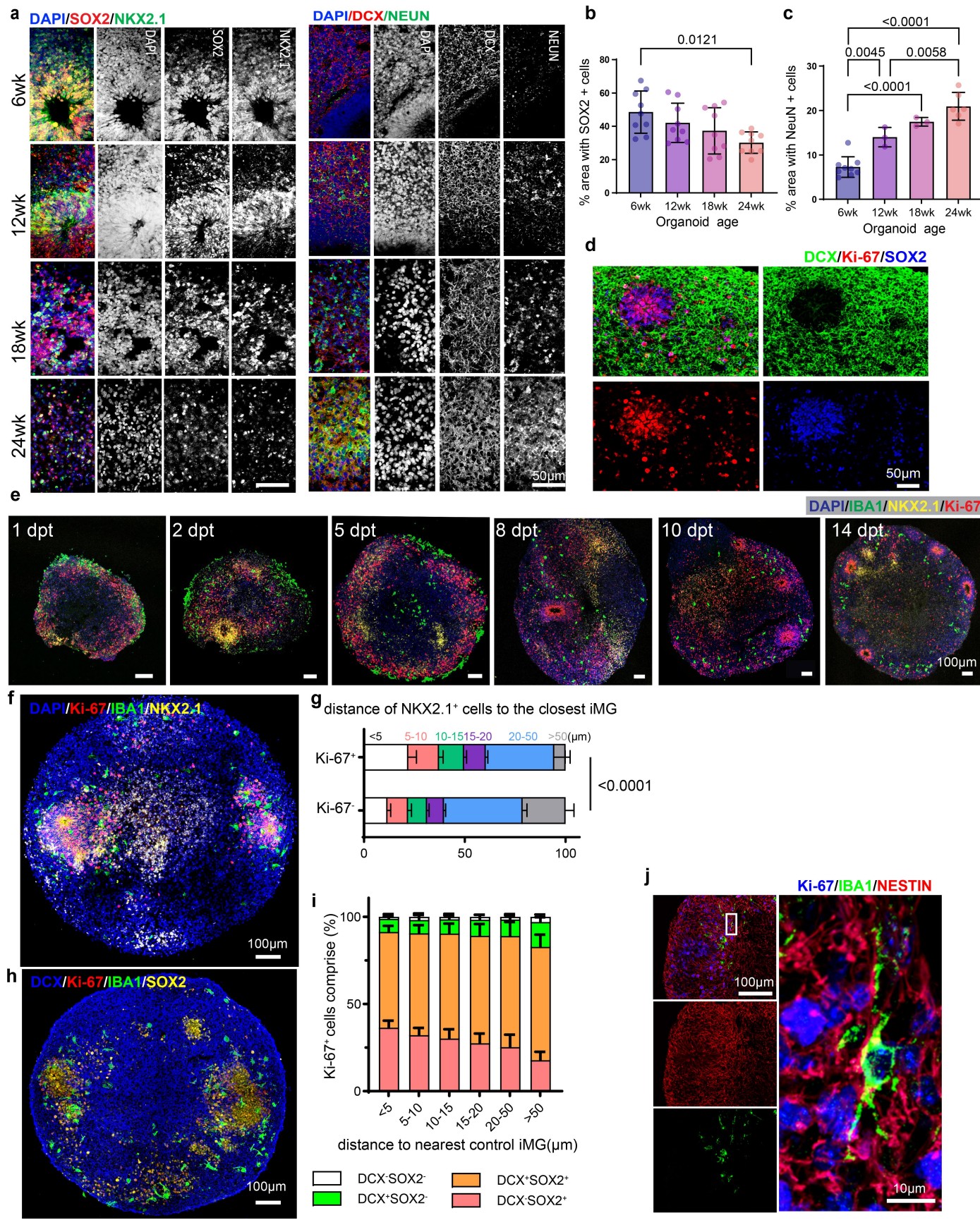

**Extended Data Fig. 5** | See next page for caption.

**Extended Data Fig. 5 | Characterization of MGEOs. a-c**, MGEOs sequentially express markers for radial glia (SOX2) and postmitotic interneurons (DCX and NeuN) as they mature. SOX2$^+$ cells are high at 6 weeks-old and gradually decrease, whereas NeuN$^+$ neurons gradually increase as MGEOs age. **d**, Ki-67$^+$ cells seen in both SOX$^+$DCX$^-$ radial glia and SOX2$^+$DCX$^+$ neuroblasts in 6-week-old MGEOs. **e**, Representative images showing the distribution of iMG in MGEOs at different days post-transplantation. **f-g**, IHC images and bar graphs showing iMG distribute closer to NKX2.1$^+$Ki-67$^+$ proliferating progenitors in comparison to NKX2.1$^+$Ki-67$^-$ cells in 6-week-old MGEOs. **h-i**, IHC images and bar graphs showing the composition of Ki-67$^+$ cells in proximity to iMG in 6-week-old MGEOs. **j**, IHC images showing microglia closely interact with NESTIN$^+$ projections in MGEOs. One-way ANOVA and post-hoc Bonferroni's test in (b) and (c); two-way repeated measures ANOVA for (g) and (i) showed significant interaction effects (P < 0.001 for (i)); N = 4 in (g); N = 5 in (i); data in (b), (c), (g), and (i) were shown as means ± SEM.

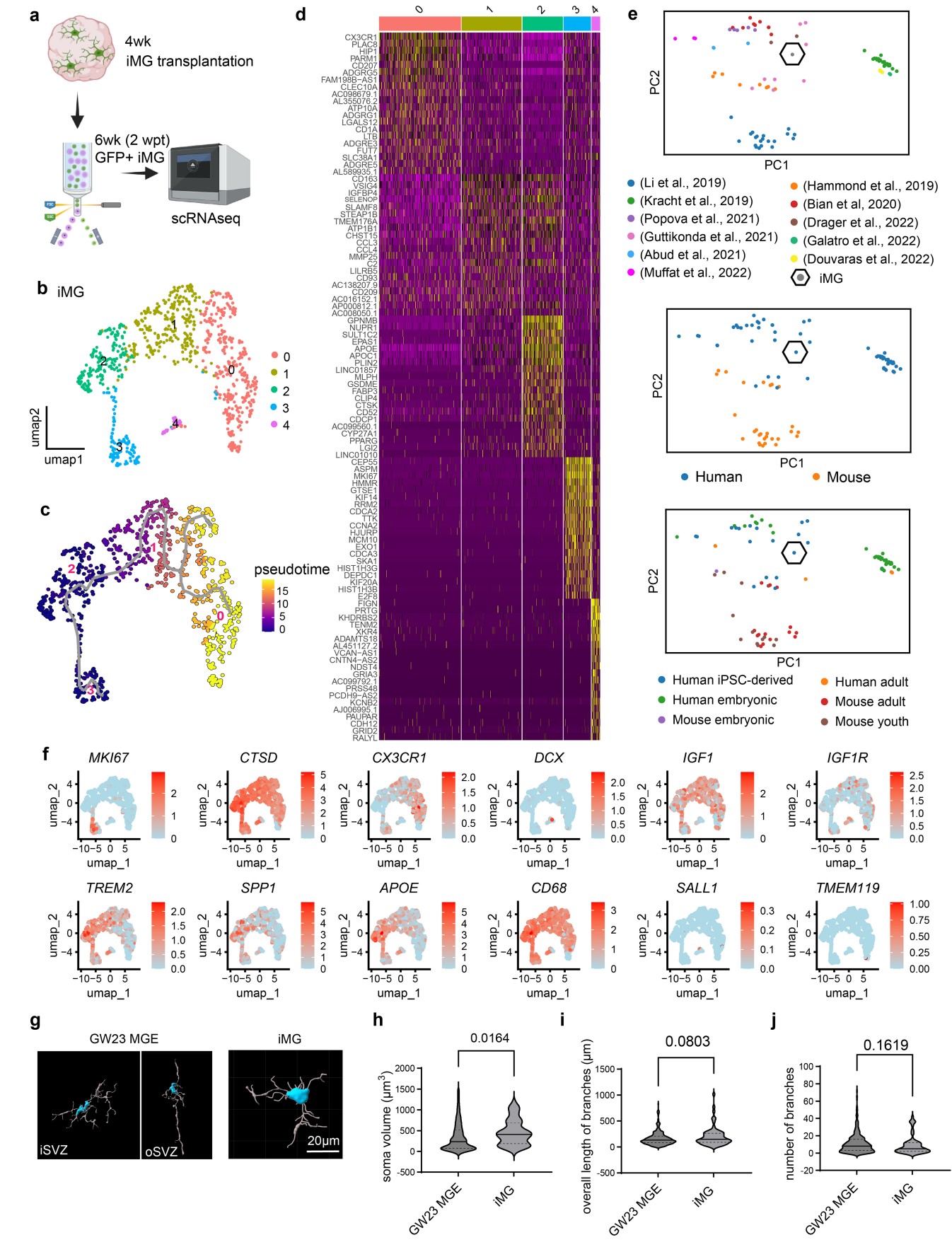

**Extended Data Fig. 6 |** See next page for caption.

**Extended Data Fig. 6 | Transcriptomic subcluster analysis reveals that iMG represents distinct states of primary human microglia. a**, GFP⁺ iMG were isolated from 6-week-old organoids (2 wpt) via FACS and subjected to scRNA-seq. **b**, UMAP visualization of iMG subclusters. **c**, Monocle3-based pseudotime analysis reveals the developmental trajectory of iMG. **d**, Heatmap displaying the top 20 marker genes for each iMG subcluster. **e**, PCA indicates that iMG exhibit transcriptomic profiles similar to published human iPSC-derived and embryonic microglia. **f**, Feature plots of selected marker genes of microglia. Trajectory analysis and marker gene expression suggest that subcluster 0 resembles homeostatic microglia, enriched with *CX3CR1*, *P2RY12*, *CSF1*, and *ADGRG1* (see Fig. 3). Subcluster 2 represents pre-mature and non-homeostatic microglia, marked with *GPNMB*, *CTSD*, *TREM2*, *SPP1*, *APOE*, and *CD68*. Subcluster 3 consists of proliferating microglia enriched in *MKI67*, *RRM2*, and *CDCA2*, while subcluster 4 resembles neuron-associated expressing neuronal genes such as *DCX*, *GRIA3*, and *GRID2*. Notably, *IGF1* expression remains comparable across subclusters, while *SALL1* and *TMEM119* are poorly detected in iMG. **g**, Representative morphologies of primary microglia from GW23 MGE (oSVZ and iSVZ) and iMGs, visualized with IMARIS. **h**, Violin plots showing iMGs exhibited significantly enlarged soma volumes compared to primary microglia from GW23 MGE. Primary microglia from both the oSVZ and iSVZ were included in the analysis. **i-j**, Violin plots showing overall lengths (**i**) and numbers (**j**) of branches in primary microglia from GW23 MGE and iMG. iMGs showed comparable levels of ramification with primary microglia from GW23 MGE. Unpaired two-tailed t test; medians (solid line) and quartiles (dash line) were shown in the violin plots (h)-(j). Illustration in **a** was created using BioRender (https://biorender.com).

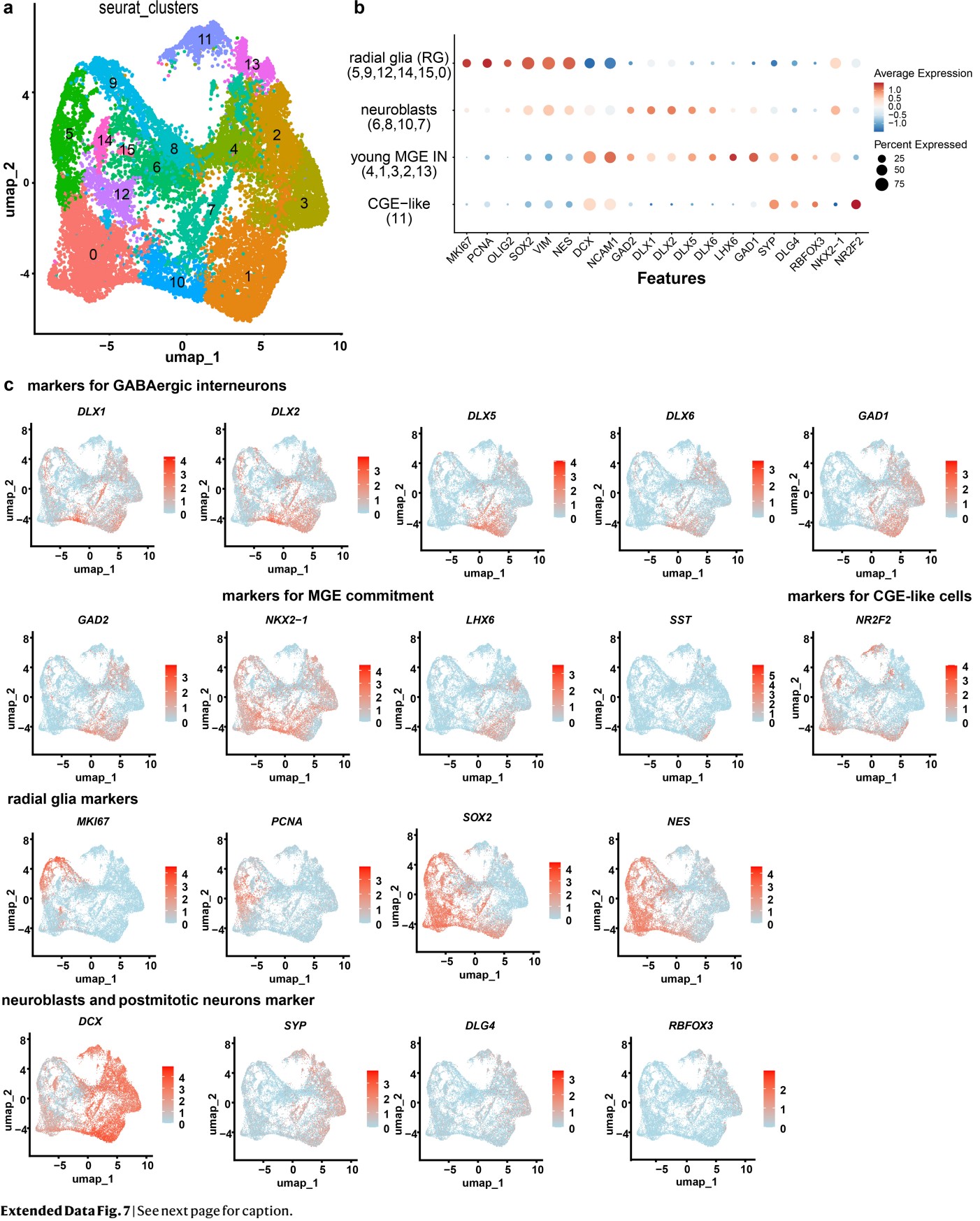

**Extended Data Fig. 7** | See next page for caption.

**Extended Data Fig. 7 | Identification of cell types in 6-week-old MGEOs.**
**a**, UMAP plots of unbiased clustering of MGEOs. **b**, Dot plots showing the level and cell percentage of canonical marker expression. **c**, Feature plots showing the expression of canonical markers. Radial glia (RG) are recognized as cell clusters that express high *MKI67*, *PCNA*, *OLIG2*, *SOX2*, *VIM*, and NESTIN (*NES*); neuroblasts are recognized as transiting clusters expressing both progenitor markers such as *MKI67*, *PCNA*, *SOX2*, and *NES*, as well as the young postmitotic neuron marker *DCX*. Young MGE-derived GABAergic interneurons (Young MGE IN) are recognized as clusters that express *DCX*, *NCAM*, *GAD1*, *GAD2*, *DLX1*, *DLX2*, *DLX5*, *DLX6*, and low levels of *SYP*, *DLG4*, and *RBFOX3*. Notably, without a FACS-based enrichment strategy, iMG were barely detected in this scRNA-seq result, likely due to their low abundance.

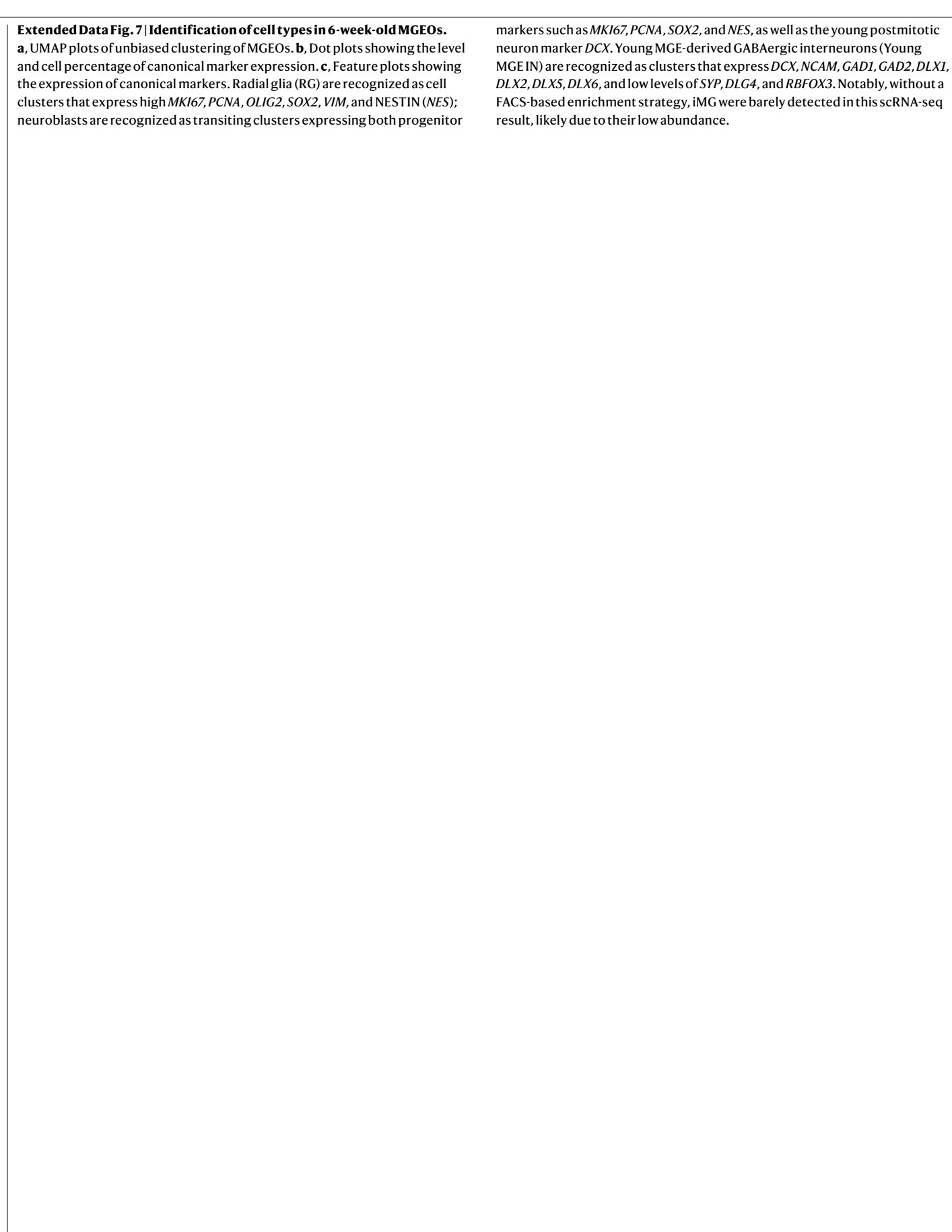

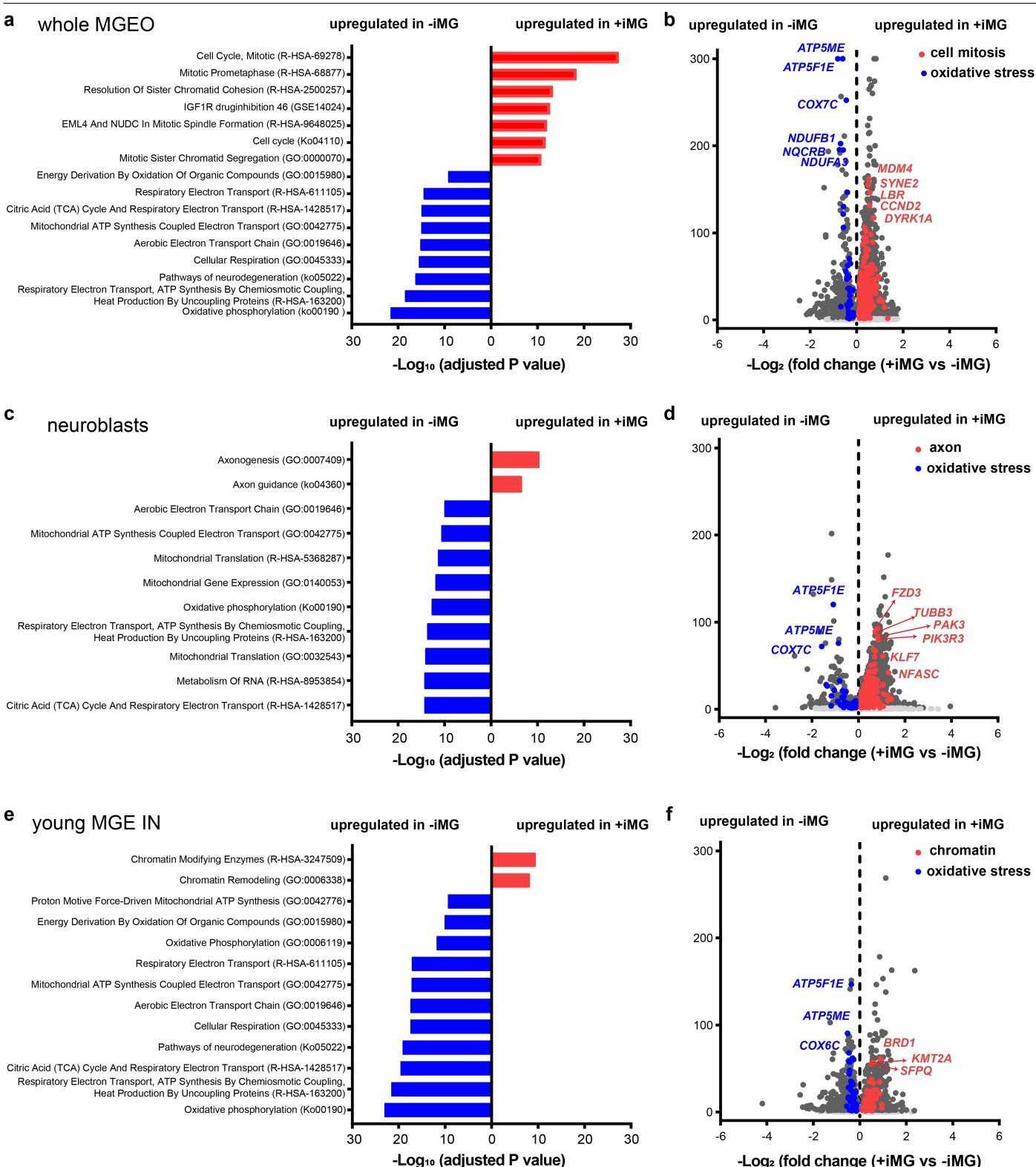

**Extended Data Fig. 8 | DEGs and pathway analysis. a-b**, Pathway enrichment analysis plot (a) and volcano plot (b) of whole MGEO showing a higher expression of genes related to cell mitosis and lower expression of genes related to oxidative stress in MGEOs with iMG. **c-d**, Pathway enrichment analysis plot (c) and volcano plot (d) of neuroblasts cluster revealing an increased expression of genes related to axon development and decreased expression of genes related to oxidative stress in MGEOs with iMG. **e-f**, Pathway enrichment analysis plot (e) and volcano plot (f) of young MGE-derived GABAergic interneuron (young MGE IN) populations showing the presence of iMG leads to an increased the expression of genes related to chromatin metabolism and decreased expression of genes related to oxidative stress. The adjusted P value in (a), (c), and (e) was calculated in Enrichr, using Benjamini-Hochberg correction. The adjusted P value in (b), (d), and (f) was conducted based on the Seurat-default non-parametric Wilcoxon rank sum test.

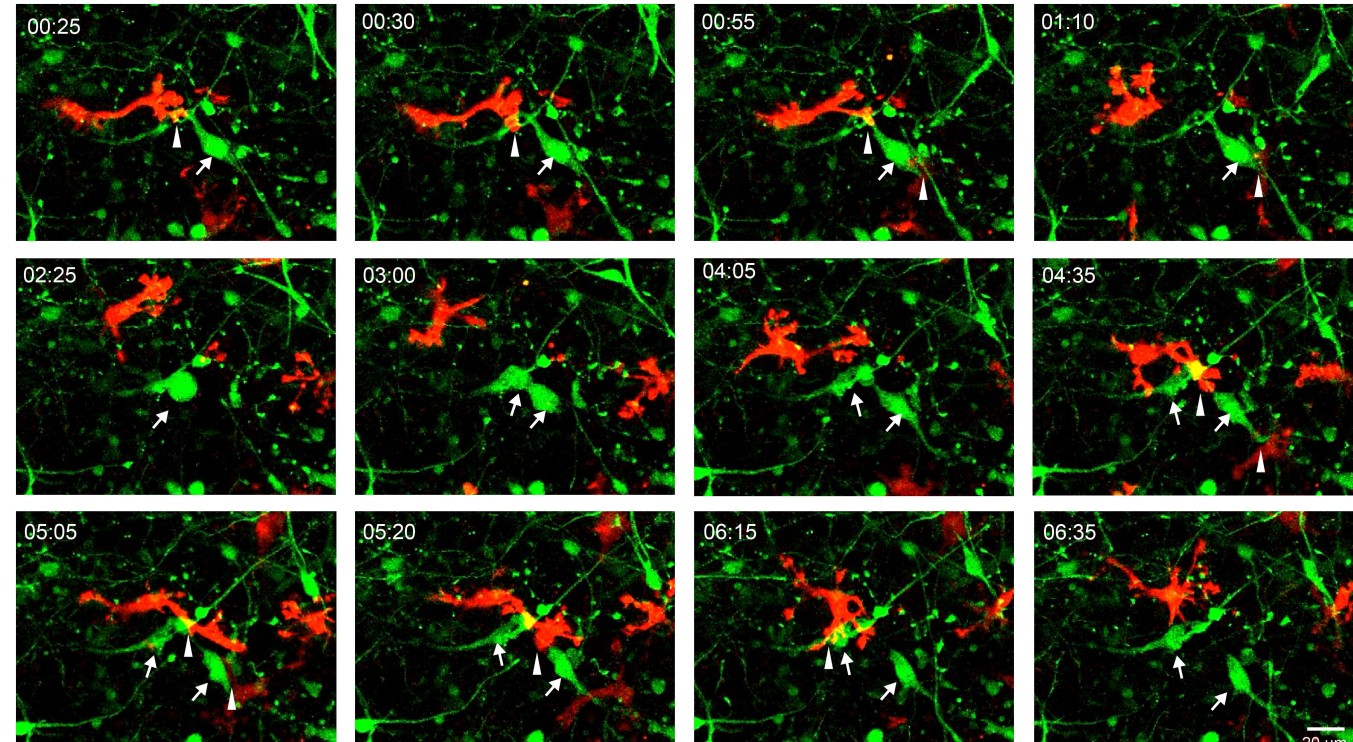

**Extended Data Fig. 9 | Time-lapse live imaging reveals active interactions between iMG and proliferating progenitors.** Arrows indicate cell proliferation, while white triangles highlight interactions between iMG (red, tdtomato labelled) and proliferating mother and daughter cells (green, NKX2.1-GFP). The recording time is shown in the top left corner of each image, formatted as 'hour:minutes'.

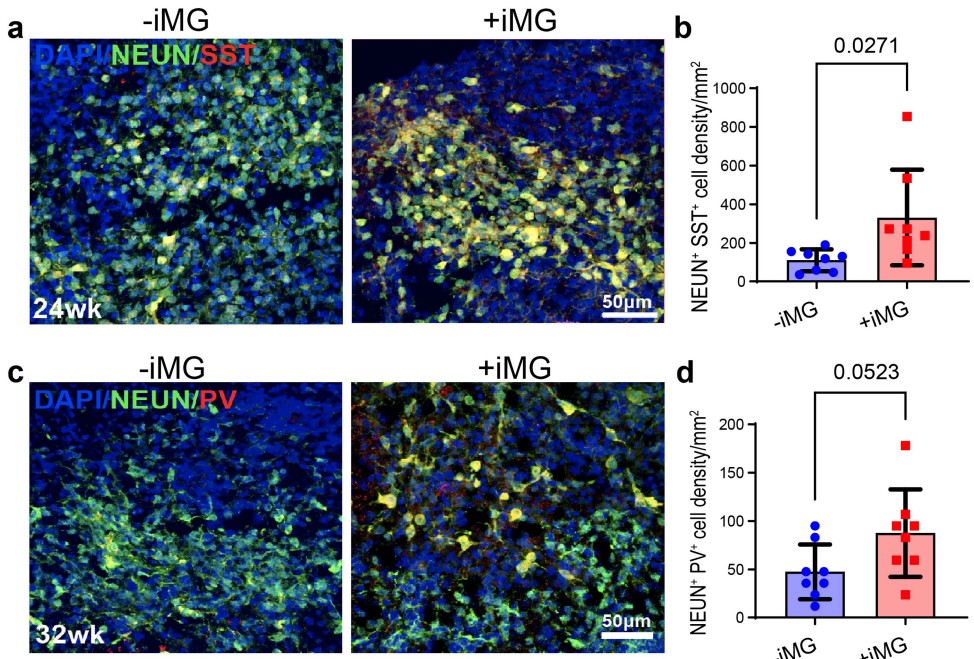

**Extended Data Fig. 10 | iMG promote MGE-derived interneuron production.**
**a-b**, Representative IHC images and bar graphs showing that the presence of iMG results in higher density of SST+NeuN+ interneurons in 24-week-old MGEOs. **c-d**, Representative IHC images and bar graphs showing iMG transplantation leads to a trend in higher density of PV+NeuN+ interneurons in 32-week-old MGEOs. N = 8 in (b) and (d); unpaired two-tailed t-test in (b) and (d); data in (b) and (d) were shown as means ± SEM.

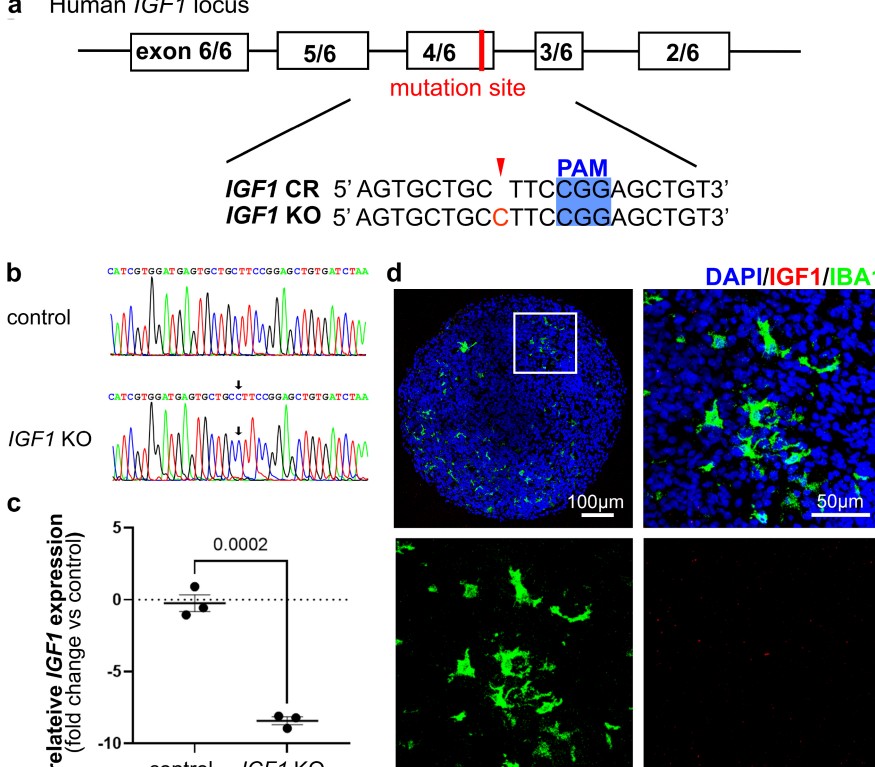

**a** Human *IGF1* locus

exon 6/6 — 5/6 — 4/6 | — 3/6 — 2/6

mutation site

*IGF1* **CR** 5' AGTGCTGC TTCCGGAGCTGT3'
*IGF1* **KO** 5' AGTGCTGCCTTCCGGAGCTGT3'

PAM

**b**

control
CATCGTGGATGAGTGCTGCTTCCGGAGCTGTGATCTAA

*IGF1* KO
CATCGTGGATGAGTGCTGCCTTCCGGAGCTGTGATCTAA

**c**

relateive *IGF1* expression
(fold change vs control)

0.0002

control    *IGF1* KO

**d** DAPI/IGF1/IBA1

100μm    50μm

**Extended Data Fig. 11 | Establishment of *IGF1* loss-of-function mutation stem cell line. a**, Strategy for generation of *IGF1* loss-of-funciton mutation cell line with Cripsr/Cas9-based NHEJ. **b**, Sanger sequencing results confirming *IGF1* mutation. **c**, RT-qPCR of *IGF1* expression in iMG. **d**, IHC of iMG in 6-week-old organoids showing no IGF1 is detected in iMG derived from *IGF1* KO cell line. N = 3 (three independent prepartions of iMG) in (c), unpaird two-tailed t-test. Data in (c) were shown as means ± SEM.

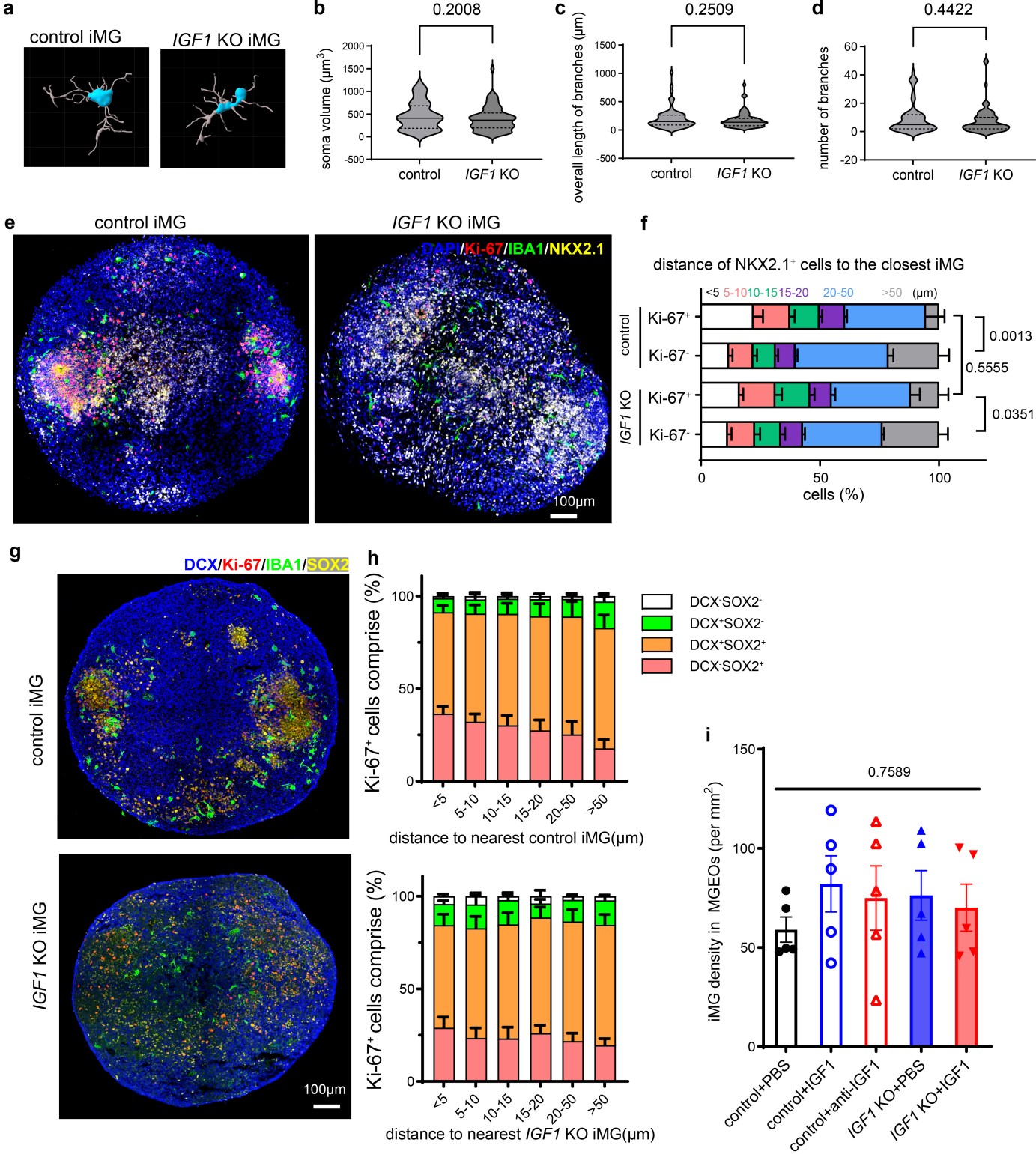

**Extended Data Fig. 12 | *IGF1* KO has minimal impact on iMG morphology, distribution and density. a**, Representative control and *IGF1* KO microglial morphology visualized with IMARIS. **b-d**, Violin plots showing *IGF1* KO does not significantly alter soma volume (b), total branch length (c), or branch number (d) in iMG. **e-f**, IHC images and bar graphs showing that *IGF1* KO iMGs, like control iMGs, localize closer to NKX2.1⁺Ki-67⁺ proliferating progenitors than to NKX2.1⁺Ki-67⁻ cells in MGEOs. **g-h**, IHC images and bar graphs showing a high percentage of SOX2⁺ cells among Ki-67⁺ proliferating progenitors located near *IGF1* KO iMGs, similar to control iMGs. **i**, iMG density in 6-week-old MGEOs is not significantly different among different IGF1 treatment and iMG genotyping conditions. Unpaired two-tailed t-test in (b)-(d); P value of interaction effect of distance x Ki-67 were labelled, two-way repeated measures ANOVA in (f); N = 4,5 in (h); N = 5 in (i), one-way ANOVA. Data in (b)-(d) were shown as violin plots with medians (solid line) and quartiles (dash line) indicated. Data in (f), (h), (i) were shown as means ± SEM.

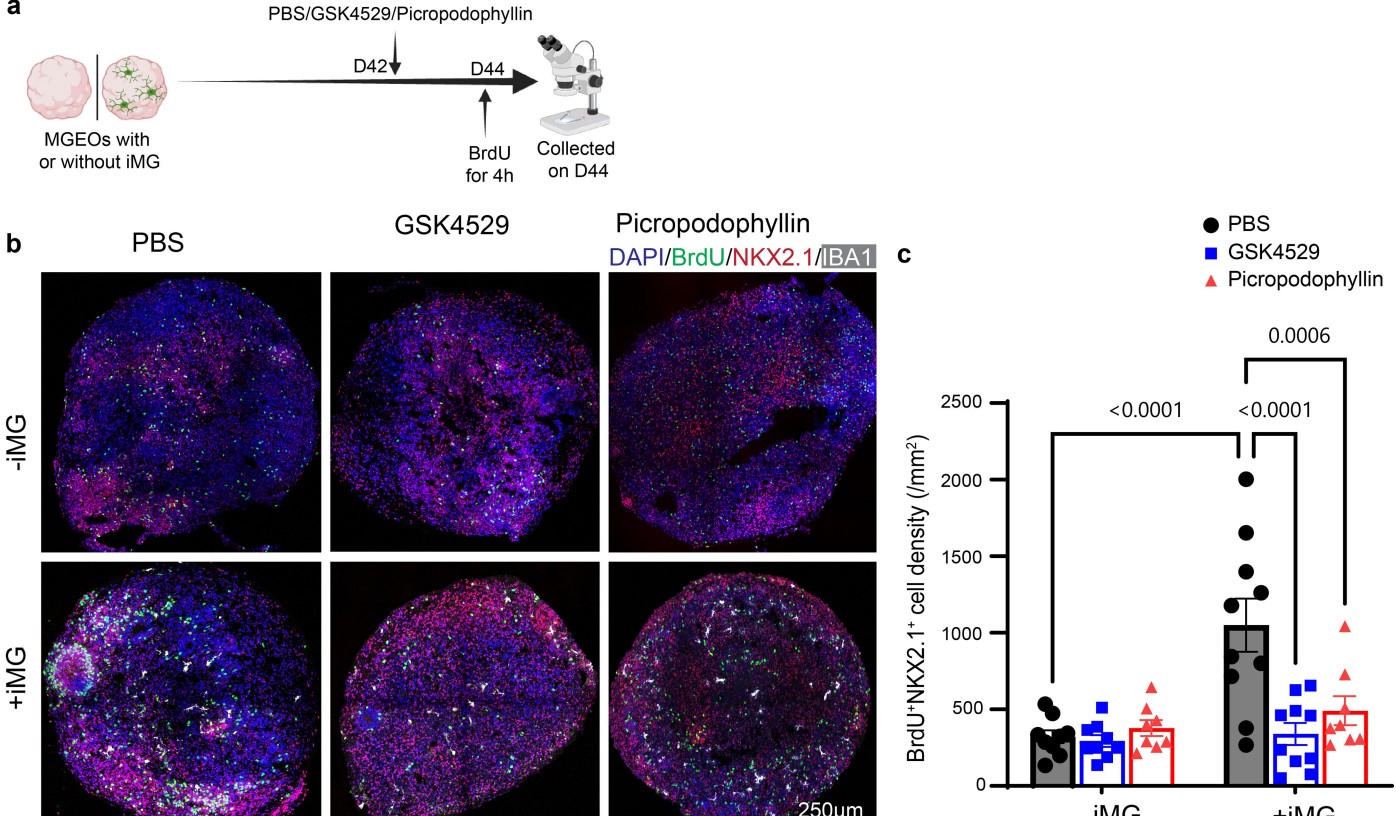

**Extended Data Fig. 13 | IGF1R inhibition prevents iMG-induced promotion of MGE proliferation. a**, Experimental schematic diagram showing that MGEOs transplanted with or without iMG were treated with PBS or IGF1R inhibitors -GSK4529 and Picropodophyllin- for two days at week 6. BrdU was administered during the last four hours to label proliferating cells. **b-c**, IHC images and bar graphs showing that GSK4529 and Picropodophyllin prevented iMG-induced promotion of MGE proliferation. N = 9, 9, 8; 10, 10, 8; two-way ANOVA and post-hoc Bonferroni's test for selected comparisons. Data in (c) were shown as means ± SEM. Illustration in **a** was created using BioRender (https://biorender.com).

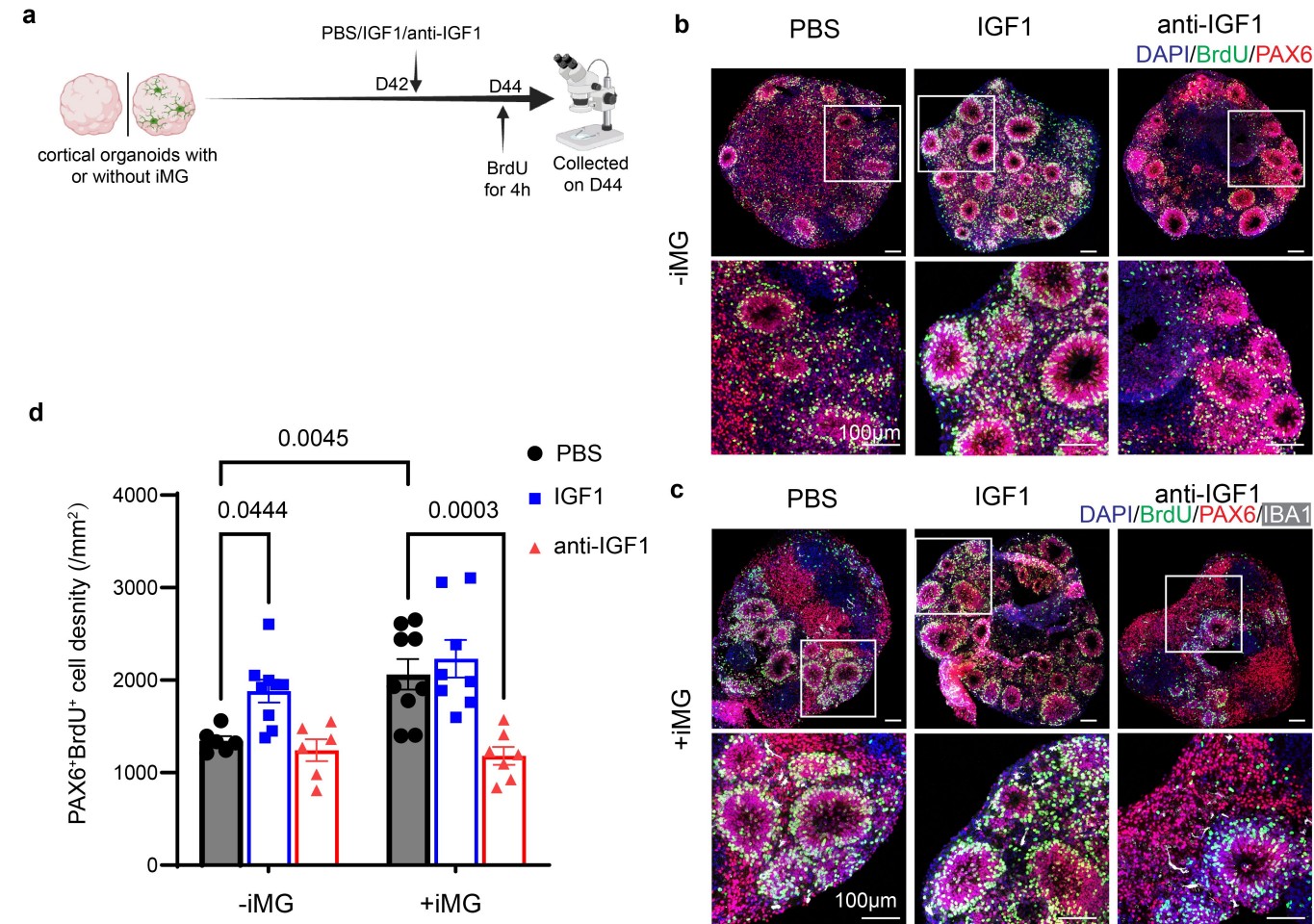

**Extended Data Fig. 14 | iMG-derived IGF1 promotes PAX6⁺ progenitor proliferation in cortical organoids. a**, Experimental schematic diagram showing cortical organoids transplanted with or without iMG were treated with PBS, IGF1, and anti-IGF1 neutralizing antibodies for two days at week 6. BrdU was administered during the last four hours to label proliferating cells. **b**, IHC images showing BrdU⁺PAX6⁺ proliferating progenitors in cortical organoids without iMG. **c**, IHC images showing BrdU⁺PAX6⁺ proliferating progenitors in cortical organoids with iMG. **d**, Bar graphs showing that iMG and IGF1 significantly increased the density of PAX6⁺BrdU⁺ proliferating progenitors in cortical organoids. IGF1-neutralizing antibody treatment abolished iMG-mediated progenitor proliferation. N = 6, 9, 6; 9, 8, 7; two-way ANOVA and post-hoc Bonferroni's test for selected comparisons. Data in (d) were shown as means ± SEM. Illustration in **a** was created using BioRender (https://biorender.com).

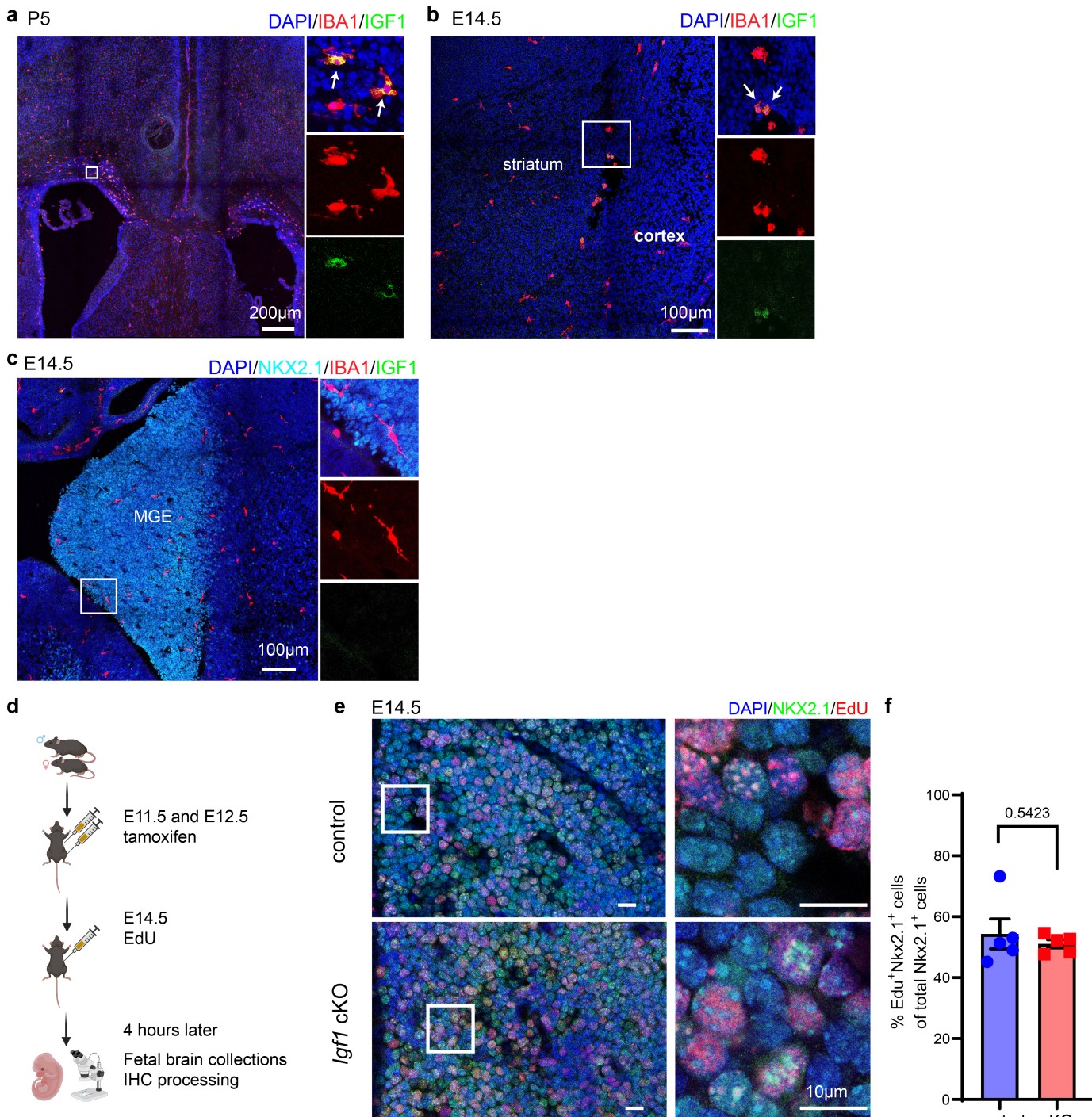

**Extended Data Fig. 15 | Microglial IGF1 is not involved in MGE proliferation in mice. a**, IGF1 is expressed by ATM located at developing white matter at P5. **b**, IGF1 is expressed by ATM-like microglia located at cortico-striato-amygdalar boundary at E14.5. Arrows indicate IGF1+ microglia in **a** and **b**. **c**, IGF1 is not detectable in microglia located in the mouse MGE at E14.5. **d**, Pregnant dams received IP tamoxifen injections at E11.5 and E12.5, followed by an EdU injection at E14.5, four hours prior to fetal brain collection and subsequent IHC processing. **e-f**, IHC images and bar graph showing that microglial *Igf1* cKO mice (*Igf1*^f/f^, *Cx3cr1*^CreERt/+^) show a comparable MGE proliferation to their littermate controls. N = 5 and 5 from three litters. Unpaired two-tailed t-test. Data in (f) were shown as means ± SEM. Illustration in **d** was created using BioRender (https://biorender.com).

# Reporting Summary

## Statistics

For all statistical analyses, confirm that the following items are present in the figure legend, table legend, main text, or Methods section.

| n/a | Confirmed | |
|-----|-----------|---|
| ☐ | ☒ | The exact sample size (*n*) for each experimental group/condition, given as a discrete number and unit of measurement |
| ☐ | ☒ | A statement on whether measurements were taken from distinct samples or whether the same sample was measured repeatedly |
| ☐ | ☒ | The statistical test(s) used AND whether they are one- or two-sided<br>*Only common tests should be described solely by name; describe more complex techniques in the Methods section.* |
| ☐ | ☒ | A description of all covariates tested |
| ☐ | ☒ | A description of any assumptions or corrections, such as tests of normality and adjustment for multiple comparisons |
| ☐ | ☒ | A full description of the statistical parameters including central tendency (e.g. means) or other basic estimates (e.g. regression coefficient) AND variation (e.g. standard deviation) or associated estimates of uncertainty (e.g. confidence intervals) |
| ☐ | ☒ | For null hypothesis testing, the test statistic (e.g. *F*, *t*, *r*) with confidence intervals, effect sizes, degrees of freedom and *P* value noted<br>*Give P values as exact values whenever suitable.* |
| ☒ | ☐ | For Bayesian analysis, information on the choice of priors and Markov chain Monte Carlo settings |
| ☒ | ☐ | For hierarchical and complex designs, identification of the appropriate level for tests and full reporting of outcomes |
| ☒ | ☐ | Estimates of effect sizes (e.g. Cohen's *d*, Pearson's *r*), indicating how they were calculated |

*Our web collection on statistics for biologists contains articles on many of the points above.*

## Software and code

Policy information about availability of computer code

| | |
|---|---|
| Data collection | Leica Application Suite X (4.7.0.28176) was used for microscopic images collection; BD FACSDiva (9.0.1) was used for FACS data collection; |
| Data analysis | Leica LAX (3.7.25997.6), Image J (1.54), and Imaris (9.8.2) used for IHC images analysis and presentation. CellRanger (v6.1.2 ), Rstudio (2024.09.00) and Scanpy (1.10.3)were used for seq data analysis. Graphpad v10 was used for statistical analysis and data presentation. FACS data were analyzed by Floreada.io (https://floreada.io/analysis)<br>Code availability: Zenodo (DOI: 10.5281/zenodo.15299853, https://zenodo.org/records/15299853);<br>        Github (https://github.com/DIANKUNYU/R-script-used-for-Yu-2025 and https://github.com/codycollier/mglia-nat25). |

For manuscripts utilizing custom algorithms or software that are central to the research but not yet described in published literature, software must be made available to editors and reviewers. We strongly encourage code deposition in a community repository (e.g. GitHub). See the Nature Portfolio guidelines for submitting code & software for further information.

## Data

Policy information about availability of data

All manuscripts must include a data availability statement. This statement should provide the following information, where applicable:
- Accession codes, unique identifiers, or web links for publicly available datasets
- A description of any restrictions on data availability
- For clinical datasets or third party data, please ensure that the statement adheres to our policy

> All raw sequencing data will be deposited and publically accessible on Gene Expression Omnibus (GSE296073 and GSE274829), NCBI.
> All other data are available upon request.

## Research involving human participants, their data, or biological material

Policy information about studies with human participants or human data. See also policy information about sex, gender (identity/presentation), and sexual orientation and race, ethnicity and racism.

| | |
|---|---|
| Reporting on sex and gender | De-identified human specimens from both males and females were applied in this study, see more details in extended table 1. |
| Reporting on race, ethnicity, or other socially relevant groupings | N/A |
| Population characteristics | Postmortem human specimens from gestinational week 15 to postnatal week 3 have been used in this study. |
| Recruitment | De-identified human specimens were collected from the Autopsy Service in the Department of Pathology at the University of California San Francisco (UCSF) (Extended Data table 2), with previous patient consent in strict observance of the legal and institutional ethical regulations. The autopsy consents and all protocols for human prenatal brain tissue procurement were approved by the Human Gamete, Embryo and Stem Cell Research Committee (Institutional Review Board GESCR# 10- 02693) at UCSF. |
| Ethics oversight | UCSF: Institutional Review Board GESCR# 10- 02693 |

Note that full information on the approval of the study protocol must also be provided in the manuscript.

# Field-specific reporting

Please select the one below that is the best fit for your research. If you are not sure, read the appropriate sections before making your selection.

☒ Life sciences ☐ Behavioural & social sciences ☐ Ecological, evolutionary & environmental sciences

For a reference copy of the document with all sections, see nature.com/documents/nr-reporting-summary-flat.pdf

# Life sciences study design

All studies must disclose on these points even when the disclosure is negative.

| | |
|---|---|
| Sample size | No methods were used to predetermine sample sizes. Regarding to experiments using postmortem human tissues, we included as many samples per age range as we can obtain. For others, minimum sample sizes were determined based on previously published studies.<br>For snRNAseq, donors=6;<br>For scRNAseq of MGEO, two batch of experiments were conducted. Each batch contains 12-16 organoids in each condition.<br>For scRNAseq of iMG in organoids, 16 6-week-old MGEOs with iMG were used for iMG library preparation.<br>For IHC with postmortem human tissues, generally 3-6 de-identified samples were used for each experiment.<br>For IHC with MGEO and Cortical organoids, 3-11 organoids were used for each used. No statistical methods were used to predetermine sample sizes.<br>For IHC with mouse tissues, 5-7 fetuses or pups from 3 litters were used for each experiment.<br>For RT-qPCR confirmation of IGF1 KO cell line, 3 biological repeats were applied. |
| Data exclusions | no data were excluded |
| Replication | For scRNAseq of MGEO to test the effect of iMG on MGEO development, two batches of organoids were used. Each batch has 12-16 organoids in each condition.<br>For IHC to test the effect of H9 hESC-induced iMG, IGF1 treatment, IGF1R inhibitor treatment, and IGF1 KO on 1323-4 hiPSC-induced MGEO development, 3-4 batches of organoids were used. For IHC to test microglia distribution in MGEOs and DAPT treatment, two batches of organoids were used. For IHC to confirm the effect in other cell lines, at least 2 batches of organoids were tested.<br>For IHC in mouse experiments, three independent litters were tested. |

The biological repeats number "N" values (<10) were indicated by displaying individual data points in the figure and were also included in the figure legend.

Randomization | Organoids were randomized to each experimental groups. For other experiments, no covariates were considered since no treatment was administered.

Blinding | For all quantification, data were acquired and quantified blindly to genotype or treatment.

# Reporting for specific materials, systems and methods

We require information from authors about some types of materials, experimental systems and methods used in many studies. Here, indicate whether each material, system or method listed is relevant to your study. If you are not sure if a list item applies to your research, read the appropriate section before selecting a response.

## Materials & experimental systems

| n/a | Involved in the study |
|---|---|
| ☐ | ☒ Antibodies |
| ☐ | ☒ Eukaryotic cell lines |
| ☒ | ☐ Palaeontology and archaeology |
| ☐ | ☒ Animals and other organisms |
| ☒ | ☐ Clinical data |
| ☒ | ☐ Dual use research of concern |
| ☒ | ☐ Plants |

## Methods

| n/a | Involved in the study |
|---|---|
| ☒ | ☐ ChIP-seq |
| ☐ | ☒ Flow cytometry |
| ☒ | ☐ MRI-based neuroimaging |

## Antibodies

Antibodies used
Doublecortin (DCX) Rabbit 1:500 Cell Signaling Technology 4604S
Doublecortin (DCX) Guinea pig 1:500 EMD Millipore AB2253
Iba1 Guinea pig 1:500 Synaptic Systems 234 308
IGF1 Rat 1:250 R&D Systems MAB2913
IGF1 Goat 1:250 R&D Systems AF791
IGF1R Goat 1:100 R&D Systems AF-305-NA
Ki-67 Mouse 1:500 BD Pharmingen 550609
Ki-67 Rat 1:200 Invitrogen 14-5698-80
BrdU Mouse 1:50 - 1:100 BD Biosciences 347580
P2RY12 Rabbit 1:500 AnaSpec, Inc. AS-55043A
SOX2 Mouse 1:500 Santa Cruz Biotechnology sc-365823
NESTIN Mouse 1:100 BD Transduction Laboratories 611658
NESTIN Mouse 1:500 Millipore MAB5326
NKX2.1 Rabbit 1:500 Abcam ab76013
DLX2 Rabbit 1:250 Abcam ab272902
LHX6 Mouse 1:500 Santa Cruz Biotechnology sc-271433
GAD67 Mouse 1:250 Chemicon International MAB5406
NeuN Guinea pig 1:200 EMD Millipore ABN90
SST Rat 1:200 EMD Millipore MAB354
PV Mouse 1:250 EMD Millipore MAB1572
PAX6 Rabbit 1:250 Cell Signaling Technology 60433S
OLIG2 Rabbit 1:2500 Abcam ab225100
PU.1 Rabbit 1:100 Cell Signaling Technology 81886S

Validation | All of these antibodies are selected from published literature and the species and application were validated by the manufacturer.

## Eukaryotic cell lines

Policy information about cell lines and Sex and Gender in Research

Cell line source(s) | The eWT-1323-4 hiPSC line 45 (female, RRID: CVCL_0G84) was obtained from the Conklin Laboratory (University of California, San Francisco (UCSF). WA09/H9 (female, RRID: CVCL_9773, NIH registration number: NIHhESC-10_0062) and WA01/H1 (male, RRID: CVCL_9771, NIH registration number: NIHhESC-10-0043) were obtained from the WiCell Research Institute (Madison, WI, USA). The NKX2.1-GFP cell line (female) was obtained from MCRI and Monash University (Parkville, Victoria, Australia).

Authentication | The cell lines were not authenticated since obtained.

Mycoplasma contamination | All stem cell lines were tested negative for mycoplasma.

| Commonly misidentified lines (See ICLAC register) | No |
|---|---|

## Animals and other research organisms

Policy information about studies involving animals; ARRIVE guidelines recommended for reporting animal research, and Sex and Gender in Research

| Laboratory animals | C57/B6 mice, age E12.5 to P5; both male and female were used in this study. 2-6 months male and female IGF1 f/f mice (Jax, 012663) and 2-6 months male and female Cx3cr1-CreERt/+ (Jax, 020940) mice were crossed to generate IGF1f/f, Cx3cr1-CreERt/+ (F2) mice for this study. E14.5 IGF1f/f, Cx3cr1-CreERt/+ fetuses and their littermates were involved in this study. Both male and female were involved in this study. |
|---|---|
| Wild animals | No wild animals were used in this study. |
| Reporting on sex | both embryonic male and female were used in this study. |
| Field-collected samples | No field collected samples were used in the study. |
| Ethics oversight | All mice were handled according to the guidelines of the Institutional Animal Care and Use Committee at the University of California, San Francisco. |

Note that full information on the approval of the study protocol must also be provided in the manuscript.

## Plants

| Seed stocks | Report on the source of all seed stocks or other plant material used. If applicable, state the seed stock centre and catalogue number. If plant specimens were collected from the field, describe the collection location, date and sampling procedures. |
|---|---|
| Novel plant genotypes | Describe the methods by which all novel plant genotypes were produced. This includes those generated by transgenic approaches, gene editing, chemical/radiation-based mutagenesis and hybridization. For transgenic lines, describe the transformation method, the number of independent lines analyzed and the generation upon which experiments were performed. For gene-edited lines, describe the editor used, the endogenous sequence targeted for editing, the targeting guide RNA sequence (if applicable) and how the editor was applied. |
| Authentication | Describe any authentication procedures for each seed stock used or novel genotype generated. Describe any experiments used to assess the effect of a mutation and, where applicable, how potential secondary effects (e.g. second site T-DNA insertions, mosiacism, off-target gene editing) were examined. |

## Flow Cytometry

### Plots

Confirm that:

☒ The axis labels state the marker and fluorochrome used (e.g. CD4-FITC).

☒ The axis scales are clearly visible. Include numbers along axes only for bottom left plot of group (a 'group' is an analysis of identical markers).

☒ All plots are contour plots with outliers or pseudocolor plots.

☒ A numerical value for number of cells or percentage (with statistics) is provided.

### Methodology

| Sample preparation | Described in Methods |
|---|---|
| Instrument | BD FACS Aria II cell sorter |
| Software | FACSDiva, Floreada.io |
| Cell population abundance | As shown in Figure 2b. |
| Gating strategy | FSC/SSC were used to identify single cell/single nuclei population. DAPI was used to identify single nuclei in single nuclei isolation. DAPI negative was used to identify live cells in single cell sorting. PU.1-PE, OLIG2-AF647, and GFP was used to identify selected population. Further sequencing results (Fig. 2c, Fig. 3l, extended data Fig. 3, extended data Fig. 9) validate the flow cytometry results. |

☒ Tick this box to confirm that a figure exemplifying the gating strategy is provided in the Supplementary Information.

