## [Peer Review file · Nature]

Microglia regulate GABAergic neurogenesis in prenatal human brain via IGF1

Corresponding Author: Dr Diankun Yu

Version 0:

Reviewer comments:

Referee #1

(Remarks to the Author)

Microglia have been implicated in brain development, mainly by removing apoptotic neurons and synaptic pruning. Here, the authors provide evidence for an intricate interplay between microglia and the generation of interneurons in the medial ganglionic eminence (MGE). They show that microglia are present and located near interneuron progenitor cells in the MGE in human embryonic tissue. Using single-nuclei RNA-seq from human embryonic and perinatal tissue, IGF1 is identified as a likely candidate protein secreted by microglia acting on progenitor cells expressing the receptor IGF1R. To further validate the role of microglia and IGF1 on the proliferation of interneuron progenitor cells, they use an in vitro human MGE organoid (MGEO) model. In organoids engrafted with iPSC-derived microglia, progenitor cell proliferation is increased in the vicinity of microglia compared to organoids lacking microglia. Blocking IGF1 with an IGF1 neutralizing antibody or knockout of IGF1 in microglia using CRISPR/Cas9 resulted in decreased proliferation of interneuron progenitors. Taken together, the authors demonstrate that microglia modulate the generation of interneurons through IGF1. The studies are carefully designed and well-conducted, using both human brain tissue and experimental models. The imaging data and gene expression analysis are carried out at a high level of technical competences and the studies provide strong evidence that microglia are integral in mediating the proliferation of interneuron progenitor cells, with one caveat discussed below. The findings are clearly presented and the main message will be of general interest to the neuroscience and microglia communities. However, to place these findings in context, the authors need to clarify the novelty and specificity of the IGF pathway for interneuron neurogenesis, improve integration of their findings with other published data sets and more clearly specify what they consider to be the novel aspects of the MGEO model.

Major comments:

1. Regarding the statement, 'Microglia derived IGF1 promotes MGE progenitor proliferation in the MGEOs', the IGF1 KO in iMG sufficiently demonstrated the importance of microglia for MGE progenitors' proliferation in the MGEO model system. However, in order to connect the cell-cell cross talk, the proximity quantification of control iMG and IGF1 KO iMG to neighboring NKX2.1+Ki67+ cells and SOX2+/DCX+ radial glia cells should be performed.
2. To validate the impact of microglia on the proliferation of interneuron progenitor cells, the authors integrate hPCs into MGEO. Microglia are known to die in a short period, and thus, to promote survival and maturation of hPSCs within MGEOs, CSF1, IL34, and TGFB1 are added. This approach has been applied successfully by other groups using forebrain organoids and thus is not entirely novel (Schafer et al. 2023). Furthermore, it is unclear how generated MGEOs differ from forebrain organoids since established forebrain organoid protocols are used (References 35, 36).
3. A role of IGF1 in mediating proliferation and promoting neurogenesis has been shown in rodent models (e.g., (Anderson et al. 2002; O'Kusky, Ye, and D'Ercole 2000)). It seems that in light of these previous findings, the observed effect of IGF1 is not specific to interneuron progenitor proliferation but is general to adult neurogenesis. It should be noted that changes in the gene expression of IGF1 between embryonic and perinatal microglia were previously shown in another study investigating embryonic and perinatal microglia (Han et al. 2023).
4. The authors state that iMG within MGEO demonstrated similar morphologies and transcriptomes compared to human

primary embryonic microglia. However, the authors did not systematically compare the morphology (e.g., soma size, number and length of branches) of integrated iMGs to microglia from human embryonic brain tissue. Likewise, no comparison with published data transcriptomic data sets (e.g., (Kracht et al. 2020; Bian et al. 2020) was conducted. While iMGs may be similar to primary microglia, iMGs in the MGEO may still be transcriptionally different from primary microglia, e.g., expression of SALL1 or TMEM119. This point needs further clarification.

5. The authors performed single-cell RNA-seq on MGEOs containing iMG, but the data presented in Figure 4 do not show any recovered microglia. This most likely is due to the low fraction of cells, but the reason should be stated. It would also be interest to know whether there are consequences of KO of IGF1 in microglia on the microglia themselves.

Minor comments:

1. Limited QC data are available on the analysis of the microglia single-cell RNAseq and single-nuclei brain data (e.g., how many reads per cell and genes per cell). Did the authors account for doublets, ambient RNA, and mitochondria? How was the integration of the different data sets done?

2. The staining of spatial distribution of microglia during development hMGE is state of art. An open question here: how to explain the increasing of microglia in both Isvz and Osvz from GW15 to GW40. Is it because of the increasing numbers of microglia during development or because the microglia migration to VZ during MGE development?

REFERENCES

- Anderson, M. F., M. A. Aberg, M. Nilsson, and P. S. Eriksson. 2002. 'Insulin-like growth factor-I and neurogenesis in the adult mammalian brain', *Brain Res Dev Brain Res*, 134: 115-22.
- Bian, Z., Y. Gong, T. Huang, C. Z. W. Lee, L. Bian, Z. Bai, H. Shi, Y. Zeng, C. Liu, J. He, J. Zhou, X. Li, Z. Li, Y. Ni, C. Ma, L. Cui, R. Zhang, J. K. Y. Chan, L. G. Ng, Y. Lan, F. Ginhoux, and B. Liu. 2020. 'Deciphering human macrophage development at single-cell resolution', *Nature*, 582: 571-76.
- Bordt, E. A., A. M. Ceasrine, and S. D. Bilbo. 2020. 'Microglia and sexual differentiation of the developing brain: A focus on ontogeny and intrinsic factors', *Glia*, 68: 1085-99.
- Han, C. Z., R. Z. Li, E. Hansen, S. Trescott, B. R. Fixsen, C. T. Nguyen, C. M. Mora, N. J. Spann, H. R. Bennett, O. Poirion, J. Buchanan, A. S. Warden, B. Xia, J. C. M. Schlachetzki, M. P. Pasillas, S. Preissl, A. Wang, C. O'Connor, S. Shriram, R. Kim, D. Schafer, G. Ramirez, J. Challacombe, S. A. Anavim, A. Johnson, M. Gupta, I. A. Glass, Laboratory Birth Defects Research, M. L. Levy, S. B. Haim, D. D. Gonda, L. Laurent, J. F. Hughes, D. C. Page, M. Blurton-Jones, C. K. Glass, and N. G. Coufal. 2023. 'Human microglia maturation is underpinned by specific gene regulatory networks', *Immunity*, 56: 2152-71 e13.
- Kracht, L., M. Borggrewe, S. Eskandar, N. Brouwer, S. M. Chuva de Sousa Lopes, J. D. Laman, S. A. Scherjon, J. R. Prins, S. M. Kooistra, and B. J. L. Eggen. 2020. 'Human fetal microglia acquire homeostatic immune-sensing properties early in development', *Science*, 369: 530-37.
- O'Kusky, J. R., P. Ye, and A. J. D'Ercole. 2000. 'Insulin-like growth factor-I promotes neurogenesis and synaptogenesis in the hippocampal dentate gyrus during postnatal development', *J Neurosci*, 20: 8435-42.
- Schafer, S. T., A. A. Mansour, J. C. M. Schlachetzki, M. Pena, S. Ghassemzadeh, L. Mitchell, A. Mar, D. Quang, S. Stumpf, I. S. Ortiz, A. J. Lana, C. Baek, R. Zaghaf, C. K. Glass, A. Nimmerjahn, and F. H. Gage. 2023. 'An in vivo neuroimmune organoid model to study human microglia phenotypes', *Cell*, 186: 2111-26 e20.
- Yagi, S., and L. A. M. Galea. 2019. 'Sex differences in hippocampal cognition and neurogenesis', *Neuropsychopharmacology*, 44: 200-13.

Referee #2

(Remarks to the Author)

This study focuses on the function of microglia-derived IGF1 in the proliferation of human medial ganglionic eminence (MGE) progenitors and the generation of GABAergic interneurons. However, the findings regarding the function of microglia and microglia-specific IGF-1 in brain progenitor proliferation and neurogenesis are consistent with earlier research that have reached similar conclusions in mouse models.

Microglia's role in neural progenitor regulation has been supported through several studies, where conditional depletion of microglia leads to the decrease in number of Tbr2+ intermediate progenitors in the SVZ (Arno et al., 2014), neural precursor cells (Antony et al., 2011; Yamamiya et al., 2019), and oligodendrocyte progenitors (Hagemeyer et al., 2017). The role of microglia in regulating PV+ interneuron development has also been shown through the work of Squarzoni et al. (2014) and Yu et al. (2022).

Furthermore, IGF-1 has been shown to play a role in proliferation, where overexpression of IGF-1 results in increased brain size and underexpression of IGF-1 leads to a decrease in brain size (Popken et al., 2004; Rusin et al., 2024). IGF-1 enhances neural stem cell proliferation in vitro (Supeno et al., 2013), and *Igf-1*^{-/-} mice show decreased cell proliferation in the subventricular zone of the lateral ventricle (Hurtado-Chong et al., 2009). IGF-1 has also been indicated to promote neural precursor cell differentiation into ganglionic eminence GABAergic neurons in vivo through the PDK1-Akt pathway (Oishi et al., 2009).

The current manuscript offers valuable insights, but due to previous literature, further work is necessary to meet Nature's publication standards. For instance, the authors have proposed an intriguing potential for a species-specific mechanism involving microglia and microglia-derived IGF1 in neural progenitor proliferation and interneuron production. Conducting additional experiments to explore the species-specific mechanisms of microglia and microglia-derived IGF1 in humans

compared to mice would significantly enhance the novelty and impact of this study.

Other concerns:

- A deeper characterization of microglia states in ex vivo tissue or of induced microglia (iMG) within the Microglia-Engrafted Organoid (MGEO) model is necessary to understand their activation state and functional role. This could be accomplished by analyzing specific activation markers, such as TREM2, which is crucial for microglial activation and function. Since microglia can exist in different phenotypic states, as shown in figure 1I, these varying states may influence brain progenitor cells differently, especially in terms of their effects on cell proliferation.
- It also remains possible that the microglia's proximity to proliferative progenitors is coincidental and does not necessarily indicate a direct functional role. Microglia may be attracted to these regions for reasons unrelated to progenitor proliferation, such as immune surveillance or debris phagocytosis. Additional experimental controls are required to eliminate these alternative explanations and confirm any direct influence microglia may have on progenitor cell activity. It would be beneficial to perform live imaging approaches to get real-time information on the functional interactions between microglia and progenitor cells, and show how microglia migrate to proliferative zones, interact with progenitor cells, and maybe impact their division processes.
- For figure. 4d, it is not clear how the analysis was done. It would be helpful to clarify if the organoids were pooled or the data is showing individual organoids to represent the reproducibility. The standard deviations in this panel are also missing.
- For figure. 5e, it would be nice to show the comparison between the second(+PBS) and the third(-IGF1) conditions.
- For figure. 5g, please show the comparison between the first condition (no iMG) and the last condition (IGF1 KO iMG+ IGF1)

Referee #3

(Remarks to the Author)

In their manuscript the authors address an important question related to the role of microglia in the development of GABAergic interneurons. They first investigate human samples by using immunohistochemistry and found that microglia associate with neurogenic radial glia in the ganglionic eminences (where interneurons originate) and around DENs (DCx+ cells-Enriched Nests). Next they performed single cell RNA sequencing and by using curated database found an enrichment of IGF1 expression in microglia (besides mature interneurons), and enrichment of IGF1R in radial glia and interneurons. The authors then hypothesized that IGF1-IGF1R signaling could play a role in radial glia-mediated interneuron generation. To this end they first established organoid-based approaches and cultured hPSC-derived MGE organoids (MGEOs) with and without addition of hESC-induced iMGs. Strikingly, the addition of iMGs increased proliferation and generation of interneurons in MGEOs. They then tested the role of IGF1 and found that addition of IGF1 increased and blocking IGF1 decreased proliferating cell density. Lastly, the authors knocked out IGF1 in iMGs and could demonstrate that the proliferation-promoting effect of iMGs in MEGOs was lost. The authors provide a model how microglia-derived IGF1 promotes the proliferation of interneuron-generating radial glia and proliferating neuroblasts in DENs.

This study provides interesting data relevant for our understanding of the role of microglia in human interneuron development. The data is convincing and the presentation of the study is quite neat. A few questions have come up while assessing this manuscript.

1. Is there a role for microglia-derived IGF1 in promoting the proliferation of radial glia that generate cortical projection neurons in human, or is the effect specific for the generation of interneurons?
2. The authors knocked out IGF1 in microglia, does the reverse experiment (i.e. knocking out IGF1R in radial glia and/or DENs) support their claims?
3. Related to the above point, IGF2 has been shown to have a role in (postnatal) neurogenesis (cf e.g. studies from Ferron laboratory). Is there evidence (expression in human dataset) that IGF2 might also influence interneuron generation? Regardless of the finding, a slightly broader discussion about the role of IGF signaling in neurogenesis might provide some more integrated context of the findings.
4. The authors show that the density of microglia is increased in mouse when compared to human. They suggest a species-specific feature underlying the difference. This is an interesting point since further investigation of the mechanisms how IGF1 might promote the proliferation of interneuron-generating radial glia could be enabled by further genetic studies in the mouse. Therefore, it would be interesting to assess whether microglia-derived IGF1 also promotes the proliferation of radial glia in mouse MGE? Assessing the generality of the findings across species would be important in order to know if the observed role for microglia-derived IGF1 is a human specific feature or conserved across evolution.

Version 1:

Reviewer comments:

Referee #1

(Remarks to the Author)

The authors have substantially strengthened their manuscript by addressing the outstanding concerns with additional experiments.

#1: Species-specific IGF1/IGF1R signaling in human interneuron genesis.

The concern was based on previous publications of mouse neuronal development. IGF1 is not specific to interneuron progenitor proliferations, and an elaborate body of literature from mouse studies has demonstrated roles of IGF1 during neurogenesis. In response, the authors carried out cross-species single cell analysis from published datasets, together with IHC studies. The authors found IGF1 expression in some microglia states, but did not detect IGF1 in MGE microglia during development. Moreover, the authors generated microglia specific conditional knock out of *Igf1* in mice. By IHC studies, the authors found that *Igf1* is not involved in MGE development in mouse microglia.

#2: The specific enrichment of microglia in SVZ.

This concern came from our open question, which was challenging to answer from direct studies in human brain tissues. The authors analyzed the single cell RNAseq from hMGEs, identified limited numbers of proliferating microglia in MGEs. Hence, they proposed that microglia enrichment in SVZ is not because of proliferation, but because of microglial migration. They later used Notch signaling inhibitor DAPT to block neural progenitor proliferation and found evenly distributed microglia in MGEO.

#3: Is IGF1/IGF1R MGE specific?

The authors also generated cortical organoids, that treated with PBS, IGF1 or anti-IGF1, and found that iMG and iMG-derived IGF1 increase PAX6+ neural progenitors in cortical organoids.

Overall, the results and conclusions stated in the abstract are of significance and generally well supported by data. The one major recommendation is to revise the below sentence in the abstract to include the word predicted (highlighted), as this relationship is experimentally validated in the organoid but cannot be directly shown in the human brain:

We show that microglia preferentially distribute in the proliferating zone and identify insulin-like growth factor 1 (IGF1) and its receptor IGF1R as the *predicted* top ligand-receptor pair underlying microglia-progenitor communication in the prenatal human brain.

We note some additional minor concerns:

1. Extended Figure 4, 5 need more detailed figure legends on the cross-talk analysis.
2. The number of biological replicates of human samples that were used for the single cell analysis was not clear. .
3. Extended figure 19 d. It's not advisable to compare UMI/cells from different single cell or single nuclei RNAseq of one gene. UMI/cell relies heavily on library preparation, sequencing depth and batch effects. It's difficult to draw the conclusion that IGF1 is expressed at significant higher levels in human microglia than compared to rodents from this comparison alone.
4. Extended Figure 16f, comparison should also be made of between control and IGF1KO.

Referee #2

(Remarks to the Author)

I enjoyed reading the revised version of the manuscript. All major concerns have been addressed. As a minor note, it would be very helpful if the authors could clarify which state of human primary microglia (ramified or polarized) is used in the morphological analysis presented in Extended Figure 9g-j. This clarification would provide additional information on whether a specific microglial state is involved in progenitor proliferation and neurogenesis.

Referee #3

(Remarks to the Author)

The authors have done a great job for their revision. They addressed all concerns of all reviewers very well and also provided new and convincing data that enhance the overall conceptual advance. It is quite intriguing to see the significance of species-specific differences (mouse vs human) in their new data set. In my opinion the findings should be reported in a timely manner.

May 3, 2025

Thank you for the opportunity to submit our revised manuscript #2024-08-17269, titled "Microglia regulate GABAergic neurogenesis in prenatal human brain through IGF1" at *Nature*. We are grateful for the constructive comments from the reviewers. In response to the reviewers' comments, we have performed additional experiments and revised our manuscript accordingly. Below, please find our point-by-point responses to the reviewers' comments.

Referee #1 (Remarks to the Author):

Microglia have been implicated in brain development, mainly by removing apoptotic neurons and synaptic pruning. Here, the authors provide evidence for an intricate interplay between microglia and the generation of interneurons in the medial ganglionic eminence (MGE). They show that microglia are present and located near interneuron progenitor cells in the MGE in human embryonic tissue. Using single-nuclei RNA-seq from human embryonic and perinatal tissue, IGF1 is identified as a likely candidate protein secreted by microglia acting on progenitor cells expressing the receptor IGF1R. To further validate the role of microglia and IGF1 on the proliferation of interneuron progenitor cells, they use an in vitro human MGE organoid (MGEO) model. In organoids engrafted with iPSC-derived microglia, progenitor cell proliferation is increased in the vicinity of microglia compared to organoids lacking microglia. Blocking IGF1 with an IGF1 neutralizing antibody or knockout of IGF1 in microglia using CRISPR/Cas9 resulted in decreased proliferation of interneuron progenitors. Taken together, the authors demonstrate that microglia modulate the generation of interneurons through IGF1. The studies are carefully designed and well-conducted, using both human brain tissue and experimental models. The imaging data and gene expression analysis are carried out at a high level of technical competences and the studies provide strong evidence that microglia are integral in mediating the proliferation of interneuron progenitor cells, with one caveat discussed below. The findings are clearly presented and the main message will be of general interest to the neuroscience and microglia communities. However, to place these findings in context, the authors need to clarify the novelty and specificity of the IGF pathway for interneuron neurogenesis, improve integration of their findings with other published data sets and more clearly specify what they consider to be the novel aspects of the MGEO model.

We appreciate the thoughtful evaluation and valuable suggestions. Below, we provide point-by-point responses to each comment. Specifically, we elaborate on the novelty by demonstrating a species-specific role of the IGF1 pathway in human interneuron neurogenesis. Additionally, we compare the morphology and transcriptomics of iMG with published microglia datasets and clarify key details of the MGEO model.

Major comments:

1. Regarding the statement, 'Microglia derived IGF1 promotes MGE progenitor proliferation in the MGEOs', the IGF1 KO in iMG sufficiently demonstrated the importance of microglia for MGE progenitors' proliferation in the MGEO model system. However, in order to connect the cell-cell cross

talk, the proximity quantification of control iMG and IGF1 KO iMG to neighboring NKX2.1+Ki67+ cells and SOX2+/DCX+ radial glia cells should be performed.

We thank the reviewer for the excellent suggestion. We have since conducted the suggested analysis and presented the results in Extended Data Fig. 8 f-i and Extended Data Fig. 16 e-h. Both control and *IGF1* KO iMG showed closer proximity to the proliferating progenitors (NKX2.1+Ki-67+) in comparison to the NKX2.1+Ki-67- non-proliferating cells. Similar to what we observed in the hMGE, Ki-67+ progenitors in proximity to microglia were largely SOX2+DCX- radial glia. We have revised the main text accordingly (line 8-10, page 5; line 43-45, page 6).

2. To validate the impact of microglia on the proliferation of interneuron progenitor cells, the authors integrate hPCs into MGEO. Microglia are known to die in a short period, and thus, to promote survival and maturation of hPSCs within MGEOs, CSF1, IL34, and TGF β 1 are added. This approach has been applied successfully by other groups using forebrain organoids and thus is not entirely novel (Schafer et al. 2023). Furthermore, it is unclear how generated MGEOs differ from forebrain organoids since established forebrain organoid protocols are used (References 35, 36).

We added “consistent with previous report³⁸” (line 4, page 5) to better describe our results in the context of previous discoveries.

We primarily followed “Option C” from (Sloan et al., *Nat Protoc*, 2018, PMID: 30202107) and the “ISRA protocol” from (Birey et al., *Nature*, 2017, PMID: 28445465), with some modifications, namely incorporating 96-well ultra-low attachment V-shaped plates to generate uniformed ventral organoids enriched in MGE progenitors and MGE-derived GABAergic neurons (Detailed in Methods). Our further immunohistochemistry (Fig. 3b-g) and scRNA-seq analysis (Fig. 4b) confirmed a predominant MGE cell fate in our organoids. We have revised the main text (line 31-34, page 4) to clarify the generation and naming of MGEOs.

3. A role of IGF1 in mediating proliferation and promoting neurogenesis has been shown in rodent models (e.g., (Anderson et al. 2002; O'Kusky, Ye, and D'Ercole 2000)). It seems that in light of these previous findings, the observed effect of IGF1 is not specific to interneuron progenitor proliferation but is general to adult neurogenesis. It should be noted that changes in the gene expression of IGF1 between embryonic and perinatal microglia were previously shown in another study investigating embryonic and perinatal microglia (Han et al. 2023).

We agree with the reviewer that IGF1 has a general role in promoting neurogenesis. We added two new paragraphs in the discussion to more thoroughly contextualize our results (page 7 to 8). Moreover, our new data suggest a species-specific expression and function of IGF1 in microglia-mediated interneuron development. We presented the result showing microglial IGF1 is not involved in mouse MGE proliferation in Extended Data Fig. 19.

4. The authors state that iMG within MGEO demonstrated similar morphologies and transcriptomes compared to human primary embryonic microglia. However, the authors did not systematically compare the morphology (e.g., soma size, number and length of branches) of integrated iMGs to microglia from human embryonic brain tissue. Likewise, no comparison with published data transcriptomic data sets (e.g., (Kracht et al. 2020; Bian et al. 2020)) was conducted. While iMGs may be similar to primary microglia, iMGs in the MGEO may still be transcriptionally different from primary microglia, e.g., expression of *SALL1* or *TMEM119*. This point needs further clarification.

We thank the reviewer for the comments. We have since performed principal component analysis (PCA) to compare our iMG with 7 published transcriptomic datasets of iPSC-induced microglia and primary human and mouse microglia. The results are shown in Extended Data Fig. 9e. We illustrated the expression of multiple genes in our iMG in feature plots (Extended Data Fig. 9f). As previously reported (Schafer et al., *Cell*, 2023, PMID: 37172564; Park et al., *Nature*, 2023, PMID: 37914940), we hardly saw any expression of SALL1 and TMEM119 in our iMG (Extended Data Fig. 9f). Furthermore, we performed a morphology comparison between iMG and primary human microglia in GW23 hMGE, using IMARIS (Extended Data Fig. 9 g-j). Our results showed the transplanted iMG demonstrated enlarged soma volume but similar ramification as human primary microglia in the hMGE around GW23. We have revised the main text accordingly (line 11-13, line 16-19, page 5).

5. The authors performed single-cell RNA-seq on MGEs containing iMG, but the data presented in Figure 4 do not show any recovered microglia. This most likely is due to the low fraction of cells, but the reason should be stated. It would also be interesting to know whether there are consequences of KO of IGF1 in microglia on the microglia themselves.

We agree with the reviewer on the reason for the low fraction of microglia. We stated “Notably, without a FACS-based enrichment strategy, iMG were barely detected in this scRNA-seq result, likely due to their low abundance,” in the main text (Line 40-41, page 5) and the legend of Extended data Fig. 10.

To compare the effect of IGF1 deletion on the microglia, we first compared the morphology of control and *IGF1* KO iMGs in 6-week-old MGEs (2 weeks post-transplantation). Our analysis revealed no significant morphological differences between the two groups (Extended Data Fig. 16 a-d). Furthermore, the spatial proximity of iMGs to NKX2.1⁺Ki-67⁺ cells and SOX2⁺ progenitors was comparable between control and *IGF1* KO conditions (Extended Data Fig. 16 e-h). The density of iMGs in MGEs was also not significantly affected by IGF1 deletion nor by administration of IGF1 protein or IGF1 neutralizing antibody (Extended Data Fig. 16i). Collectively, these results indicate that IGF1 knockout does not have an obvious effect on the morphology, distribution, or density of iMGs. We have revised the main text accordingly (line 43-45, page 6).

Minor comments:

1. Limited QC data are available on the analysis of the microglia single-cell RNAseq and single-nuclei brain data (e.g., how many reads per cell and genes per cell). Did the authors account for doublets, ambient RNA, and mitochondria? How was the integration of the different data sets done?

We thank the reviewer for the kind suggestion. We now added two sentences in the main text: “After quality control (QC) and doublets removal (see details in Methods), we recovered 124,411 nuclei with a median of 7,341 unique molecular identifiers (UMIs) and 2,925 genes per cell.” (line 41-43, page 3) and “We recovered 21,136 cells with a median of 5,045 UMIs and 2,629 genes per cell.” (line 37, page 5). Furthermore, we included more detailed steps in the Methods (page 27) including doublet removal, filtering outliers and removing genes - *MALAT1*, mitochondria genes (*MT*-), ribosomal protein encoding genes (*RPS*- and *RPL*-), and hemoglobin genes (*HB*-). In our snRNA-seq and scRNA-seq data analysis, we merged Seurat datasets and constructed a Seurat object following standard protocols without applying batch correction. In the revised Extended Data Fig. 9e, we downloaded and analyzed published data. We have added a new Methods section to describe the analysis (page 27 and page 28). We have also deposited processed Seurat objects and custom codes for analysis to Zenodo and Github for reference.

2. The staining of spatial distribution of microglia during development hMGE is state of art. An open question here: how to explain the increasing of microglia in both Isvz and Osvz from GW15 to GW40. Is it because of the increasing numbers of microglia during development or because the microglia migration to VZ during MGE development?

This is an excellent question. Due to our inability to directly observe microglial behavior in the developing hMGE, we cannot draw a definitive conclusion. We presented our limited understanding below and in the manuscript (line 4-7, page 5; line 20-32, page 5):

1. Human microglia demonstrate low proliferation capacity at this developmental stage according to previous studies (Menassa et al., *Dev Cell*, 2022, PMID: 35977545). Indeed, our snRNAseq results showed that 1.64% (75/4561) and 0.15% (14/9631) microglia were *MKI67* (Ki-67) positive at GW23-30 and GW40, respectively. We did not observe any Ki-67⁺ microglia in the GW23 MGE from our staining (n=3). These data challenge proliferation as the primary mechanism of microglia accumulation in both iSVZ and oSVZ.
2. We examined the distribution of iMG in MGEs at 1, 3, 5, 8, 10, and 14 days post-transplantation (dpt) (Extended Data Fig. 8e). We observed that iMG invaded MGEs after 24 hours, at 1 day post transplantation (dpt), and have reached organoid centers at 5 dpt. The preferential accumulation around rosette-like proliferative centers became apparent at 8-14 dpt (Extended Data Fig. 8e).
3. To further investigate whether proliferating progenitors promote microglial chemotaxis towards the proliferating zone, we blocked cell proliferation by administering Notch pathway inhibitor DAPT (Ciceri et al., *Nature*, 2024, PMID: 38297124) from 10 dpt to 14 dpt. DAPT treatment effectively eliminated rosette formation and Ki-67⁺ cells. Astonishingly, we observed iMGs became evenly distributed throughout the organoids (Fig. 3n-q). Thus, our results support that microglia are likely to migrate to SVZ of hMGE and the proliferating zone of MGEs in response to proliferating progenitors.

Referee #2 (Remarks to the Author):

This study focuses on the function of microglia-derived IGF1 in the proliferation of human medial ganglionic eminence (MGE) progenitors and the generation of GABAergic interneurons. However, the findings regarding the function of microglia and microglia-specific IGF-1 in brain progenitor proliferation and neurogenesis are consistent with earlier research that have reached similar conclusions in mouse models.

Microglia's role in neural progenitor regulation has been supported through several studies, where conditional depletion of microglia leads to the decrease in number of Tbr2⁺ intermediate progenitors in the SVZ (Arno et al., 2014), neural precursor cells (Antony et al., 2011; Yamamiya et al., 2019), and oligodendrocyte progenitors (Hagemeyer et al., 2017). The role of microglia in regulating PV⁺ interneuron development has also been shown through the work of Squarzoni et al. (2014) and Yu et al. (2022).

Furthermore, IGF-1 has been shown to play a role in proliferation, where overexpression of IGF-1 results in increased brain size and underexpression of IGF-1 leads to a decrease in brain size (Popken et al., 2004; Rusin et al., 2024). IGF-1 enhances neural stem cell proliferation in vitro (Supeno et al., 2013), and *Igf-1*^{-/-} mice show decreased cell proliferation in the subventricular zone of the lateral ventricle (Hurtado-Chong et al., 2009). IGF-1 has also been indicated to promote neural precursor cell differentiation into ganglionic eminence GABAergic neurons in vivo through the PDK1-Akt pathway (Oishi et al., 2009).

The current manuscript offers valuable insights, but due to previous literature, further work is necessary to meet Nature's publication standards. For instance, the authors have proposed an intriguing

potential for a species-specific mechanism involving microglia and microglia-derived IGF1 in neural progenitor proliferation and interneuron production. Conducting additional experiments to explore the species-specific mechanisms of microglia and microglia-derived IGF1 in humans compared to mice would significantly enhance the novelty and impact of this study.

We thank the reviewer for the insightful comments and valuable suggestions. To address the species-specific involvement of microglial IGF1 in MGE neurogenesis, we first examined the expression of IGF1 protein in mouse microglia. Consistent with literature, we saw that IGF1 is expressed by axon tract-associated microglia (ATM) (Hammond et al., *Immunity*, 2019, PMID: 30471926) in the developing white matter at P5 (Extended Data Fig. 19a) and ATM-like microglia (Lawrence et al., *Cell*, 2024, PMID: 38309258) at cortico-striato-amygdalar boundary at E14.5 (Extended Data Fig. 19b), but is not detectable in microglia located in the mouse MGE at E14.5 (Extended Data Fig. 19c). Indeed, mining published cross-species single-cell data revealed that IGF1 is expressed at significantly higher levels in human microglia compared to rodents (Geirsdottir et al., *Cell*, 2019, PMID: 31835035) (Extended Data Fig. 19d). Next, we investigated MGE proliferation in microglial *Igf1* conditional knockout (cKO) mice (*Igf1^{fl/fl}, Cx3cr1^{CreERT/+}*). We injected tamoxifen at E11.5 and E12.5 to induce *Igf1* deletion, followed by EdU pulse labelling of proliferating progenitors four hours prior to brain collection at E14.5, a peak time for late MGE neurogenesis (Hu et al., *Development*, 2017, PMID: 29089360). Interestingly, microglial *Igf1* cKO didn't affect MGE proliferation at E14.5 (Extended Data Fig. 19e-g). Combined with the differential distribution of microglia and proliferating progenitors in the MGE between humans and mice (Fig. 1 and Extended Data Fig. 1), our results indicate a species-specific mechanism in microglial regulation of MGE proliferation (line 9-25, page 7). We have revised the Discussion accordingly and added two additional paragraphs to more thoroughly contextualize our results (page 7 to 8).

Other concerns:

- A deeper characterization of microglia states in ex vivo tissue or of induced microglia (iMG) within the Microglia-Engrafted Organoid (MGEO) model is necessary to understand their activation state and functional role. This could be accomplished by analyzing specific activation markers, such as TREM2, which is crucial for microglial activation and function. Since microglia can exist in different phenotypic states, as shown in figure 11, these varying states may influence brain progenitor cells differently, especially in terms of their effects on cell proliferation.

We thank the reviewer for the valuable suggestion. We performed additional scRNA-seq analysis of iMG isolated from MGEOs by FACS-based isolation. In our new Extended Data Fig. 9 (line 11-19, page 5 in the main text), we showed the subcluster and developmental trajectory of iMG, a heatmap of iMG subclusters, principal component analysis (PCA) of iMG compared to selected published microglia transcriptomic datasets, and feature plots of key microglial markers, including *TREM2*. Our analysis revealed subclusters of iMG including proliferating, non-homeostatic/pre-mature, homeostatic, and neuron-associate iMG. The expression of IGF1 remains consistent across all subclusters.

- It also remains possible that the microglia's proximity to proliferative progenitors is coincidental and does not necessarily indicate a direct functional role. Microglia may be attracted to these regions for reasons unrelated to progenitor proliferation, such as immune surveillance or debris phagocytosis. Additional experimental controls are required to eliminate these alternative explanations and confirm any direct influence microglia may have on progenitor cell activity. It would be beneficial to perform live imaging approaches to get real-time information on the functional interactions between microglia

and progenitor cells, and show how microglia migrate to proliferative zones, interact with progenitor cells, and maybe impact their division processes.

This is an excellent and challenging question. It partially overlaps with minor comment #2 from reviewer #1. In addition to our responses to reviewer #1 above (Minor comment#2; main text: line 4-7, page 5; line 20-32, page 5; Fig. 3n-q, Extended Data Fig. 8e), we present our limited understanding from the following perspectives:

1. The center of organoids is enriched with dead cells and cell debris (Amiri et al., *Science*, 2018, PMID: 30545853; Pollen et al., *Cell*, 2019, PMID: 30735633; Lancaster et al., *Nature*, 2013, PMID: 23995685). In our results, most rosette-like proliferating centers and progenitors were distributed closer to the surface of MGEOs (Fig 3b, h, j, o-q; Fig 4g, I; Fig 5d, f). The relative enrichment of iMG around rosettes at the edge, but not the center of MGEOs where dead cells are in abundance (Fig. 3o, q), suggests debris phagocytosis is unlikely to play a dominating role in iMG distribution. However, we have no ability to evaluate/manipulate microglial immune surveillance in our organoids.
2. We generated MGEOs using NKX2.1-GFP iPSCs and transplanted with tdTomato-labelled iMG to conduct time-lapse live imaging to explore the interaction between iMG and NKX2.1-expressing cells. However, the large and spherical shape of MGEOs makes them unsuitable for long time live imaging. They cannot be sliced, as slicing can alter the status and behavior of iMG. Additionally, the long cell cycle of neural progenitors (9-25 hours (Molina et al., *Development*, 2022, PMID: 35588250), ~25 hours (Brandt, Hubner, and Storch. *Stem Cells*, 2012, PMID: 22987479), 36 ± 7 h (Subramanian et al., *Nat Commun*, 2017, PMID: 28139695)) and the rapid dynamics of iMGs further limit conclusive observation. Nonetheless, we were able to capture up to 10 hours of iMG interaction with NKX2.1-GFP progenitors. In one instance, we observed an NKX2.1-expressing cell dividing after direct contact from microglia (Extended Data Fig. 13 and Extended Data video 1; main text: 10-14, page 6).
3. We acknowledged the limitations of our study “Due to limitations in current technologies, we were unable to clarify the real-time dynamics of microglial migration into hMGE proliferative zones, their live interactions with progenitor cells, and their direct effects on progenitor division.” in the last paragraph of the main text (line 44-46, page 8)

- For figure. 4d, it is not clear how the analysis was done. It would be helpful to clarify if the organoids were pooled or the data is showing individual organoids to represent the reproducibility. The standard deviations in this panel are also missing.

We performed a chi-square test using scRNA-seq data generated from a library pooled from 12–16 organoids (described in Methods). The test was described at the end part of the figure legend “ χ^2 test, $1146(\text{Ki-67}^+ \text{ cell number})/9456(\text{overall cell number})$ vs $2269/11680$, $\chi^2=206.0$ in (d);” This chi-square test didn’t generate standard deviations. We further added “the χ^2 tests in (c) and (d) are based on the fractions of targeted cells among the total cells recovered in scRNA-seq data from 6-week-old organoids;” in the legend of Fig. 4 (line 11-12, page 16) as suggested to make it clearer.

- For figure. 5e, it would be nice to show the comparison between the second(+PBS) and the third(-IGF1) conditions.

We revised Fig. 5e as suggested. The comparison is not statistically significant ($P = 0.3180$).

- For figure. 5g, please show the comparison between the first condition (no iMG) and the last condition (IGF1 KO iMG+ IGF1)

We revised Fig. 5g as suggested. The comparison shows a significant difference ($*P < 0.05$).

Referee #3 (Remarks to the Author):

In their manuscript the authors address an important question related to the role of microglia in the development of GABAergic interneurons. They first investigate human samples by using immunohistochemistry and found that microglia associate with neurogenic radial glia in the ganglionic eminences (where interneurons originate) and around DENs (DCx+ cells-Enriched Nests). Next they performed single cell RNA sequencing and by using curated database found an enrichment of IGF1 expression in microglia (besides mature interneurons), and enrichment of IGF1R in radial glia and interneurons. The authors then hypothesized that IGF1-IGF1R signaling could play a role in radial glia-mediated interneuron generation. To this end they first established organoid-based approaches and cultured hPSC-derived MGE organoids (MGEOs) with and without addition of hESC-induced iMGs. Strikingly, the addition of iMGs increased proliferation and generation of interneurons in MGEOs. They then tested the role of IGF1 and found that addition of IGF1 increased and blocking IGF1 decreased proliferating cell density. Lastly, the authors knocked out IGF1 in iMGs and could demonstrate that the proliferation-promoting effect of iMGs in MEGOs was lost. The authors provide a model how microglia-derived IGF1 promotes the proliferation of interneuron-generating radial glia and proliferating neuroblasts in DENs.

This study provides interesting data relevant for our understanding of the role of microglia in human interneuron development. The data is convincing and the presentation of the study is quite neat. A few questions have come up while assessing this manuscript.

1. Is there a role for microglia-derived IGF1 in promoting the proliferation of radial glia that generate cortical projection neurons in human, or is the effect specific for the generation of interneurons?

This is a good question. To investigate whether microglia-derived IGF1 promotes the progenitor proliferation of cortical radial glia, we generated cortical neuroimmune organoids and treated 6-week-old organoids with PBS, IGF1, or anti-IGF1. iMG transplantation and IGF1 treatment significantly increased the density of Pax6⁺BrdU⁺ cells, whereas anti-IGF1 significantly reduced it, suggesting that iMG and iMG-derived IGF1 enhance proliferation of PAX6⁺ neural progenitors (cells that give rise to cortical projection neurons) in cortical organoids (Extended Data Fig. 18; line 3-8, page 7).

Previous studies have reported conflicting findings regarding the distribution of microglia in the cortical germinal zones. Cunningham et al. (2013) found that microglia are enriched in the iSVZ of the human cortex at GW10 (Cunningham, Martinez-Cerdeno, and Noctor. *J Neurosci*, 2013, PMID: 23467340). In contrast, Menassa et al. (2022) reported that microglia are more prevalent in the pre-subplate GW 9–12 and become enriched in the ventricular zone from GW16–26 (Menassa et al., *Dev Cell*, 2022, PMID: 35977545). Additionally, Cunningham et al. (2013) demonstrated that microglial distribution varies significantly across cortical areas in macaques, suggesting that cortical heterogeneity may contribute to discrepancies among human studies. Therefore, although our organoid experiments suggest a potential role for microglia-derived IGF1 in cortical progenitor proliferation, we cautiously discussed it in the Discussion section (line 44, page 7- line 4, page 8).

2. The authors knocked out IGF1 in microglia, does the reverse experiment (i.e. knocking out IGF1R in radial glia and/or DENs) support their claims?

It is a good suggestion. We attempted to generate *IGF1R* knockout iPSC lines using CRISPR/Cas9-based non-homologous end joining—the same technique we used to generate an *IGF1* knockout cell line. We recovered and Sanger-sequenced 115 colonies: 47 colonies were wild-type, 68 carried heterogeneous mutations, but none exhibited a homozygous *IGF1R* loss-of-function mutation. To circumvent this challenge, we took chemical inhibitor approach by treating MGE neuroimmune organoids with two validated IGF1R inhibitors, Picropodophyllin (Li et al., *Cell Rep Med*, 2025, PMID: 39914384; Gao et al., *Cell Rep*, 2022, PMID: 35385750) and GSK4529 (Kang et al., *Cell Death Dis*, 2012, PMID: 22739988). Our results showed that both inhibitors abolished microglia-induced MGE proliferation, supporting the involvement of IGF1R in this effect (Extended data Fig. 17; line 45-46, page 6).

3. Related to the above point, IGF2 has been shown to have a role in (postnatal) neurogenesis (cf e.g. studies from Ferron laboratory). Is there evidence (expression in human dataset) that IGF2 might also influence interneuron generation? Regardless of the finding, a slightly broader discussion about the role of IGF signaling in neurogenesis might provide some more integrated context of the findings.

We did not detect any *IGF2* expression in microglia but observed it in endothelial cells, consistent with the findings from literature (Ferron et al., *Nat Commun*, 2015, PMID: 26369386; Lehtinen et al., *Neuron*, 2011, PMID: 21382550). We included feature and violin plots in this revised manuscript showing the expression patterns of *IGF2* and *IGF2R* (Extended Data Fig. 6). Additionally, we expanded the Discussion to address the broader role of IGF signaling in neurogenesis, with specific mention of IGF2 in line 5-11, page 8.

4. The authors show that the density of microglia is increased in mouse when compared to human. They suggest a species-specific feature underlying the difference. This is an interesting point since further investigation of the mechanisms how IGF1 might promote the proliferation of interneuron-generating radial glia could be enabled by further genetic studies in the mouse. Therefore, it would be interesting to assess whether microglia-derived IGF1 also promotes the proliferation of radial glia in mouse MGE? Assessing the generality of the findings across species would be important in order to know if the observed role for microglia-derived IGF1 is a human specific feature or conserved across evolution.

We thank the reviewer for the insightful comments. This comment is similar to the major concern raised by reviewer #2. Please refer to our new data and our detailed response above (Extended Data Fig. 19; line 9-25, page 7).

Sincerely,

Xianhua Piao, MD, PhD
Benioff Professor in Children's Health
Director of the Newborn Brain Research Institute
University of California, San Francisco

We appreciate the positive feedback from all referees and are grateful for the constructive comments provided. We respectfully submit our point-by-point responses to all reviewer comments below.

Referees' comments:

Referee #1 (Remarks to the Author):

The authors have substantially strengthened their manuscript by addressing the outstanding concerns with additional experiments.

#1: Species-specific IGF1/IGF1R signaling in human interneuron genesis.

The concern was based on previous publications of mouse neuronal development. IGF1 is not specific to interneuron progenitor proliferations, and an elaborate body of literature from mouse studies has demonstrated roles of IGF1 during neurogenesis. In response, the authors carried out cross-species single cell analysis from published datasets, together with IHC studies. The authors found IGF1 expression in some microglia states, but did not detect IGF1 in MGE microglia during development. Moreover, the authors generated microglia specific conditional knock out of Igf1 in mice. By IHC studies, the authors found that Igf1 is not involved in MGE development in mouse microglia.

#2: The specific enrichment of microglia in SVZ.

This concern came from our open question, which was challenging to answer from direct studies in human brain tissues. The authors analyzed the single cell RNAseq from hMGEs, identified limited numbers of proliferating microglia in MGEs. Hence, they proposed that microglia enrichment in SVZ is not because of proliferation, but because of microglial migration. They later used Notch signaling inhibitor DAPT to block neural progenitor proliferation and found evenly distributed microglia in MGEO.

#3: Is IGF1/IGF1R MGE specific?

The authors also generated cortical organoids, that treated with PBS, IGF1 or anti-IGF1, and found that iMG and iMG-derived IGF1 increase PAX6+ neural progenitors in cortical organoids.

Overall, the results and conclusions stated in the abstract are of significance and generally well supported by data. The one major recommendation is to revise the below sentence in the abstract to include the word predicted (highlighted), as this relationship is experimentally validated in the organoid but cannot be directly shown in the human brain:

We show that microglia preferentially distribute in the proliferating zone and identify insulin-like growth factor 1 (IGF1) and its receptor IGR1R as the *predicted* top ligand-receptor pair underlying microglia-progenitor communication in the prenatal human brain.

We revised our abstract/summary paragraph as suggested.

We note some additional minor concerns:

1. Extended Figure 4, 5 need more detailed figure legends on the cross-talk analysis.

We added more detailed information in legends of extended data Fig. 4 and Fig. 5 as suggested (page 26).

2. The number of biological replicates of human samples that were used for the single cell analysis was not clear. .

We specified the use of 'N=6 donors' in the main text (line 34, page 2). Additional information about the donors and experimental design is provided in Extended Data Table 1. UMAP data for each biological sample and experimental condition are shown in Extended Data Figure 3.

3. Extended figure 19 d. It's not advisable to compare UMI/cells from different single cell or single nuclei RNAseq of one gene. UMI/cell relies heavily on library preparation, sequencing depth and batch effects. It's difficult to draw the conclusion that IGF1 is expressed at significant higher levels in human microglia than compared to rodents from this comparison alone.

We agree with the comment. Since this figure panel presents non-essential evidence that is better contextualized and readily accessible in the original publication (Geirsdottir et al., *Cell*, 2019, PMID: 31835035), we have removed this panel, while better describe and cite the original publication in the revised manuscript (line 15-17, page 6)

4. Extended Figure 16f, comparison should also be made of between control and IGF1KO. We conducted the comparison ($P=0.5555$) and labelled it in the figure.

Referee #2 (Remarks to the Author):

I enjoyed reading the revised version of the manuscript. All major concerns have been addressed. As a minor note, it would be very helpful if the authors could clarify which state of human primary microglia (ramified or polarized) is used in the morphological analysis presented in Extended Figure 9g-j. This clarification would provide additional information on whether a specific microglial state is involved in progenitor proliferation and neurogenesis.

We added the statement "Primary microglia from both the oSVZ and iSVZ were included in the analysis." to the figure legend (line 1-2, page 28).

Referee #3 (Remarks to the Author):

The authors have done a great job for their revision. They addressed all concerns of all reviewers very well and also provided new and convincing data that enhance the overall conceptual advance. It is quite intriguing to see the significance of species-specific differences (mouse vs human) in their new data set. In my opinion the findings should be reported in a timely manner.